

# Combining hyperspectral remote sensing and eddy covariance data streams for estimation of vegetation functional traits.

Javier Pacheco-Labrador[1], Tarek S. El-Madany[1], M. Pilar Martin[2], Rosario Gonzalez-Cascon[3], Arnaud Carrara[4], Gerardo Moreno[5], Oscar Perez-Priego[6], Tiana Hammer[1], Heiko Moossen[1], Kathrin Henkel[1], Olaf Kolle[1], David Martini[1], Vicente Burchard[2], Christiaan van der Tol[7], Karl Segl[8], Markus Reichstein[1] and Mirco Migliavacca[1].

[1]Max Planck Institute for Biogeochemistry, Hans Knöll Straße 10, Jena, D-07745, Germany
[2]Environmental Remote Sensing and Spectroscopy Laboratory (SpecLab), Institute of Economic, Geography and Demography (IEGD-CCHS), Spanish National Research Council (CSIC), C/Albasanz 26-28, 28037 Madrid, Spain
[3]Department of Environment, National Institute for Agriculture and Food Research and Technology (INIA), Ctra. Coruña, Km. 7,5, 28040 Madrid, Spain
[4]Fundación Centro de Estudios Ambientales del Mediterráneo (CEAM), Charles Darwin 14, Parc Tecnològic, 46980 Paterna, Spain
[5]Forest Research Group - INDEHESA University of Extremadura, 10600 Plasencia, Spain
[6]Department of Biological Sciences Macquarie University, 6 Wally's Walk, NSW 2109, Australia.
[7]Faculty of Geo-Information Science and Earth Observation (ITC), University of Twente, PO Box 217, AE Enschede 7500, The Netherlands
[8]Helmholtz Centre Potsdam–GFZ German Research Centre for Geosciences, Section 1.4 Remote Sensing, Telegrafenberg, 14473 Potsdam, Germany

*Correspondence to*: Javier Pacheco-Labrador (jpacheco@bgc-jena.mpg.de)

**Abstract.** Remote Sensing (RS) has traditionally provided estimates of key biophysical properties controlling light interaction with the canopy (e.g., chlorophyll content ($C_{ab}$) or leaf area index ($LAI$)). However, recent and upcoming developments in hyperspectral RS are expected to lead to a new generation of products such as vegetation functional traits that control leaf carbon and water gas exchange. This information is pivotal to improve our understanding and capability to predict biosphere-atmosphere fluxes at global scale. Yet, the retrieval of key functional traits such as maximum carboxylation rate ($V_{cmax}$) or the Ball-Berry stomatal sensitivity parameter ($m$) remains challenging, as they only have a weak and indirect influence on optical reflectance factors. Recently, the assimilation of different observations in coupled soil-vegetation-atmosphere transfer (SVAT) and radiative transfer models (RTM) is allowing $V_{cmax}$ and $m$ estimates; notably using the Soil Canopy Observation of Photosynthesis and Energy fluxes (SCOPE) model. In this work we assess the potential of airborne and satellite emulated hyperspectral imagery jointly with eddy covariance (EC) data for the retrieval of functional traits. Specifically, we made use of time series of gross primary production ($GPP$) and thermal irradiance measured with net radiometers, together with 17 hyperspectral airborne images. The potential of satellite-borne sensors was tested with emulated EnMAP imagery from the airborne data. EnMAP was selected because of the availability of the emulator, and because is one of the foreseen hyperspectral satellite missions expected to contribute to a new generation of RS products. We estimated ecosystem functional traits by inverting the senSCOPE model, a novel version of SCOPE





adapted to represent partly senescent canopies. The experiment takes place in a Mediterranean tree-grass ecosystem subject of a large scale manipulation experiment with nitrogen and nitrogen plus phosphorus, monitored by three EC towers. Parameter estimates and predicted fluxes were evaluated using both ground observations and pattern-oriented model
evaluation approach. The method developed in this study provided robust estimates of functional and biophysical parameters for both airborne and synthetic EnMAP datasets. $C_{ab}$ and $V_{cmax}$ estimates followed observed relationships with leaf nitrogen concentration; whereas $m$ and predicted underlying water use efficiency showed expected relationships with discrimination of $^{13}$C isotope in leaves. Results prove that the inversion of coupled RTM-SVAT models against a combination of hyperspectral imagery (e.g., EnMAP), and time series of *GPP* and thermal irradiance provides reliable estimates of key
functional parameters of vegetation that are robust to several sources of uncertainty. The forthcoming satellite hyperspectral missions combined with ecosystem station networks (e.g. Integrated Carbon Observation System (ICOS), NEON, FLUXNET, etc…), offers unique possibilities to characterize the spatiotemporal distribution of functional parameters relevant for terrestrial biosphere modeling.

## 1 Introduction

Accurate representation of terrestrial carbon and water fluxes is critical to adequately understand and monitor the response of ecosystems to climate change. These fluxes are continuously monitored in limited areas of Earth surface by a growing network of Eddy Covariance (EC) stations (Baldocchi, 2008, 2014), which is the reference for the calibration and validation of several terrestrial biosphere models (TBM) predicting such fluxes at global scale (Friend et al., 2007;Schwalm et al., 2010). These models, however, still incur in large prediction errors which are partly due to the lack of spatiotemporal
information on key plant functional traits such as the maximum carboxylation rate ($V_{cmax}$), or the Ball-Berry stomatal sensitivity ($m$) among others (Rogers, 2014;Rogers et al., 2016;Schaefer et al., 2012). This lack of information is most commonly addressed by assigning typified values according to different plant functional types (PFT). However, it has been proved that dynamic modeling (both in space and in time) of these parameters can improve the accuracy and precision of TBM predictions (Bonan et al., 2011;Rogers, 2014;Walker et al., 2017;Luo et al., 2019). Estimates of plant functional traits
usually rely on semi-empirical models and / or empirical relationships derived from global databases (Walker et al., 2014;Rogers, 2014;Bonan et al., 2011;Wullschleger et al., 2014), or from the inversion of mechanistic models against observations of these parameters (Ali et al., 2016). They are also directly estimated inverting the models against fluxes (Zheng et al., 2017;Zhou et al., 2014;Reichstein et al., 2003). Some early works used Remote Sensing (RS) information such as surface albedo to constrain models (Alton, 2011), or to evaluate estimates of functional parameters constrained with EC
data (Zhou et al., 2014).

RS provides global coverage of Earth's surface and therefore offers great potential to provide global information about plant function. A new generation of sensors with enhanced spectral, spatial and temporal resolutions is expected to lead to new products of vegetation functional traits; which will require the development of new algorithms, models and data assimilation





schemes (Schimel et al., 2019). Early research of the RS community on this topic focused on the estimation of Light Use

Efficiency (*LUE*) or other plant stress indicators such as stomatal conductance, leaf water potential or canopy temperature (Drolet et al., 2005;Hernández-Clemente et al., 2011;Suárez et al., 2009;Zarco-Tejada et al., 2013); mainly exploiting the Photochemical Reflectance Index, *PRI* (Gamon et al., 1992). Recent works have used hyperspectral information and statistical analyses (e.g., partial least squares regression) to estimate *LUE* (Huemmrich et al., 2019), $V_{cmax}$ (Serbin et al., 2015;Silva-Perez et al., 2018;Dechant et al., 2017) or directly gross primary production (*GPP*) (DuBois et al., 2018).

Alternatively, remote estimates of chlorophyll sun induced fluorescence (SIF) have been used to predict *GPP* (Frankenberg et al., 2011;Guanter et al., 2012;Joiner et al., 2013;Sun et al., 2017) and transpiration (Alemohammad et al., 2017;Lu et al., 2018;Shan et al., 2019). Also RS in the thermal infrared (TIR) domain has been used to infer plant functioning. Leaf temperature influences photosynthetic efficiency (Farquhar et al., 1980;Hikosaka et al., 1999;Bunce, 2000;Farquhar and von Caemmerer, 1982), and it is in part controlled by leaves transpiration, regulated in turn by stomatal conductance (Gates,

1968;Pallas et al., 1967). Therefore, thermal RS has been used to determine vegetation stress (Zarco-Tejada et al., 2012;Zarco-Tejada et al., 2013;Jarolmasjed et al., 2018;Sepulcre-Cantó et al., 2006). However, most of these works have relied on empirical relationships between functional and RS variables; which are difficult to generalize and hard to transfer between sensors (e.g., Hill et al., (2006)).

RS of vegetation function faces the fact that optical and TIR signals that are mechanistically related with photosynthesis and

transpiration originate inside the leaf. Therefore they vary within the canopy (light or wind speed gradients, among others) and are modified in their transfer from leaves to top of the canopy (TOC), according to sun-view geometry and vegetation structure. This complicates the application of empirical methods to understand or generalize the connection between RS observations and plant function. For this reason, there is an increasing on the use of models physically describing both the radiative transfer and physiological process. Due to the complexity of these processes and the weak influence that some

functional traits have on RS observations, some authors argued about the need to jointly use RS and EC data for their retrieval. For instance, Pacheco-Labrador,  et al., (2019) showed the successful retrieval of $V_{cmax}$, *m*, and Cab, jointly constraining the Soil Canopy Observation of Photosynthesis and Energy fluxes (SCOPE) (van der Tol et al., 2009) model using with chamber-based fluxes and proximal sensing data. In fact, in the last years the interest of two different communities seems to converge. TBM have used RS data to constrain traits or to inform biophysical variables into models

(Alton, 2017;Xie et al., 2018) whereas RTM have been also coupled to simplified vegetation dynamic (Koetz et al., 2005) or physiological models (Xin et al., 2015), as well as to TBM (Migliavacca et al., 2009) or crop models (Thorp et al., 2012;Dorigo et al., 2007) in order to assimilate RS observations. However, to our knowledge an attempt to jointly retrieve functional traits using hyperspectral imagery combined with EC data is lacking in the literature.

Nowadays, some RTM simulate optical (e.g., *PRI*, SIF) and / or TIR signals related with plant photosynthesis taking place at

leaf level (van der Tol et al., 2009;Vilfan et al., 2018;Yang et al., 2017;Hernández-Clemente et al., 2017). However, this modeling is not informative of the physiological processes originating such signals. This connection can only be achieved coupling *RTM* to models representing such processes as for example the soil-vegetation-atmosphere transfer models



(SVAT). The state-of-the-art model coupling optical, SIF and TIR RTM with energy balance and photosynthesis models is SCOPE (van der Tol et al., 2009). Further improvements of this model are mSCOPE (Yang et al., 2017), which allows

representing vertically heterogeneous canopies, and senSCOPE (Pacheco-Labrador et al., 2020), which improves the representation of canopies featuring mixed green and senescent leaves (Pacheco-Labrador et al., 2020). Using satellite imagery, SCOPE has been used to obtain estimates of physiological parameters of vegetation such as $V_{cmax}$ and / or $m$ exploiting reflectance factors ($R_\lambda$, where $\lambda$ denotes spectral) and SIF (Zhang et al., 2014), $R_\lambda$, SIF and EC fluxes (Zhang et al., 2018), $R_\lambda$ and TIR data (Bayat et al., 2018), or $R_\lambda$ and EC fluxes (Dutta et al., 2019). Also, proximal sensing data have been

used to constrain SCOPE providing estimates of functional parameters (Pacheco-Labrador et al., 2019;Hu et al., 2018); and more recently, airborne imagery has been used to retrieve $V_{cmax}$ from $R_\lambda$ and SIF (Camino et al., 2019).

From all these works, only Camino et al, (2019) validated their retrievals against actual measurements of functional traits ($V_{cmax}$), as gas exchange leaf-level measurements usually needed to obtain validation data of functional parameters are time consuming and are not feasible for large areas and / or diverse natural ecosystems. In these circumstances, pattern-oriented

model evaluation has been used to assess the suitability of different models and parameter estimates (Carvalhais et al., 2014;Reichstein et al., 2011;Migliavacca et al., 2013;Grimm and Railsback, 2012;Luo et al., 2012). Pacheco-Labrador et al, (2019) used pattern-oriented model evaluation approach to assess the suitability of functional and biophysical estimates obtained by SCOPE and senSCOPE model inversion against ground observations. On one hand, the relationships between variables measured at canopy scale (e.g., nitrogen concentration) with parameter estimates (e.g., $V_{cmax}$ and chlorophyll

content ($C_{ab}$)) were compared with relationships expected from the theory or the literature (e.g. Feng and Dietze (2013)). On the other hand, variables derived from predicted fluxes (e.g., evaporative fraction ($EF$)) were compared with observations from a nearby EC system. Such evaluations provided relevant information about models structure and ill-posed solutions of their inversion.

The new generation of hyperspectral satellite-borne sensors (Rast and Painter, 2019) brings new possibilities for the

spatiotemporal characterization functional traits of vegetation, even though the algorithms to convert these data into information must be developed and evaluated. Some of these sensors are already operational (DESIS (Kerr et al., 2016), PRISMA (Galeazzi et al., 2008)); or will be in the next few years (e.g., EnMAP (Guanter et al., 2015), HyspIRI (Lee et al., 2015), etc. see Rast and Painter (2019) for a complete overview). Meanwhile up-coming missions also offer emulation capabilities (e.g., the EnMAP End-to-End simulator (Segl et al., 2012)); which allows the evaluation of their potential for

different applications before they operate.

In this work we aim at developing a methodology to use of hyperspectral remote sensing in combination with EC stations to estimate functional and biophysical parameters of vegetation at the ecosystem scale. We constrain senSCOPE model with a combination of multiple hyperspectral $R_\lambda$ in different phenological periods, diel tower-based TIR observations and $GPP$ estimates in a Mediterranean tree-grass ecosystem manipulated with nitrogen (N) and N plus phosphorus (P). The inversion

is tested both on airborne as well as on synthetic EnMAP hyperspectral imagery, simulated to increase uncertainties and



down-grade resolutions to levels expected in a space-borne sensor. Biophysical and functional parameters are assessed using ground observations and pattern-oriented model evaluation approach.

## 2 Methods

### 2.1 Study site

The study area is located in Majadas de Tiétar (39º56'24.68'' N, 5º45'50.27'' W; Cáceres, Spain). This is a Mediterranean tree-grass ecosystem where a large scale manipulation experiment with N and P was conducted at the ecosystem scale (MANIP, see El-Madany et al., (2018) and Nair et al., (2019)). Three EC towers monitor three areas under different manipulation regimes. The first one (CT) operates since 2003 and is used as control, the second and the third ones operate since 2014 and their footprints -~ 24 ha-, have been fertilized with N (NT) and N+P (NPT) respectively. Towers are

sufficiently separated (~500 m) to prevent 80 % iso-lines of their respective footprint climatology to overlap (El-Madany et al., 2018). These areas were fertilized twice: 50 and 10 kg P ha$^{-1}$ in the form of triple superphosphate (Ca(H$_2$PO$_4$)$_2$) were applied in November 2014 and March 2016, respectively in NPT. 100 and 20 kg N ha$^{-1}$ were applied in March 2015 and March 2016 in the form of ammonium nitrate (NH$_4$NO$_3$) in NPT, while in NT was in form of calcium ammonium nitrate (5Ca(NO$_3$)$_2$NH$_4$NO$_3$) to compensate the Ca added in NPT with the P fertilizer. Fertilization modified species, ecosystem-

level biophysical and functional properties and fluxes (Nair et al., 2019).

   The climate is Continental Mediterranean, mean annual temperature and precipitations are 16.7 ºC and ~650 mm, respectively. There is a strong seasonality with hot and dry summers reaching ~40 ºC of maximum daily temperature, and with scarce precipitation - less than the 6 % of the annual rainfall (Casals et al., 2009). The ecosystem combines two vegetation layers: grass and trees. The grassland layer is composed of a large number of annual species comprehending three

main PFT (grasses, forbs and legumes, see Migliavacca et al, (2017)), and supports low grazing pressure (<0.3 cows / ha). It features a strong phenological behavior controlled by radiation and water availability in spring-summer, which produces a mayor and a minor greening peaks in spring and autumn, respectively; and a dry season in summer (Luo et al., 2018). Replacement of species and mortality accumulates senescent and death standing material through the year, which can already represent up to the 30 % of the leaf area before the beginning of the dry season. The ecosystem is composed by scattered

trees, mainly *Quercus ilex* L. subsp. *ballota* [Desf.] Samp (Evergreen Holm Oak). Trees fractional cover is ~20 %, average tree distance, tree height, horizontal and vertical crown radius are 18.8 m (standard deviation, $\sigma = 5.0$ m) and 7.9 m ($\sigma = 0.9$ m), 4.2 m ($\sigma = 0.9$ m) and 2.7 m ($\sigma = 0.9$ m), respectively (Pacheco-Labrador et al., 2016). Evergreen Holm oak leaves partially renovate every spring so leaves of up to three cohorts can coexist within the same crown. Some years, new leaves can sprout twice, though biophysical and spectral properties leaves from different cohorts are quite similar after the first 9

months of development (González-Cascón et al., 2016;Gonzalez-Cascon et al., 2018;Pacheco-Labrador et al., 2017b).



## 2.2 Hyperspectral Airborne and EnMAP synthetic imagery

Airborne images were acquired in seven different campaigns that took place in the study site between May 2011 and May 2017 covering the spring growing and the summer dry seasons (Table 1). A total of 17 sites-images were used for the analysis. Imagery was acquired by the Compact Airborne Spectrographic Imager CASI-1500i (Itres Research Ltd., Calgary,

AB, Canada), operated by the Instituto Nacional de Técnica Aeroespacial (INTA). CASI featured 144 spectral bands, spectral range ~350-1050 nm, full width half maximum ~5.5 nm, and field of view ~40 °. Images were acquired around solar noon, at least always once in the main wind direction (east-northeast and west-southwest) along the longest axis of the EC tower footprints, with a pixel size ~0.90 x 1.66 m approximately. However, for all the campaigns after 2012 also overpasses with different azimuth orientations also were acquired. Images were atmospherically corrected by INTA using ATCOR-4[TM]

(ReSe Applications GmbH, Germany) and improved with empirical line correction (Smith and Milton, 1999) using ground calibration targets measured during flight campaigns with an ASD Fieldspec3[TM] spectroradiometer (Analytical Spectral Devices, Inc., Boulder, CO, USA). For most of the campaigns, ancillary measurements of water vapor and aerosol optic thickness were acquired using a CIMEL CE318-NE sun photometer (Cimel Electronique, Paris, France). Nearest neighbor resampling was used during geometric correction to prevent spectral mixture of different covers; output spatial resolution

was 1 m. For the campaigns after 2012, several overpasses acquired in the main wind direction as well as in the solar principal plane were combined in mosaics featuring pixels of 1.5 m, which were used instead of the individual overpasses to reduce directional effects of the sun-view geometry. The combination of single overpasses and mosaicked is not ideal, since the assimilated data might be subject to different directional effects (see Discussion section). Dates of each overpass can be seen in Table 1.

**Table 1.** Dates of airborne field campaigns and EC towers overpassed.

| Field campaign | Towers | Mosaicked |
|---|---|---|
| 05-May-2011 | CT | No |
| 04-Oct-2012 | CT | No |
| 08-Apr-2014 | CT, NT, NPT | Yes |
| 23-Apr-2015 | CT, NT, NPT | Yes |
| 03-Jul-2015 | CT, NT, NPT | Yes |
| 03-May-2016 | CT, NT, NPT | Yes |
| 19-May-2017 | CT, NT, NPT | Yes |

In order to test the potential of satellite-borne hyperspectral imagery featuring higher uncertainties and coarser spectral and spatial resolutions than airborne data, we used the airborne images as a reference and simulated synthetic EnMAP imagery with the EnMAP End-to-End simulator (Segl et al., 2012), available for this work. Since no other satellite-borne

hyperspectral data was available at the site, the simulation was a reasonable alternative. Synthetic imagery featured EnMAP



spectral configuration (FWHM $\epsilon$ [5.75, 9.81] nm between 423 and 930 nm), spatial resolution (~30 m) and include geometric and atmospheric uncertainties (Guanter et al., 2015). However, in fact these uncertainties were added to those already present in the CASI datasets; which made these images a rougher test for the methodology under evaluation. Squared cut-outs of 1 km long centered on each EC tower were used for the later analyses both for CASI and EnMAP synthetic images.

## 2.3 Eddy Covariance and Footprint Climatology

Three EC towers monitored ecosystem carbon dioxide ($CO_2$), and water vapor concentrations with an enclosed-path infrared gas analyzer (LI-7200, LI-COR Biosciences Inc., Lincoln, NE, USA) positioned at a height of 15 m. 3-D wind velocity components and sonic temperature were measured using an ultra-sonic anemometer (R3-50; Gill Instruments Limited, Lymington, UK) at 20 Hz. Flux processing was conducted with EddyPro (version 5.2.0, LI-COR Biosciences Inc., Lincoln, NE, USA). Short wave and long wave down and up welling irradiances were measured with net radiometers (CNR4, Kipp and Zonen, Delft, Netherlands) also at a height of 15 m. Quality assessment and quality checks were conducted according to Mauder and Foken (2005). Low quality data, and data under low turbulent mixing (Papale et al., 2006) or during rainy events were discarded. *GPP* was computed subtracting ecosystem respiration estimated according to Reichstein et al. (2005) from net ecosystem exchange using the REddyProc R Package (Wutzler et al., 2018). From the final dataset only daytime high-quality GPP data (i.e. coming from half hours where NEE was measured with good quality) were used. Fluxes uncertainty ($\sigma$) was computed from the standard deviation of the marginal distribution sampling of the gap-filling procedure (Reichstein et al., 2005). Footprint climatology was calculated every half-hour according to Kljun et al. (2015) and used later minimize the spatial mismatch between EC footprint and RS imagery. Further details can be found in Perez-Priego et al., (2017) and El-Madany et al., (2018).

## 2.4 Biophysical variables sampling and up-scaling

Biophysical variables were estimated at ecosystem scale for evaluation of parameter estimated via inversion of senSCOPE. Destructive sampling of grass and tree leaves was always performed simultaneously to each airborne campaign, as well as during several additional campaigns between 2009 and 2019 (Melendo-Vega et al., 2018;Mendiguren et al., 2015;Pacheco-Labrador et al., 2014;González-Cascón et al., 2016;Gonzalez-Cascon and Martin, 2018). Grass samples were collected from 25 x 25 cm quadrants (20 to 26 depending on the campaign) of from where leaf area index (*LAI*, $m^2$ $m^{-2}$), leaf water content ($C_w$, g $cm^{-2}$) and dry matter content ($C_{dm}$, g $cm^{-2}$) were determined in the laboratory with gravimetric methods and a scanner following protocols described in Mendiguren et al, (2015) and Melendo-Vega et al, (2018). For the grass layer, after 2012, these variables were separately measured for green and senescent plants; which allowed determining the fraction of green leaf area ($f_{green}$). Trees $f_{green}$ was assumed 1. Tree leaves from the current year and previous years were sampled from the north and south sides of between 5-18 crowns in each campaign; $C_w$ and $C_{dm}$ were determined using the same methods than used for the grasses. Tree *LAI* was determined combining different methods: indirect measurements using the LICOR LAI 2200-C instrument (2018) and direct estimations using litter traps and leaf turnover rates (Melendo-Vega et al., 2018). Due



to lower temporal frequency of tree *LAI* measurements, a seasonal model was used to predict tree *LAI* as a function of DoY combining these data. The reduced tree fraction cover and tree *LAI* variability compared with grasses minimize the

uncertainties introduced by this approach when integrated at ecosystem scale. Nitrogen concentration per total ($N_{mass}$, %) and green unit mass ($N_{mass,green}$, %) was also determined. When missing, seasonal models were used to predict $N_{mass}$ and $N_{mass,green}$ as well as $f_{green}$ (the last two in grasses). Trees $C_{ab}$ and carotenoids content ($C_{ca}$) were determined from SPAD measurements and a model calibrated from samples of the site (Gonzalez-Cascon et al., 2017). From 2016 on, grass $C_{ab}$ and $C_{ar}$ were determined via destructive sampling (Gonzalez-Cascon and Martin, 2018). The relationship found between $C_{ab}$ per unit total

leaf area and $N_{mass}$ (Pacheco-Labrador et al., 2020) was used to estimate grass $C_{ab}$ in the campaigns where it was missing. $N_{mass,green}$ was also used to estimate grass $V_{cmax}$ using the relationship in Feng and Dietze (2013); since no other data were available, a constant $V_{cmax} = 45$ μmol m$^{-2}$ s$^{-1}$ was estimated as an acceptable value for trees according to literature (Vaz et al., 2010;Vaz et al., 2011;Limousin et al., 2010).

Since 2014, the herbaceous layer was sampled in between 6 and 33 quadrants of 30 x 30 cm inside each tower footprint for

determination of isotope signature of plant material ($\delta^{13}$C, ‰) during the airborne campaigns. Tree leaves were sampled inside each tower footprint, a total of 8 trees for each footprint and 4 branches sampled along 4 different cardinal positions. This sampling was done in winter, when the differences in terms of phenology between leave cohorts are minimal across trees and the traits more representative of the average conditions over the season. The carbon isotopic composition of dried samples was analyzed using a Delta$^{Plus}$ isotope ratio mass spectrometer (Thermo Fisher, Bremen, Germany) coupled via a

ConFlowIII open-split to an elemental analyzer (Carlo Erba 1100 CE analyzer; Thermo Fisher Scientific, Rodano, Italy), and measured according to Werner et al. (1999;2001) and Brooks et al. (2003). $\delta^{13}$C was calculated using Eq. 1 (Brand, 2013;Coplen, 2011), scaled to the $\delta^{13}$C$_{IAEA-603-LSVEC}$ scale, based on calibrated in-house-standard (Acetanilide: $-30.06 \pm 0.05$ ‰):

$$\delta^{13}C = \frac{^{13}R_{sample} - ^{13}R_{standard}}{^{13}R_{standard}}, \tag{1}$$

where $^{13}R_{sample}$ and $^{13}R_{standard}$ are $^{13}$C/$^{12}$C ratio of the sample and of the standard, respectively.

All the variables defined per leaf area were up-scaled to ecosystem level according to grass and tree *LAI*, fraction vegetation cover tress and grasses ($f_{VC,grass} = 100$ %, $f_{VC,trees} = 20$ %, respectively), and seasonal estimates of fractions of current and former years leaves for the trees (see Melendo-Vega et al, (2018)). Grass $V_{cmax}$ was scaled using *LAI* in the green fraction. Variables relative to mass, such as $N_{mass}$, $N_{mass,green}$ and $\delta^{13}$C were scaled considering also $C_{dm}$. Scaled $\delta^{13}$C was used to

compute $^{13}$C photosynthetic discrimination ($\Delta^{13}$C) as a function of atmospheric $CO_2$ and plant-measured isotope signature ($\delta^{13}$C$_{atm}$ and $\delta^{13}$C$_{grass}$, respectively) following Eq. 2:

$$\Delta^{13}C = \frac{\delta^{13}C_{atm} - \delta^{13}C_{plant}}{1 + \delta^{13}C_{plant}}, \tag{2}$$



## 2.5 senSCOPE model parametrization

senSCOPE (Pacheco-Labrador et al., 2020) is an extended version of the model SCOPE that separates radiative transfer,

energy balance and photosynthesis for green and senescent leaves randomly mixed within a homogeneous canopy. While SCOPE parametrizes a single leaf type, senSCOPE requires leaf parameters for both senescent and green leaves; where senescent leaves only present senescent pigments and green ones feature any pigment but senescent. Like in Pacheco-Labrador et al., (2020), during inversion we assumed the same $C_{dm}$ for both leaf types, but that $C_w$ of green leaves is four times higher than in senescent leaves (Kidnie et al., 2015). These assumptions allowed us to later invert the model

optimizing averaged leaf parameters rather than the individual parameters of each leaf type. senSCOPE includes $f_{green}$, representing the fraction of leaf area corresponding to green leaves; as in Pacheco-Labrador (2020) we used a neural network model to predict $f_{green}$ as a function of the averaged leaf parameters ($X$). Therefore, during inversion $f_{green}$ is predicted as a function of the averaged leaf parameters, and then the parameters corresponding to green and senescent leaves ($X_{green}$ and $X_{senes}$, respectively) are calculated solving Eq. 3:

$$X = X_{green} \cdot f_{green} + r \cdot X_{senes} \cdot f_{senes} ,\qquad\qquad(3)$$

where $r$ is the ratio $X_{senes}/X_{green}$, which is different for each parameter according to the assumptions described above; and $f_{senes}$ = 1- $f_{green}$ is the fraction of leaf area corresponding to senescent leaves. The neural network predicting $f_{green}$ was trained over a look-up table (LUT) of 5000 samples generated via Latin Hypercube Sampling of different leaf parameters and $f_{green}$. $C_{ca}$ was forced as a function of $C_{ab}$ according to Sims and Gamon (2002), with random normal noise of $\mu = 0$ and $\sigma = 4.5$ µg cm$^{-2}$.

Parameters were limited according to the same bounds set for the inversion (see section 2.6); except $C_{ab}$, $C_{ca}$ and senescent pigments content ($C_s$) whose maximum values were 50 µg cm$^{-2}$, 20 µg cm$^{-2}$ and 4 arbitrary units (a.u.), respectively. Different test showed that the emulator must be trained pigment contents bounded with values close to those of the vegetation represented when $f_{green}$ is close to 1 or 0. Averaged parameter values were simulated mixing the LUT pigments according the simulated $f_{green}$, assuming that in green leaves $C_s = 0$ a.u., and that in senescent leaves $C_{ab} = C_{ca} = 0$ µg cm$^{-2}$.

No assumptions were done about $C_w$ and $C_{dm}$ and the LUT values were used as the averaged value from both leaf types. The neural network was then trained to predict $f_{green}$ from the average values using the 60 % of the LUT for training and the remaining 40 % for validation. Training $R^2$, root mean squared error ($RMSE$) and mean error ($ME$) were 0.817, 0.124 and -.001, respectively. Analogously, validation statistics were 0.778, 1.75 and 0.106.

senSCOPE uses the brightness-shape-moisture (BSM) to represent soil optical properties. In this work we used the

parameters determining soil bright estimated in Pacheco-Labrador et al, (2019); and we used superficial (5 cm) soil moisture content ($SM_p$ , vol vol$^{-1}$) registered in the EC towers to force the effect of moisture on soil reflectance factors.

Other model parameters were also determined as a function of forcing variables. Roughness length for momentum of the canopy ($z_0$, m) and canopy displacement height ($d$, m) were calculated as fractions of canopy height ($h_c$, m) using the factors 0.15 (Plate and Quraishi, 1965) and 0.66 (Brutsaert, 1982), respectively. Also, leaf drag coefficient ($C_{d,l}$) was calculated

according to Campbell (1977) as a function of wind speed ($u$, m) and $d$ (Eq. 4)





$$C_{\mathrm{d,l}} = \left[ 388 \cdot u \cdot \sqrt{d/u} \right]^{-1}, \tag{4}$$

Soil boundary layer resistance ($r_{\mathrm{bs}}$, s m$^{-1}$) was determined as a function of the friction velocity ($u^{*}$, m s$^{-1}$) internally calculated by the senSCOPE, according with Thom (1972) (Eq. 5)

$$r_{\mathrm{bs}} = 6.2 \cdot u^{*-\frac{2}{3}}, \tag{5}$$

Soil resistance to evaporation from the pore space ($r_{\mathrm{ss}}$) was prescribed as a function of $SM_{\mathrm{p}}$ using the model fitted in Pacheco-Labrador et al, (2019) in the same study site from lysimeters data (Perez-Priego et al., 2017); however, during the second step of the inversion (see Sect. 2.6), this variable was also estimated.

**2.6 Model inversion and evaluation**

Fig. 1 summarizes the approach followed for the inversion of senSCOPE. As in Pacheco-Labrador et al, (2019) the inversion
is divided in two steps; for both of them we used the numerical optimization algorithm in the Matlab$^{\mathrm{TM}}$ function LSQNONLIN.

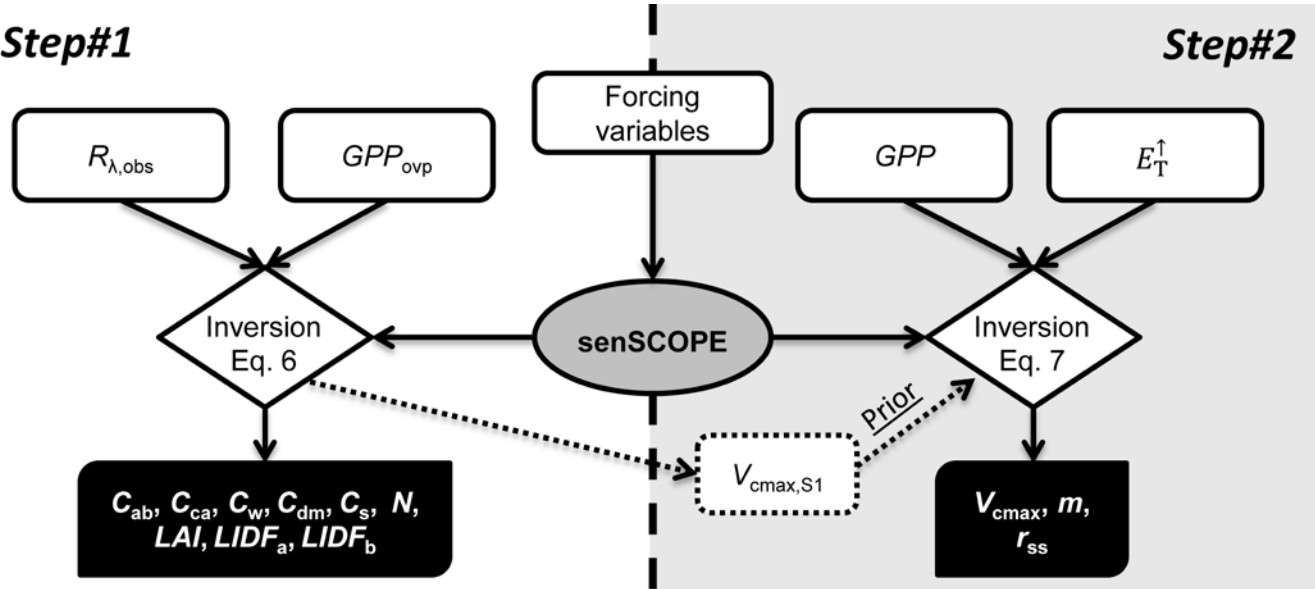

**Figure 1.** Schematic diagram of senSCOPE inversion in two steps.

In Step#1 we combined the *GPP* value corresponding to the time of each overpass, close to solar noon, with a $R_{\lambda}$
characteristic of the vegetation in the EC climatology footprint ($R_{\lambda,\mathrm{obs}}$). This $R_{\lambda,\mathrm{obs}}$ resulted of the convolution of the imagery with the instantaneous climatology footprints calculated $\pm 3$ days around the day of each overpass. From the half-hourly convolutions, the average $R_{\lambda}$ and the corresponding standard deviation ($\sigma_R$) were calculated. Due to the noisy nature of EC *GPP* and since Step#1 relies on a single *GPP* value, this variable was smoothed by averaging all the *GPP* observations 1.5 h





around the overpass ($GPP_{ovp}$). In Step#1, the biophysical parameters of the model (Table 2) and a first guess of $V_{cmax}$

($V_{cmax,S1}$) were estimated minimizing Eq. 6.

$$\chi^2 = \left(\frac{GPP_{ovp,obs} - GPP_{ovp,pred}}{\sigma_{GPP}}\right)^2 + \sum_{\lambda=400}^{930} \left(\frac{R_{\lambda,obs} - R_{\lambda,pred}}{\sigma_{R_\lambda}}\right)^2, \tag{6}$$

where subscripts "obs" and "pred" stand for observed and predicted variables, $\lambda$ is the wavelength in nm and $\sigma_{GPP}$ is the
uncertainty of $GPP$. All the forcing variables of senSCOPE (meteorological conditions, $SM_p$, etc.) were linearly interpolated
to the time of the overpass.

**Table 2.** Parameters estimated inverting sensSCOPE model.

| Parameter | Symbol | Units | Step | Inversion bounds |
|---|---|---|---|---|
| Leaf chlorophyll content | $C_{ab}$ | µg cm$^{-2}$ | #1 | [0, 100] |
| Leaf carotenoids content | $C_{ca}$ | µg cm$^{-2}$ | #1 | [0, 40] |
| Senescent material | $C_s$ | - | #1 | [0, 7.5] |
| Leaf water content | $C_{w,}$ | g cm$^{-2}$ | #1 | [6.3·10$^{-5}$, 0.06] |
| Leaf dry matter content | $C_{dm}$ | g cm$^{-2}$ | #1 | [0.0019, 0.03] |
| Leaf structural parameter | $N$ | layers | #1 | [1, 3.6] |
| Leaf area index | $LAI$ | m$^2$ m$^{-2}$ | #1 | [0, 8] |
| Leaf inclination distribution function | $LIDF_a$ | - | #1 | [-1, 1]; $|LIDF_a +$ |
| Bimodality of the leaf inclination | $LIDF_b$ | - | #1 | $LIDF_b| \leq 1$ |
| Maximum carboxylation capacity | $V_{cmax}$ | µmol m$^{-2}$ s$^{-1}$ | #1 & #2 | [0, 200] |
| Ball-Berry sensitivity parameter | $m$ | - | #2 | [0, 50] |
| Soil resistance to evaporation from the pore space | $r_{ss}$ | s m$^{-1}$ | #2 | [0, 50000] |

Then, parameter posterior uncertainties were calculated as proposed in Omlin and Reichter (1999); where the Jacobian
matrix ($J$) provided correlated posterior distributions of the parameters that were truncated using the truncated Normal and
Student's t-distribution toolbox (Botev, 2017;Botev and Ecuyer, 2015). The standard deviation of 200 realizations of $V_{cmax,S1}$

was used as the posterior uncertainty of this variable ($\sigma_{V_{cmax,S1}}$) with a minimum threshold of 5 µmol m$^{-2}$ s$^{-1}$.

In Step #2, diel cycles of $GPP$ and up-welling longwave irradiance ($E_T^\uparrow$) ±1 day around the overpass were used to estimate
the functional parameters $V_{cmax}$ and $m$; only daytime (down-welling short wave irradiance ($R_{in}$) > 60 W m$^2$ and sun zenith
angle ($\theta_s$) < 60°) data were used. Since Step#2 relies on a large number of observations, half-hourly $GPP$ and $E_T^\uparrow$ were not
further smoothed. Also, the first guess of $V_{cmax,S1}$ and its posterior uncertainty (with a minimum value) were used to

regularize the solution. Unlike in Pacheco-Labrador et al, (2019), $r_{ss}$ was not prescribed in Step#2, but also estimated in this
step assuming a constant value for the three days where predictions and observations were compared. Attempts to prescribe





this parameter suggested that the estimates might not be representative of the heterogeneity of soil conditions in the whole area. Step#2 cost function is presented in Eq. 7:

$$\chi^2 = \sum_{t=0}^{n}\left(\frac{GPP_{\text{obs},t}-GPP_{\text{pred},t}}{\sigma_{GPP_t}}\right)^2 + \sum_{t=0}^{n}\left(\frac{E_{\text{T,obs},t}^{\uparrow}-E_{\text{T,pred},t}^{\uparrow}}{\sigma_{E_{\text{T,obs},t}^{\uparrow}}}\right)^2 + \frac{V_{\text{cmax,S1}}-V_{\text{cmax,S2}}}{\gamma\cdot\max\left(\sigma_{V_{\text{cmax,S1}}},5\right)},\tag{7}$$

where $t$ represents each timestamp of the time series of GPP and $E_{\text{T}}^{\uparrow}$, $\sigma_{E_{\text{T,obs}}^{\uparrow}}$ is the uncertainty of $E_{\text{T}}^{\uparrow}$ and and $\gamma$ is a regularization factor equal 100 and $V_{\text{cmax,S2}}$ is the estimated $V_{\text{cmax}}$ in Step #2.

After Step#2, the $J$ is numerically computed for all the estimated parameters in Step#1 and #2, and used to estimate their corresponding posterior uncertainties as formerly described. Uncertainties were propagated to the fluxes and $R_\lambda$ predicted by senSCOPE running 200 realizations of the model.

**3. Results**

Fig. 2 shows an example of the averaged footprint climatology computed for each of the towers one week around the airborne campaign in July 2015. Fig2.a and b show respectively the Normalized Difference Vegetation Index (*NDVI*) corresponding to the CASI and the synthetic EnMAP images together with the normalized probability distribution function (*PDF*) of the footprint climatologies. As can be seen, the lower spatial resolution of the EnMAP image does not allow
discriminating trees from the grassland, which are clearly differentiated in the CASI image; footprints align approximately in the main (annual) wind direction, and the areas that contribute the most to EC sensors are located at the west-southwest of the towers.



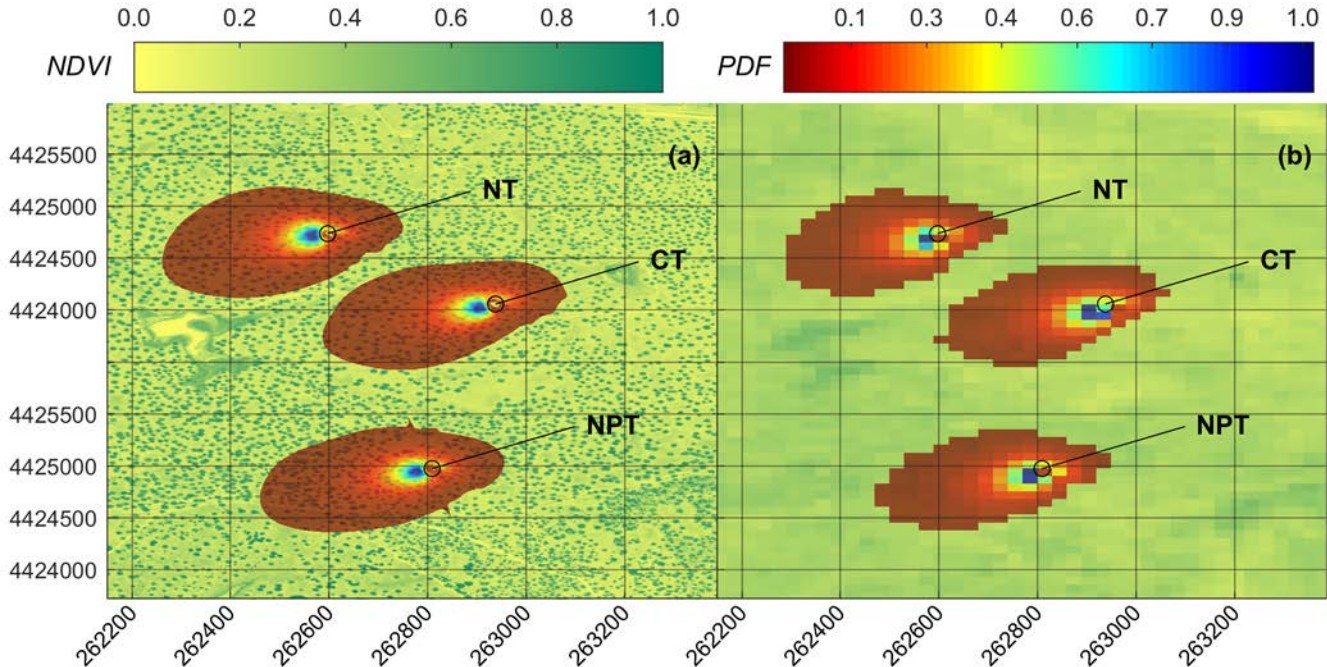

**Figure 2.** Normalized Difference Vegetation Index calculated from CASI (a) and the synthetic EnMAP (b) imagery in July 2015. The
Probability Distribution Function normalized between 0 and 1 of the footprint climatology 1 week around the campaign calculated for each
EC tower, control (CT), fetilized with nitrogen (NT) and fertilized with nitrogen and phosphorous (NPT), are presented at the
corresponding spatial resolutions. Footprint *PDF* < 0.01 have been filtered for clarity.

Fig. 3 summarizes the fit of $R_\lambda$ during inversion. Fig. 3a,b present the observed and predicted *NDVI* for CASI and EnMAP,

respectively. Observed NDVI is computed from the $R_\lambda$ in each tower used to constrain the inversion. Analogously, Fig. 3c,d

present the *ME* of the fit of $R_\lambda$ ($ME_R$) in the visible and the NIR regions for both sensors. Observed *NDVI* shows the

phenological state of vegetation in each campaign, and reveals differences induced by fertilization since Apr-2015, being the

*NDVI* of fertilized towers higher than CT. Campaigns in Oct-2012 and Jul-2015 coincided with the dry season, and *NDVI*

values are low, whereas the campaign in May-2017 took place during the transition from the green peak to senescence.

*NDVI* differences induced by fertilization disappear during the dry season.

$R_\lambda$ shows a good fit after the inversion. $ME_R$ were similar for both sensors, but in general $R_\lambda$ was more accurately fit for

EnMAP. The largest errors were found during the dry season, when $R_\lambda$ was underestimated in the visible region and

overestimated in the NIR. CASI *RMSE* in the visible and NIR regions were 0.0039 and 0.0093, respectively, whereas

EnMAP *RMSE* in the same regions were 0.0028 and 0.0080, respectively.





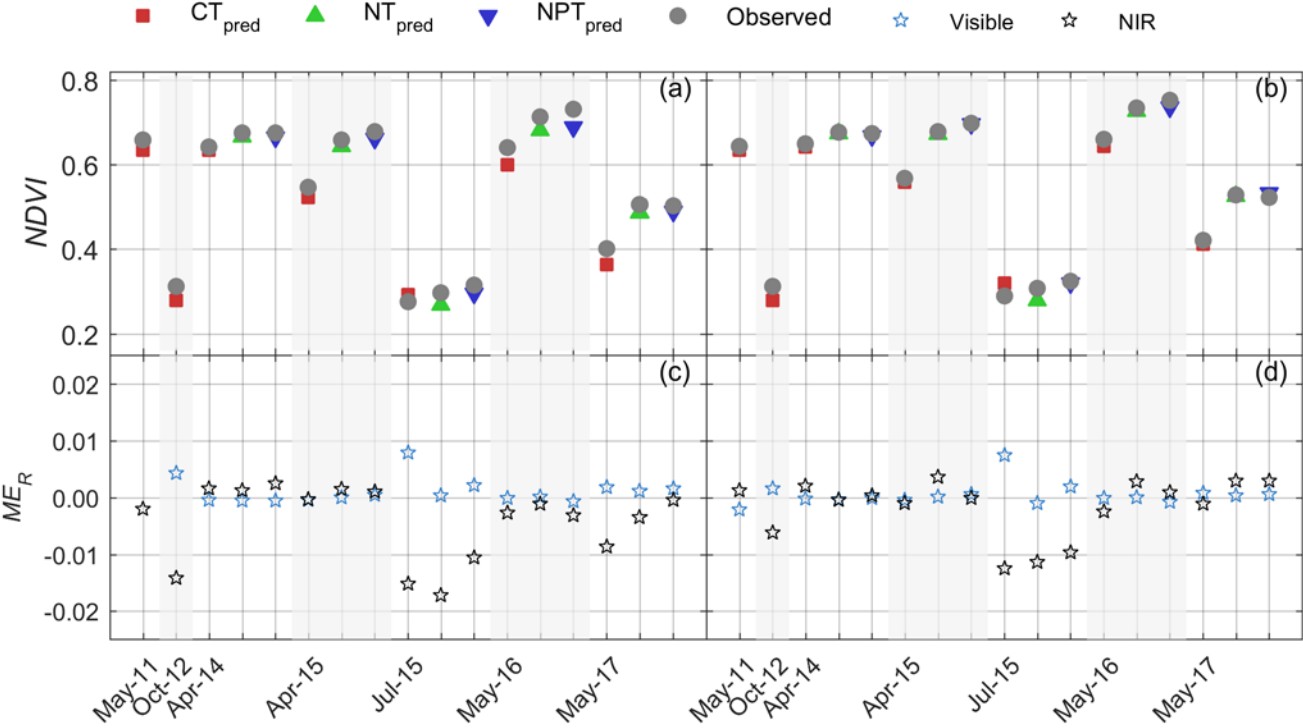

**Figure 3.** Observed and predicted normalized difference vegetation index for CASI (a) and EnMAP (b) for the different towers and campaigns. Mean error of the fit of the reflectance factors in the visible and the near infrared spectral regions for CASI (c) and EnMAP (d) for the different towers and campaigns. Shaded areas separate different campaigns.

Fig. 4 compares the observed biophysical parameters $LAI$, $f_{green}$ and $C_{dm}$ with their corresponding estimates inverting senSCOPE against CASI (Fig. 4a, b, c) and EnMAP imagery (Fig. 4d, e, f) using Total Least Squares to account for similar error on the y and x axis (Golub and Loan, 1980). Green and orange colors indicate spring (NDVI ≥ 0.40) and dry season ($NDVI$ < 0.40), respectively. $LAI$ (Fig. 4a,d) was slightly overestimated, especially for low values; however mid values were underestimated with CASI (Fig. 4a). $f_{green}$ (Fig. 4b,e) was underestimated for both sensors in the dry season, but more closely predicted in spring. $C_{dm}$ (Fig. 4c,f) was overestimated, and in some cases it hit the inversion upper bound; however it was significantly correlated with observed values.





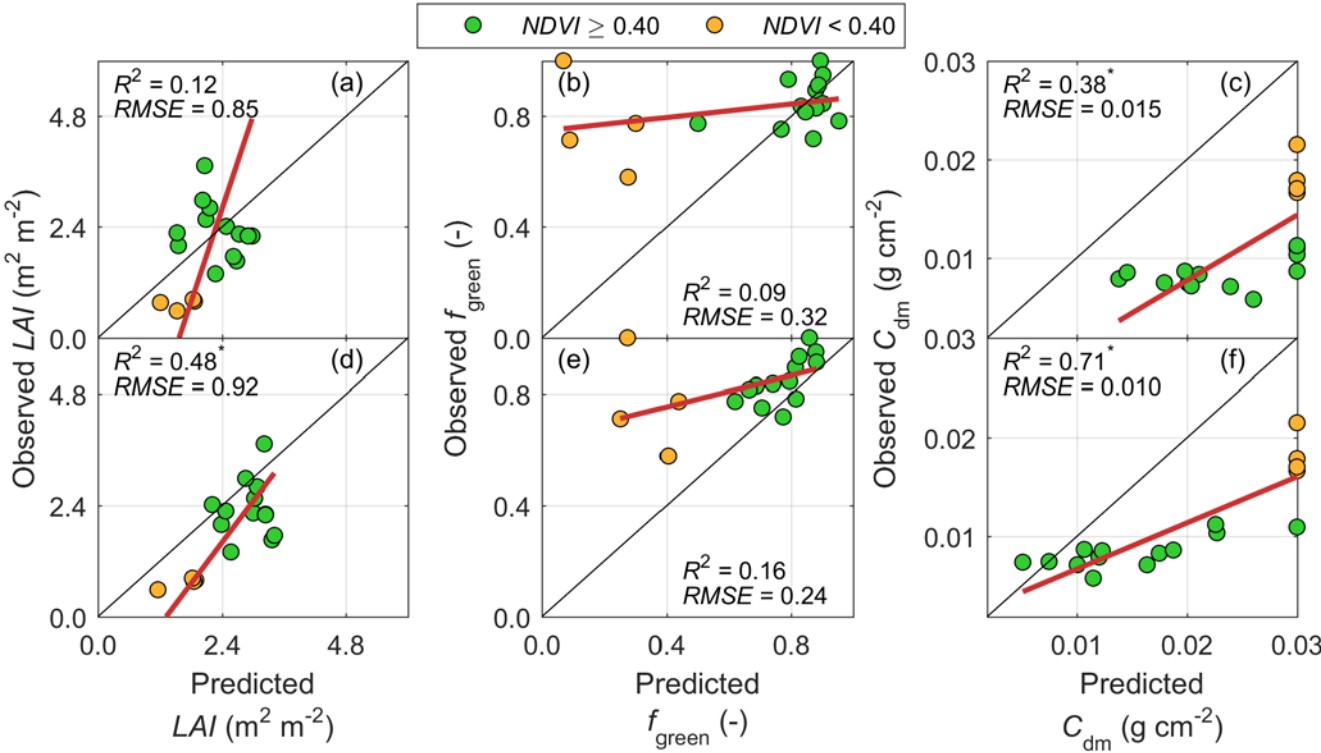

**Figure 4.** Observed vs predicted biophysical parameters inverting senSCOPE against CASI (upper row) and synthetic EnMAP ty(lower row): leaf area index (a,e), fraction of green leaf area (b,f), leaf dry matter content (c,g). Red lines are the relationship estimated with Total Least Squares (Golub and Loan, 1980), black lines show the 1:1 relationship. Subplot axis are limited to the inversion bounds. $^*$ stands for $p < 0.05$ and $^•$ for $0.05 \leq p < 0.10$.

Fig. 5 summarizes pattern-oriented evaluation of senSCOPE after inversion against CASI (Fig. 5 a-d) and EnMAP (Fig. 5 e-h). Fig. 5a,b evaluate predicted and observed chlorophyll content per unit ground area ($C_{ab,ground}$) against N content in green vegetation per unit ground area ($N_{ground,green}$). Estimated parameters pretty well reproduced the $C_{ab}$-N relationship followed by field observations and literature. Fig. 5d,c shows $V_{cmax}$ (per unit total leaf area, $V_{cmax,total}$) against $N_{mass}$; in all the cases, $V_{cmax,total}$ estimates positively correlated with $N_{mass}$, but values are lower than the ones estimated from field data and

literature. The relationships are not significant (CASI) or weak (EnMAP); this is in part due to the low  range of variation of $N_{mass}$ (and therefore of $V_{cmax,total}$) (e.g., see Pacheco-Labrador et (2019)), so that the uncertainty of the retrieval prevented significant relationships. Field $V_{cmax,total}$ was estimated using $N_{mass,green}$ and relationships from the literature. As expected, both estimates of $C_{ab,ground}$ and $V_{cmax,total}$ presented the lowest values during the dry season. Similarly, spring $m$ (Fig. 5e,f) estimates and predicted underlying water use efficiency ($uWUE$) (Fig. 5g,h) were evaluated against measurements of $\Delta^{13}C$. Significant

positive relationships were found between $m$ and $\Delta^{13}C$ for CASI, whereas significant and negative relationships were found between $uWUE$ simulated with senSCOPE after the inversion and $\Delta^{13}C$ for both sensors.



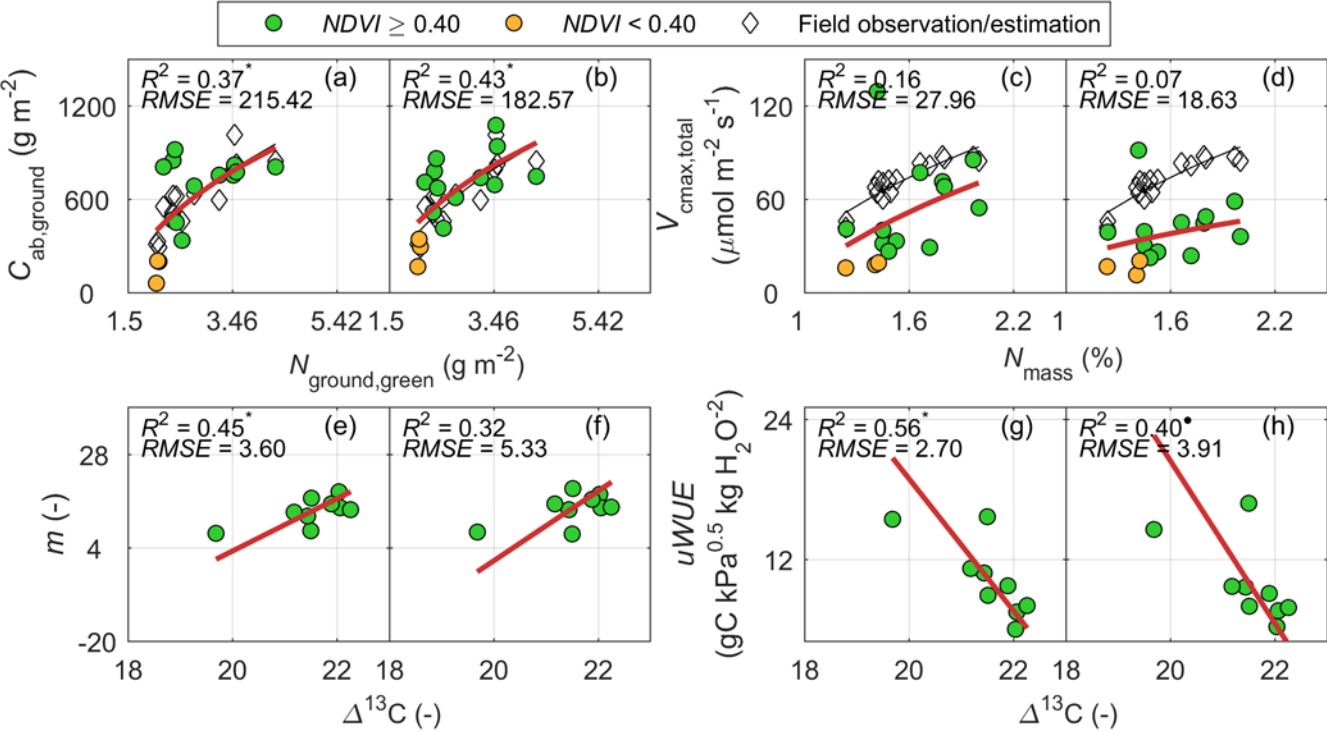

**Figure 5.** Pattern-oriented model evaluation of: chlorophyll content per unit ground area vs. nitrogen content per unit ground area (a,b), maximum carboxylation rate per unit total leaf area vs. nitrogen concentration (c,d), Ball-Berry stomatal sensitivity vs. $^{13}$C isotope discrimination (e,f), and predicted underlying water use efficiency vs. $^{13}$C isotope discrimination (g,h). CASI estimates/predictions are presented always on the left (a,c,e,g) and EnMAP estimates on the right (b,d,f,h) of each pair of subplots. * stands for $p < 0.05$ and • for $0.05 \leq p < 0.10$

Fig. 6 compares observed and predicted fluxes after inversion for CASI (Fig 6 a-e) and EnMAP (Fig 6 f-j). *GPP* (Fig. 6a,f) and $E_T^\uparrow$ (Fig. 6b,g) are constraints of the inversion; the first was accurately fit, whereas the second was overestimated (shifted ~30 W m$^{-2}$). $\lambda E$ (Fig. 6c,h) was overestimated in the dry season, whereas *H* (Fig. 6d,i) was biased (slopes ~0.45). Consequently, *EF* resulted overestimated in the dry period, and underestimated for low values in spring. However, it was well predicted for values > 0.8, when evapotranspiration dominates. Differences between predicted and observed $\lambda E$ and *H* fluxes were strongly related with the energy balance closure error ($\varepsilon_{EB}$) of the EC data (Fig. A1); showing $R^2$ ~ 0.84 and *RMSE* ~ 32.5 W m$^{-2}$. Median $\varepsilon_{EB}$ was ~111 W m$^{-2}$ ($\sigma$ ~ 81 W m$^{-2}$), which in relative terms represents the ~21.5 % ($\sigma$ ~ 14.2 %) of the observed net radiation. However, during the dry season, predicted $\lambda E$ and *H* error are slightly biased respect to $\varepsilon_{EB}$.


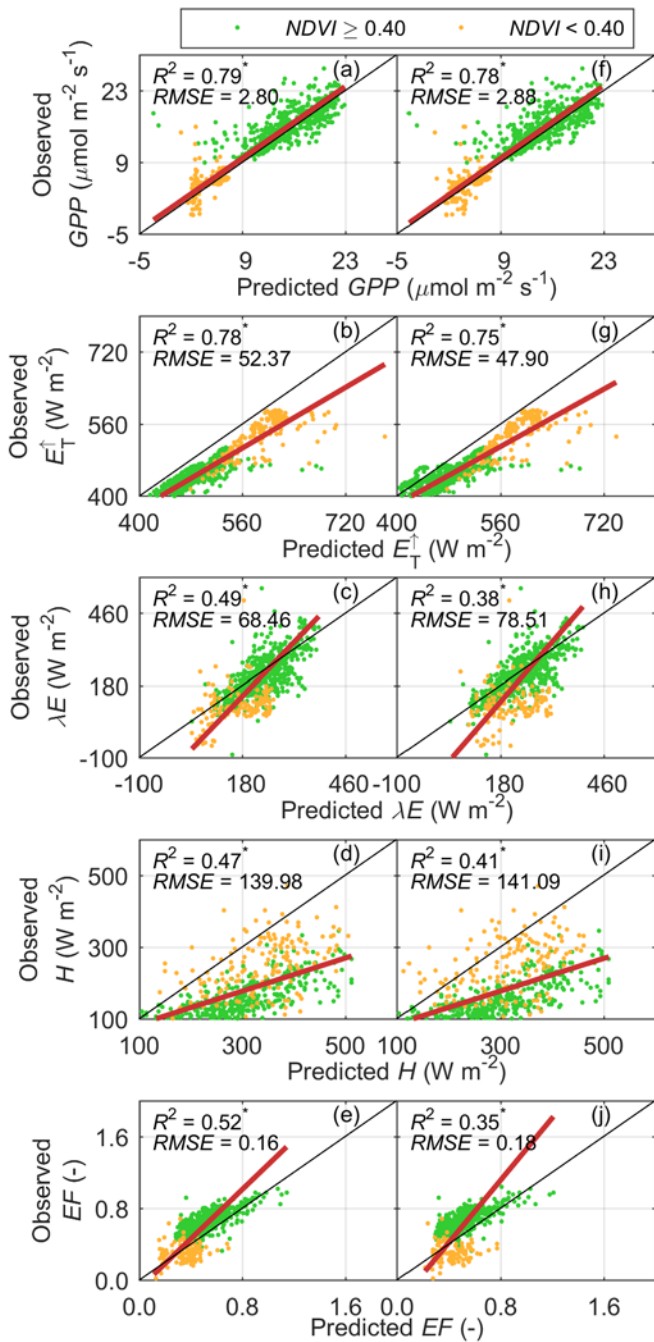

**Figure 6.** Predicted vs observed fluxes for CASI (left column) and EnMAP (right column): Gross primary production (a,f), up-welling thermal irradiance (b,g), latent heat flux (c,h), sensible heat flux (d,i) and evaporative fraction (e,j). Red lines are the relationship estimated with Total Least Squares (Golub and Loan, 1980), black lines show the 1:1 relationship. $^*$ stands for $p < 0.05$ and $^{\bullet}$ for $0.05 \leq p < 0.10$






Fig. 7 presents the estimates of $r_{ss}$ compared vs $SM_p$. and the exponential decay model was fit. CASI estimates (Fig. 7a) were very low in spring for $SM_p > 12$ %, and then increase below this threshold. Results for EnMAP are similar (Fig. 7b). In both sensors, some of the estimates in $SM_p \sim 15$ % present high values, separating from the expected decreasing relationship.

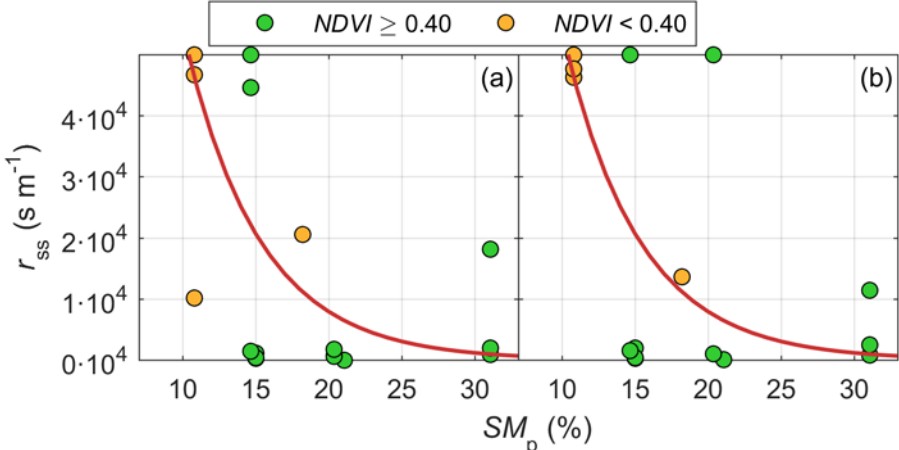

**Fig. 7.** Evaluation of the CASI (a) and EnMAP (b) estimates of soil resistance to evaporation from the pore space against soil water content. Red lines represent a fit exponentially decreasing model.

## 4. Discussion

The new generation of hyperspectral imaging satellite missions is expected to improve the spatiotemporal characterization of vegetation function and functional traits (Schimel et al., 2019;Butler et al., 2017). To achieve this target, advances in
uncertainty quantification of multiple data streams, algorithm development, as well as modelling and data integration are essential (Schimel et al., 2019). In this context, we prove the capability of hyperspectral imagery jointly with data from EC stations and coupled RTM-SVAT models to inform about vegetation functional traits. We show that an inversion of senSCOPE model in a Mediterranean tree-grass ecosystem using multiple constraint ($GPP$, $E_T^\uparrow$ and remote hyperspectral reflectance factors) lead to robust estimates of key biophysical and functional traits of vegetation. The characterization of
vegetation functional traits is necessary to improve our understanding and simulation of terrestrial water and carbon fluxes. In this context, EC networks are a keystone for the development, evaluation and benchmarking of TBM (Bonan et al., 2011;Williams et al., 2009) as well as statistical models (Jung et al., 2011). These models are expected to predict later flux information 'everywhere, all of the time' (Baldocchi, 2014); partly thanks to the use of global information about vegetation biophysical parameters provided by RS. One of the limits of this approach is the lack of knowledge on the spatiotemporal
distribution of key functional traits controlling these fluxes (Rogers, 2014;Rogers et al., 2016;Schaefer et al., 2012); which cannot be directly inferred by RS since they have little influence on the outgoing radiance observed from space. The estimation of functional parameters such as $V_{cmax}$ or $m$ is complicated even from field observations. These parameters must be estimated from leaf level measurements, which are time and labor demanding and not always feasible. Then, even if





feasible, leaf-level estimates need to be up-scaled to the ecosystem level considering the intra and inter-specific variability of
the vegetation around the EC tower (Sprintsin et al., 2012), increasing their uncertainty. For this reason, process-based model
inversion has been used to estimate these parameters from fluxes in EC stations (Zheng et al., 2017;Reichstein et al., 2003).
However, coherent estimation or scaling of these parameters strongly depends on the description of canopy structure
(Sprintsin et al., 2012); so that lack or uncertain knowledge on the biophysical properties of vegetation around the tower can
result in uncertain prediction of fluxes (Wang et al., 2019) and consequently in uncertain estimates of the functional traits.
Alternatively, hyperspectral imagery is an independent source of information on canopy structure, biochemistry (Ustin and
Gamon, 2010;Schaepman et al., 2009) , and partly to function (Serbin et al., 2015); its numerous and fine bands are sensitive
to narrow spectral features and allow solving over-determined systems, which increases the accuracy of the retrievals (Goetz,
2009). The present manuscript demonstrates that the combination of hyperspectral and EC data in models accurately
representing radiative transfer, energy balance and photosynthesis allows a coherent estimation of both biophysical and
functional traits. This is possible because the multiple-constraint imposed by the combination of RS and EC observations is
more robust against model errors than the constraint imposed by RS data only (Pacheco-Labrador et al., 2019b). Therefore,
this combination might be also more suitable than the mere ingestion of existing RS products of biophysical parameters such
as *LAI*, which in some cases include ecosystem-dependent uncertainties (Wang et al., 2019). In a scenario where  both EC
flux stations networks are developing quickly (e.g. FLUXNET, ICOS, OzFLUX, and NEON) and hyperspectral satellite
missions are increasing (e.g. DESIS, PRISMA, HyspIRI, EnMAP, etc.), our work calls for a combination of hyperspectral
and EC data together with models that accurately represent radiative transfer, energy balance and photosynthesis (RTM-
SVAT). The characterization of functional (and biophysical) traits in these networks could be later used to inform, constrain
and evaluate TBM, to inform TMB input uncertainties (Prichard et al., 2019), or to benchmark estimates provided by
different models and methods (e.g., Luo et al., (2019), Croft et al., (2017), Zhou et al., (2014) or Xie et al., (2018)). Also,
when comprehensive enough, they could be predicted at global scale using data-driven approaches (Jung et al., 2011) or RS
(Serbin et al., 2015).

The application of the multiple-constrained inversion of a coupled SVAT-RTM model at ecosystem scale implies facing
sources of error and uncertainty that were not present in previous works carried out at close range over grassland plots
(Pacheco-Labrador et al., 2020;Pacheco-Labrador et al., 2019). These can be summarized as: 1) the spatial mismatch
between optical, thermal and EC footprints, 2) uncertainties in the estimation of *GPP* and EC energy balance closure, 3) the
atmospheric correction of airborne imagery and the addition of uncertainties in the EnMAP End-to-End simulator, and 4)
errors induced by the representation of a Mediterranean tree-grass ecosystem featuring species with very different function
(grasses and trees) when using a 1-D model to represent light interaction and photosynthesis. In some cases, we have applied
measures to correct some of these uncertainties that could be generally used. Despite of these uncertainties, the estimated
parameters are coherent with field data and pattern-oriented model evaluation demonstrated the robustness of the proposed
approach. This can be explained since the combination of multiple constraints carrying different and relevant information on
the parameters to be estimated is able to counterweight the effect of uncertainties both in model and observations (Keenan et





al., 2011;Wutzler and Carvalhais, 2014). An evaluation of different sets of constraints of the SCOPE showed the strong

constrain of *GPP* on functional traits, and its capability to prevent deviations of biophysical parameters due to uncertainties

(Pacheco-Labrador et al., 2019). The robustness of the method proposed is promising for its application in different EC

stations monitoring different ecosystems, which is still to be tested.

The first abovementioned source of uncertainty is the spatial mismatch between optical and an EC footprints, which is

inherent to the combination of RS and EC data. To minimize this mismatch, we used footprint climatology approach (Kljun

et al., 2015); which can be applied when pixel size is relatively small compared with EC footprint size. However, to simplify

modeling and assimilation, we averaged $R_\lambda$ convolved with half-hourly footprint *PDF* to provide a representative spectral

signature of the area generating the fluxes. Pacheco-Labrador et al., (2017a) found that the spatial variability of the EC

footprint induced limited changes in the spectral signal integrated from CASI imagery in this site; and that this variability

had little impact in the modeling of *GPP*. For this reason the spectral signal integrated the week around the image acquisition

was considered representative enough of the fluxes source; whereas the spectral variability induced by half-hourly EC

footprints was used to weight the cost function in inversion. Nonetheless, this approach might not be advisable for sites

where the properties of the source area strongly vary with footprint displacement, and its suitability should be evaluated at

each EC station. A second mismatch exists between the footprints of the $E_T^\uparrow$ hemispherical sensor and the EC / RS footprints

(Marcolla and Cescatti, 2018). Radiation sensors in EC towers are supposed to be representative of the ecosystem monitored,

but this is a challenge in many EC sites (Leuning et al., 2012). This mismatch is especially problematic in ecosystems that

are heterogeneous and / or that feature scattered 3D volumes; in these cases the contribution of sunlit and shaded vegetation

and different plant types to the hemispherical sensors can vary through the day as a function of sensor location and sun

position. These contributions might not be coherent with the radiative regime of the EC footprint in all the cases (Marcolla

and Cescatti, 2018), which induces part of the error in the closure of the energy balance (Leuning et al., 2012). In this work,

we did not correct for this mismatch, which would require detailed 3D information of the tower environment and of the

location of the sensor. Footprint differences might have induced the overestimation of $E_T^\uparrow$ (Fig. 6b,g) and increased $\varepsilon_{EB}$ since

the contributions of colder tree crowns might be enhanced by the closer distance to the sensor, and because cooler shaded

grass patches are not represented by the model.

A second source of uncertainty is related to the EC data. We attempted to reduce the inherent noise to EC *GPP* data by

smoothing time series (Damm et al., 2010) around the RS observation time. This was relevant for the estimation of

biophysical parameters in Step#1, which relies on a single *GPP* value. These parameters are strongly constrained by *GPP* via

*APAR* (Pacheco-Labrador et al., 2019), and unlike $V_{cmax}$, are not refined in Step#2. For this reason, we also aggregated *GPP*

data 1.5 h around each overpass. Results suggest that estimated biophysical parameters are robust or at least consistent

within the dataset. Additionally, it must be considered that the energy balance closure gap in real EC datasets is relatively

high (median $\varepsilon_{EB}$ ~21.5 % of $R_n$ in our case) compared to the gap achieved by models, in this case senSCOPE (median $\varepsilon_{EB}$

~0.12 % of $R_n$). Large observational $\varepsilon_{EB}$ hinders the direct comparison of predicted and observed water and energy fluxes.

For example Fig.5 shows disagreement between these fluxes, resulting $\lambda E$ overestimated but in the dry season and *H* biased





(slope ~ 0.45), especially during spring. However, these differences strongly correlated with $\varepsilon_{EB}$ (Fig. A1) which suggest that this disagreement might be produced by observation uncertainties rather than by biased predictions. The evaluation of *EF* provides a more independent comparison of these fluxes and is closer to the 1:1 line than the original fluxes; but, still

deviations from this reference are observed. So far, different inversions of SCOPE have relied either on remote thermal imagery (Bayat et al., 2018), on EC TIR irradiance (this work) or on $\lambda E$ (Dutta et al., 2019) to constrain functional parameters related with transpiration such as the Ball-Berry sensitivity parameter; but no comparison of these constraints has been yet carried out. Some of the advantages and disadvantages of each variable are clear: TIR radiation measured in EC towers from hemispherical diffusors offer high temporal frequency, but the radiometric footprint does not match the EC

footprint. The use of TIR imagery would allow the use of footprint climatology models minimizing the mismathc, however diel series of TIR imagery feature low spatial resolution. On the other hand $\lambda E$ and $H$ match the *GPP* footprint, but they present large uncertainties related with the lack of closure of the energy balance. The impact of these uncertainties in the inversion of SCOPE and similar models is still unknown.

RS data also include uncertainties related to the instrumentation, the atmospheric correction and directional effects; the last

partly increased by the wide FOV of CASI (~40 °). We tried to minimize directional effects mosaicking different overpasses when available. Space-borne hyperspectral imagery do not often provide multi-angular data which makes unlikely mosaicking or correction of angular effects; but at the same time it also presents cross-track field of views much narrower than our airborne sensor and therefore a lower range of observation angles (Guanter et al., 2015;Ungar et al., 2003;Lee et al., 2015). In the case of this study, directional effects are also expected to be minimized by the convolution of imagery with

footprint climatology; which enhances the contribution of those pixels closer to the EC towers, where the flight tracks were centered and therefore observed with lower zenith angles. Compared to the airborne imagery, different and even larger radiometric uncertainties (e.g., no empirical line correction is applied) and lower spatial and spectral resolutions are expected form space sensors. The applicability of the method had to be tested on a dataset comparable to space-borne hyperspectral data. Since no such imagery was available at the site; we used the EnMAP End-to-End simulator to produce synthetic images

including spatial, instrumental and atmospheric uncertainties representative of this sensor (Segl et al., 2012). These uncertainties were added to the ones already existing in the CASI dataset, producing an even rougher test for the method. Despite of lower resolutions and larger uncertainties of the EnMAP images, results were consistent between both sensors. This suggests that space-borne hyperspectral sensors similar to EnMAP could be successfully used to estimate vegetation functional traits with the approach evaluated in this manuscript.

The fourth source of uncertainty is model error. The Mediterranean tree-grass ecosystem under study presents features not well represented by senSCOPE and other models. Pacheco-Labrador et al, (2019) inverted SCOPE in a Mediterranean grassland of this site and found that the model was not suitable to characterize radiative transfer and physiological processes in such ecosystem due to the presence of large fractions of senescent leaves. This problem is partly corrected by senSCOPE, a modified version of SCOPE that separately represents green and senescent leaves. However, the problem is not completely

solved. In that work senSCOPE improved the retrieval of $C_{ab}$ via inversion, but still overestimated NIR $R_\lambda$ and





underestimated *LAI* suggesting that the optical properties of senescent material were not adequately represented. This conclusion was in agreement with those of of Melendo-Vega et al., (2018) in the same site. Moreover, we must consider that in the present work, senSCOPE (a 1-D model) is used to represent a Mediterranean tree-grass ecosystem featuring 1) a strong 3-D structure, 2) a mixture of species (grasses and trees) with very different function and ecological strategies, and 3)

a grassland layer featuring large fractions of senescent leaves and death standing material, whose optical properties are misrepresented. These conditions portray a harsh test for the inversion method; however, model parameters seem to be robustly estimated, are coherent with ancillary observations and between airborne and synthetic EnMAP imagery. Still, the effect of some of these uncertainties can be analyzed. For example, angular effects can be important in savanna-like ecosystems featuring a strong geometrical scattering component. Melendo-Vega et al, coupled the 1-D RTM PROSAIL

(Jacquemoud et al., 1996; Verhoef, 1984) with the 3-D RTM FLIGHT (North, 1996) to represent tree-grass ecosystems and compared this model with PROSAIL, showing that for a low tree cover, differences in predicted $R_\lambda$ minimized at nadir. Consequently, the reduction of off-nadir view directional effects produced by footprint integration and mosaicking might have contributed to reduce the impact of 3-D structure during inversion. However, the 1-D model used does not reproduce tree crown shading on the grassland; which should affect $R_\lambda$ and *APAR* during the day course. Also, the misrepresentation of

optical properties of the senescent material biases the estimates of some parameters in an attempt to compensate low NIR scattering. At grass plot scale *LAI*, NIR $R_\lambda$ were underestimated and $f_{green}$ was overestimated (Pacheco-Labrador et al., 2020); whereas at ecosystem scale *LAI* and NIR $R_\lambda$ were not so strongly biased, and $f_{green}$ was underestimated in the dry period. In both cases $C_s$ and $C_{dm}$ were large, especially in the dry season, since both reduce NIR $R_\lambda$. At ecosystem scale tree crowns increase NIR $R_\lambda$ in all the seasons (Pacheco-Labrador et al., 2017a) which might have prevented the underestimation of *LAI*.

$C_{ab}$ and $V_{cmax}$ estimates and their relationship with nitrogen are consistent with field data. Retrieved $V_{cmax}$ is lower than independent estimates based on nitrogen concentration and literature not specific of semi-arid Mediterranean ecosystems. These estimates might present values too high for the study site. For example, the $N_{mass}$-$V_{cmax}$ curve reported in Feng and Dietze (2013) and used to generate the evaluation dataset predict higher $V_{cmax}$ than values estimated in the grass plots in the same ecosystem (Pacheco-Labrador et al., 2019). An alternative hypothesis is related to the fact that tree crowns reduce

down-welling irradiance impinging the grassland during the day, which is not adequately represented by the 1-D RTM. Overestimation of *APAR* (and therefore *GPP*) might have been compensated decreasing the photosynthetic leaf surface ($f_{green}$), or $V_{cmax}$ to fit observed *GPP*. Pattern-oriented model evaluation of *m* and *uWUE* against $\Delta^{13}$C provided expected relationships when comparing spring *m* and *uWUE* estimates with $\Delta^{13}$C, a seasonally-integrated indicator of plant water use efficiency. In the dry season *m* estimates showed high values for both sensors (CASI and synthetic EnMAP), suggesting low

*uWUE*. Wolf *et al.,* (2006) also estimated large *m* values in a grassland during the dry season, and related them with decoupled simulation of transpiration with other processes (which is not the case of the model used here), and to assumptions about leaf thermal properties. Contrarily, Bayat et al., (2018) did not found such large values during summer in a different grassland. However Bayat et al., (2018) inverted SCOPE model where all the leaf area can transpire; whereas in senSCOPE transpiration is limited to the green fraction. Looking at the fluxes can be noticed that in the dry season, predicted *H* is in





agreement with observations, whereas $\lambda E$ is overestimated (Fig. 6); these discrepancies are slightly biased respect to EC $\varepsilon_{EB}$ (Fig. A1). We hypothesize that $m$ estimates in the dry season might not be realistic: Overestimation of $E_T^{\uparrow}$ might have demanded large transpiration during summer time; when low $GPP$ and $f_{green}$ (also underestimated) might have reduced the impact of $m$ on the cost function, allowing large values with little effect on $E_T^{\uparrow}$. It must be also considered that the Ball-Berry model implemented in SCOPE and derived models relies on relative humidity; this approach does not well represent

response leaf responses of both conductance and intracellular $CO_2$ concentration to stressing humidity conditions, and alternative models have been proposed (Leuning, 1990, 1995). These models might also prevent the overestimation of $m$ during stress conditions. Further work is needed to understand the challenges in the seasonal retrieval and validation of $m$ estimates form RS.

The previous analyses showed the need and the added value of evaluating functional trait estimates in the context of the

development of new RS products of vegetation function and functional traits. Since direct observations of these traits would not be as usual as other biophysical variables that are easier to measure in the field; new methods such as pattern oriented evaluation should be developed in parallel. Thorough evaluation contributes to understand the effect of different sources of uncertainty, model structure and to identify possibilities of improvement; and therefore it will be fundamental to support the development of the next generation of RS products. To the three challenges to overcome proposed by Schimel et al., (2019)

we should also add the validation and evaluation of the new functional trait estimates as an additional challenge to overcome.

## 5. Conclusions

This work proves the potential of the new generation of hyperspectral space missions in combination with EC networks to characterize the spatiotemporal distribution of vegetation functional parameters relevant for the prediction of water and carbon fluxes. The method proposed has proven robust to different sources of uncertainty; some of which were minimized

with different approaches (e.g., footprint climatology convolution, or the use of senSCOPE model). Repeated acquisition of images over the same EC stations can provide not only seasonal variability of key functional and biophysical parameters, but also, it can characterize relationships between biophysical and functional parameters and other variables, such as those of $C_{ab}$ or $V_{cmax}$ with nitrogen, $m$ or $uWUE$ with $\Delta^{13}C$, or $r_{ss}$ with $SM_p$. Spatiotemporal characterization of functional traits as well as these relationships could useful for validation, modeling or inversion of TBM and other models. Results also show that

biases in the prediction of $\lambda E$ and $H$ strongly correlate with observational energy balance closure error. Pros and cons of the use of energy fluxes or TIR radiance from proximal or remote sensors to constrain leaf water exchange are discussed; further research is needed to determine which of these constraints is the most suitable for the estimation of functional parameters in EC sites. Finally, the evaluation of the functional trait estimates is complex and direct observations might not be often available. Pattern-oriented model evaluation has provided relevant information to understand the effect of observational and

model uncertainties; evaluation methodologies should developed in parallel with the new remote sensing products of vegetation function.



## Appendix A. Energy balance closure and heat and water flux prediction errors.

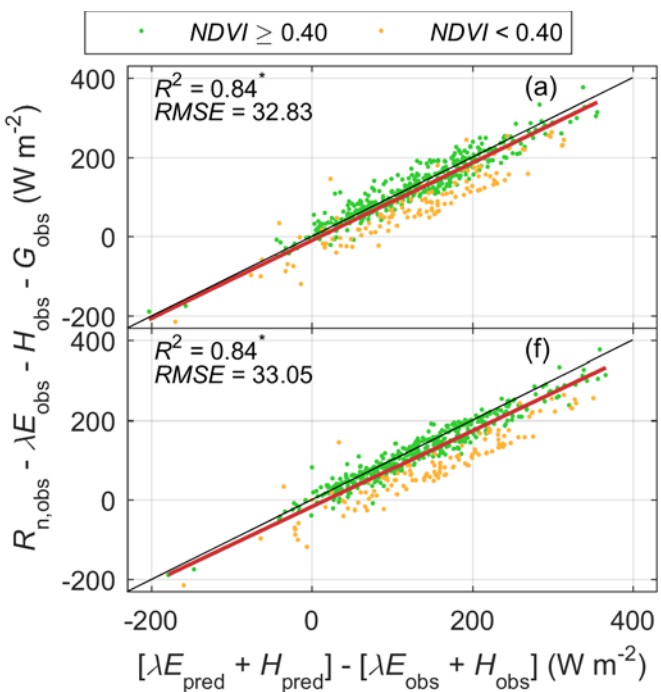

**Figure A1.** Prediction error of latent and sensible heat fluxes versus observed energy balance closure errors for CASI (a) and synthetic EnMAP (b).

## Code availability

senSCOPE code and further developments, as well as the code for the multiple constraint inversion of the model are publicly available at https://github.com/JavierPachecoLabrador/senSCOPE.

## Data availability

All data included in this study are available upon request by contact with the corresponding author.

## Author contributions

JPL, MM and TSEM developed data assimilation and evaluation approaches. TSEM, PM, RGC, AC, GM, JPL, OPP, TH, HM, KH, OK, KS, MR and MM provided measurements of fluxes, vegetation parameters and hyperspectral imagery. JPL, MM, TSEM, PM, RGC, OPP, DM, VB, AC, CvdT and MR contributed to discussion and interpretation of the results. JPL, MM, PM, OPP, RGC, GM, TSEM and DM wrote and corrected the manuscript.



**Competing interests**

The authors declare that there are no competing interests

**Acknowledgements**

JPL, MM and MR acknowledge the EnMAP project MoReDEHESHyReS "Modelling Responses of Dehesas with
Hyperspectral Remote Sensing" (Contract No. 50EE1621, German Aerospace Center (DLR) and the German Federal
Ministry of Economic Affairs and Energy). DM, VB and MM received funding from the European Union's Horizon 2020
research and innovation programme via the TRUSTEE project under the Marie Skłodowska-Curie grant agreement No.
721995. Authors acknowledge the Alexander von Humboldt Foundation for supporting this research with the Max-Planck
Prize to Markus Reichstein; the project SynerTGE "Landsat-8+Sent inel-2: exploring sensor synergies for monitoring and
modelling key vegetation biophysical variables in tree-grass ecosystems" (CGL2015-69095-R, MINECO/FEDER,UE); the
project FLUχPEC "Monitoring changes in water and carbon fluxes from remote and proximal sensing in Mediterranean
'dehesa' ecosystem" (CGL2012-34383, Spanish Ministry of Economy and Competitiveness); and the project BIOSPEC
"Linking spectral information at different scales with biophysical parameters of Mediterranean vegetation in the context of
global change" (CGL2008-02301, Spanish Ministry of Economy and Innovation). Authors are very thankful to the MPI-
BGC Freiland Group and especially as well as Ramón López-Jiménez (CEAM) for technical assistance. We are grateful to
all the colleagues from MPI-BGC, University of Extremadura, University of Milano-Bicocca, SpecLab-CSIC, INIA and
CEAM which have collaborated in any of the field and laboratory works. We acknowledge the Majadas de Tiétar city
council for its support.

**References[i]**

Alemohammad, S. H., Fang, B., Konings, A. G., Aires, F., Green, J. K., Kolassa, J., Miralles, D., Prigent, C., and Gentine,
P.: Water, Energy, and Carbon with Artificial Neural Networks (WECANN): a statistically based estimate of global surface
turbulent fluxes and gross primary productivity using solar-induced fluorescence, Biogeosciences, 14, 4101-4124,
10.5194/bg-14-4101-2017, 2017.
Ali, A. A., Xu, C., Rogers, A., Fisher, R. A., Wullschleger, S. D., Massoud, E. C., Vrugt, J. A., Muss, J. D., McDowell, N.
G., Fisher, J. B., Reich, P. B., and Wilson, C. J.: A global scale mechanistic model of photosynthetic capacity (LUNA V1.0),
Geosci. Model Dev., 9, 587-606, 10.5194/gmd-9-587-2016, 2016.
Alton, P. B.: How useful are plant functional types in global simulations of the carbon, water, and energy cycles?, Journal of
Geophysical Research: Biogeosciences, 116, 10.1029/2010JG001430, 2011.



Alton, P. B.: Retrieval of seasonal Rubisco-limited photosynthetic capacity at global FLUXNET sites from hyperspectral
satellite remote sensing: Impact on carbon modelling, Agricultural and Forest Meteorology, 232, 74-88,
https://doi.org/10.1016/j.agrformet.2016.08.001, 2017.

Baldocchi, D.: Breathing of the terrestrial biosphere: lessons learned from a global network of carbon dioxide flux
measurement systems, Australian Journal of Botany, 56, 1-26, https://doi.org/10.1071/BT07151, 2008.

Baldocchi, D.: Measuring fluxes of trace gases and energy between ecosystems and the atmosphere – the state and future of
the eddy covariance method, Global Change Biology, 20, 3600-3609, 10.1111/gcb.12649, 2014.

Bayat, B., van der Tol, C., and Verhoef, W.: Integrating satellite optical and thermal infrared observations for improving
daily ecosystem functioning estimations during a drought episode, Remote Sensing of Environment, 209, 375-394,
https://doi.org/10.1016/j.rse.2018.02.027, 2018.

Bonan, G. B., Lawrence, P. J., Oleson, K. W., Levis, S., Jung, M., Reichstein, M., Lawrence, D. M., and Swenson, S. C.:
Improving canopy processes in the Community Land Model version 4 (CLM4) using global flux fields empirically inferred
from FLUXNET data, Journal of Geophysical Research: Biogeosciences, 116, 10.1029/2010JG001593, 2011.

Botev, Z. I., and Ecuyer, P. L.: Efficient probability estimation and simulation of the truncated multivariate student-t
distribution, 2015 Winter Simulation Conference (WSC), 2015, 380-391,

Botev, Z. I.: The normal law under linear restrictions: simulation and estimation via minimax tilting, Journal of the Royal
Statistical Society: Series B (Statistical Methodology), 79, 125-148, 10.1111/rssb.12162, 2017.

Brand, W. A.: Atomic weights: not so constant after all, Analytical and Bioanalytical Chemistry, 405, 2755-2761,
10.1007/s00216-012-6608-0, 2013.

Brooks, P. D., Geilmann, H., Werner, R. A., and Brand, W. A.: Improved precision of coupled δ13C and δ15N
measurements from single samples using an elemental analyzer/isotope ratio mass spectrometer combination with a post-
column six-port valve and selective CO2 trapping; improved halide robustness of the combustion reactor using CeO2, Rapid
Communications in Mass Spectrometry, 17, 1924-1926, 10.1002/rcm.1134, 2003.

Brutsaert, W.: The Surface Roughness Parameterization, in: Evaporation into the Atmosphere: Theory, History and
Applications, edited by: Brutsaert, W., Springer Netherlands, Dordrecht, 113-127, 1982.

Bunce, J. A.: Acclimation of photosynthesis to temperature in eight cool and warm climate herbaceous C3 species:
Temperature dependence of parameters of a biochemical photosynthesis model, Photosynthesis Research, 63, 59-67,
10.1023/A:1006325724086, 2000.

Butler, E. E., Datta, A., Flores-Moreno, H., Chen, M., Wythers, K. R., Fazayeli, F., Banerjee, A., Atkin, O. K., Kattge, J.,
Amiaud, B., Blonder, B., Boenisch, G., Bond-Lamberty, B., Brown, K. A., Byun, C., Campetella, G., Cerabolini, B. E. L.,
Cornelissen, J. H. C., Craine, J. M., Craven, D., de Vries, F. T., Díaz, S., Domingues, T. F., Forey, E., González-Melo, A.,
Gross, N., Han, W., Hattingh, W. N., Hickler, T., Jansen, S., Kramer, K., Kraft, N. J. B., Kurokawa, H., Laughlin, D. C.,
Meir, P., Minden, V., Niinemets, Ü., Onoda, Y., Peñuelas, J., Read, Q., Sack, L., Schamp, B., Soudzilovskaia, N. A.,
Spasojevic, M. J., Sosinski, E., Thornton, P. E., Valladares, F., van Bodegom, P. M., Williams, M., Wirth, C., and Reich, P.





B.: Mapping local and global variability in plant trait distributions, Proceedings of the National Academy of Sciences, 114, E10937, 10.1073/pnas.1708984114, 2017.

Camino, C., Gonzalez-Dugo, V., Hernandez, P., and Zarco-Tejada, P. J.: Radiative transfer Vcmax estimation from hyperspectral imagery and SIF retrievals to assess photosynthetic performance in rainfed and irrigated plant phenotyping trials, Remote Sensing of Environment, 111186, https://doi.org/10.1016/j.rse.2019.05.005, 2019.

Campbell, G.: Principles of Environmental Biophysics, 1977.

Carvalhais, N., Forkel, M., Khomik, M., Bellarby, J., Jung, M., Migliavacca, M., Mu, M., Saatchi, S., Santoro, M., Thurner,

M., Weber, U., Ahrens, B., Beer, C., Cescatti, A., Randerson, J. T., and Reichstein, M.: Global covariation of carbon turnover times with climate in terrestrial ecosystems, Nature, 514, 213, 10.1038/nature13731

https://www.nature.com/articles/nature13731#supplementary-information, 2014.

Casals, P., Gimeno, C., Carrara, A., Lopez-Sangil, L., and Sanz, M. J.: Soil CO2 efflux and extractable organic carbon fractions under simulated precipitation events in a Mediterranean Dehesa, Soil Biology and Biochemistry, 41, 1915-1922,

http://dx.doi.org/10.1016/j.soilbio.2009.06.015, 2009.

Coplen, T. B.: Guidelines and recommended terms for expression of stable-isotope-ratio and gas-ratio measurement results, Rapid Communications in Mass Spectrometry, 25, 2538-2560, 10.1002/rcm.5129, 2011.

Croft, H., Chen, J. M., Luo, X., Bartlett, P., Chen, B., and Staebler, R. M.: Leaf chlorophyll content as a proxy for leaf photosynthetic capacity, Global Change Biology, 23, 3513-3524, 10.1111/gcb.13599, 2017.

Damm, A., Elbers, J. A. N., Erler, A., Gioli, B., Hamdi, K., Hutjes, R., Kosvancova, M., Meroni, M., Miglietta, F., Moersch, A., Moreno, J., Schickling, A., Sonnenschein, R., Udelhoven, T., Van Der Linden, S., Hostert, P., and Rascher, U. W. E.: Remote sensing of sun-induced fluorescence to improve modeling of diurnal courses of gross primary production (GPP), Global Change Biology, 16, 171-186, 10.1111/j.1365-2486.2009.01908.x, 2010.

Dechant, B., Cuntz, M., Vohland, M., Schulz, E., and Doktor, D.: Estimation of photosynthesis traits from leaf reflectance

spectra: Correlation to nitrogen content as the dominant mechanism, Remote Sensing of Environment, 196, 279-292, https://doi.org/10.1016/j.rse.2017.05.019, 2017.

Dorigo, W. A., Zurita-Milla, R., de Wit, A. J. W., Brazile, J., Singh, R., and Schaepman, M. E.: A review on reflective remote sensing and data assimilation techniques for enhanced agroecosystem modeling, International Journal of Applied Earth Observation and Geoinformation, 9, 165-193, https://doi.org/10.1016/j.jag.2006.05.003, 2007.

Drolet, G. G., Huemmrich, K. F., Hall, F. G., Middleton, E. M., Black, T. A., Barr, A. G., and Margolis, H. A.: A MODIS-derived photochemical reflectance index to detect inter-annual variations in the photosynthetic light-use efficiency of a boreal deciduous forest, Remote Sensing of Environment, 98, 212-224, http://dx.doi.org/10.1016/j.rse.2005.07.006, 2005.

DuBois, S., Desai, A. R., Singh, A., Serbin, S. P., Goulden, M. L., Baldocchi, D. D., Ma, S., Oechel, W. C., Wharton, S., Kruger, E. L., and Townsend, P. A.: Using imaging spectroscopy to detect variation in terrestrial ecosystem productivity

across a water-stressed landscape, Ecological Applications, 28, 1313-1324, 10.1002/eap.1733, 2018.



Dutta, D., Schimel, D. S., Sun, Y., van der Tol, C., and Frankenberg, C.: Optimal inverse estimation of ecosystem parameters from observations of carbon and energy fluxes, Biogeosciences, 16, 77-103, 10.5194/bg-16-77-2019, 2019.

El-Madany, T. S., Reichstein, M., Perez-Priego, O., Carrara, A., Moreno, G., Pilar Martín, M., Pacheco-Labrador, J., Wohlfahrt, G., Nieto, H., Weber, U., Kolle, O., Luo, Y.-P., Carvalhais, N., and Migliavacca, M.: Drivers of spatio-temporal

variability of carbon dioxide and energy fluxes in a Mediterranean savanna ecosystem, Agricultural and Forest Meteorology, 262, 258-278, https://doi.org/10.1016/j.agrformet.2018.07.010, 2018.

Farquhar, G. D., von Caemmerer, S., and Berry, J. A.: A biochemical model of photosynthetic CO2 assimilation in leaves of C3 species, Planta, 149, 78-90, 10.1007/BF00386231, 1980.

Farquhar, G. D., and von Caemmerer, S.: Modelling of Photosynthetic Response to Environmental Conditions, in:

Physiological Plant Ecology II: Water Relations and Carbon Assimilation, edited by: Lange, O. L., Nobel, P. S., Osmond, C. B., and Ziegler, H., Springer Berlin Heidelberg, Berlin, Heidelberg, 549-587, 1982.

Feng, X., and Dietze, M.: Scale dependence in the effects of leaf ecophysiological traits on photosynthesis: Bayesian parameterization of photosynthesis models, New Phytologist, 200, 1132-1144, 10.1111/nph.12454, 2013.

Frankenberg, C., Fisher, J. B., Worden, J., Badgley, G., Saatchi, S. S., Lee, J.-E., Toon, G. C., Butz, A., Jung, M., Kuze, A.,

and Yokota, T.: New global observations of the terrestrial carbon cycle from GOSAT: Patterns of plant fluorescence with gross primary productivity, Geophysical Research Letters, 38, doi:10.1029/2011GL048738, 2011.

Friend, A. D., Arneth, A., Kiang, N. Y., Lomas, M., OgÉE, J., RÖDenbeck, C., Running, S. W., Santaren, J.-D., Sitch, S., Viovy, N., Ian Woodward, F., and Zaehle, S.: FLUXNET and modelling the global carbon cycle, Global Change Biology, 13, 610-633, 10.1111/j.1365-2486.2006.01223.x, 2007.

Galeazzi, C., Sacchetti, A., Cisbani, A., and Babini, G.: The PRISMA Program, IGARSS 2008 - 2008 IEEE International Geoscience and Remote Sensing Symposium, 2008, IV - 105-IV - 108,

Gamon, J. A., Peñuelas, J., and Field, C. B.: A narrow-waveband spectral index that tracks diurnal changes in photosynthetic efficiency, Remote Sensing of Environment, 41, 35-44, https://doi.org/10.1016/0034-4257(92)90059-S, 1992.

Gates, D. M.: Transpiration and Leaf Temperature, Annual Review of Plant Physiology, 19, 211-238,

10.1146/annurev.pp.19.060168.001235, 1968.

Goetz, A. F. H.: Three decades of hyperspectral remote sensing of the Earth: A personal view, Remote Sensing of Environment, 113, S5-S16, https://doi.org/10.1016/j.rse.2007.12.014, 2009.

Golub, G. H., and Loan, C. F. v.: An Analysis of the Total Least Squares Problem, SIAM Journal on Numerical Analysis, 17, 883-893, 10.1137/0717073, 1980.

González-Cascón, M. R., Pacheco-Labrador, J., and Martín Isabel, M. P.: Variación temporal del comportamiento espectral y la composición química en el dosel arbóreo de una dehesa, Revista de la Asociación Española de Teledetección, 31-43, 2016.



Gonzalez-Cascon, R., Jiménez-Fenoy, L., Verdú-Fillola, I., and Martín, M. P.: Short communication: Aqueous-acetone extraction improves the drawbacks of using dimethylsulfoxide as solvent for photometric pigment quantification in Quercus
ilex leaves, 2017, 26, 10.5424/fs/2017262-11099, 2017.

Gonzalez-Cascon, R., and Martin, M. P.: Protocol for pigment content quantification in herbaceous covers: sampling and analysis, October 2019, dx.doi.org/10.17504/protocols.io.qs6dwhe, 2018.

Gonzalez-Cascon, R., Pacheco-Labrador, J., Moreno, G., Migliavacca, M., and Martín, M. P.: Estimating canopy biochemistry in mediterranean holm oak (*Quercus ilex*) from proximal and airborne hyperspectral data, International
Geoscience and Remote Sensing Symposium, IGARSS 2018, 22-27 July 2019, Valencia, Spain. (Poster), 2018,

Grimm, V., and Railsback, S. F.: Pattern-oriented modelling: a 'multi-scope' for predictive systems ecology, Philosophical transactions of the Royal Society of London. Series B, Biological sciences, 367, 298-310, 10.1098/rstb.2011.0180, 2012.

Guanter, L., Frankenberg, C., Dudhia, A., Lewis, P. E., Gómez-Dans, J., Kuze, A., Suto, H., and Grainger, R. G.: Retrieval and global assessment of terrestrial chlorophyll fluorescence from GOSAT space measurements, Remote Sensing of
Environment, 121, 236-251, https://doi.org/10.1016/j.rse.2012.02.006, 2012.

Guanter, L., Kaufmann, H., Segl, K., Foerster, S., Rogass, C., Chabrillat, S., Kuester, T., Hollstein, A., Rossner, G., Chlebek, C., Straif, C., Fischer, S., Schrader, S., Storch, T., Heiden, U., Mueller, A., Bachmann, M., Mühle, H., Müller, R., Habermeyer, M., Ohndorf, A., Hill, J., Buddenbaum, H., Hostert, P., van der Linden, S., Leitão, J. P., Rabe, A., Doerffer, R., Krasemann, H., Xi, H., Mauser, W., Hank, T., Locherer, M., Rast, M., Staenz, K., and Sang, B.: The EnMAP Spaceborne
Imaging Spectroscopy Mission for Earth Observation, Remote Sensing, 7, 10.3390/rs70708830, 2015.

Hernández-Clemente, R., Navarro-Cerrillo, R. M., Suárez, L., Morales, F., and Zarco-Tejada, P. J.: Assessing structural effects on PRI for stress detection in conifer forests, Remote Sensing of Environment, 115, 2360-2375, https://doi.org/10.1016/j.rse.2011.04.036, 2011.

Hernández-Clemente, R., North, P. R. J., Hornero, A., and Zarco-Tejada, P. J.: Assessing the effects of forest health on sun-
induced chlorophyll fluorescence using the FluorFLIGHT 3-D radiative transfer model to account for forest structure, Remote Sensing of Environment, 193, 165-179, https://doi.org/10.1016/j.rse.2017.02.012, 2017.

Hikosaka, K., Murakami, A., and Hirose, T.: Balancing carboxylation and regeneration of ribulose-1,5- bisphosphate in leaf photosynthesis: temperature acclimation of an evergreen tree, Quercus myrsinaefolia, Plant, Cell & Environment, 22, 841-849, 10.1046/j.1365-3040.1999.00442.x, 1999.

Hill, M. J., Held, A. A., Leuning, R., Coops, N. C., Hughes, D., and Cleugh, H. A.: MODIS spectral signals at a flux tower site: Relationships with high-resolution data, and CO2 flux and light use efficiency measurements, Remote Sensing of Environment, 103, 351-368, https://doi.org/10.1016/j.rse.2005.06.015, 2006.

Hu, J., Liu, X., Liu, L., and Guan, L.: Evaluating the Performance of the SCOPE Model in Simulating Canopy Solar-Induced Chlorophyll Fluorescence, Remote Sensing, 10, 10.3390/rs10020250, 2018.



Huemmrich, K. F., Campbell, P., Landis, D., and Middleton, E.: Developing a common globally applicable method for optical remote sensing of ecosystem light use efficiency, Remote Sensing of Environment, 230, 111190, https://doi.org/10.1016/j.rse.2019.05.009, 2019.

Industrial Secret: Acta 1741/2018, de 23 de abril, Secreto Industrial titulado: "Protocolo de medición de índice de área foliar en encinas con el instrumento LICOR LAI-2200-C", 2018.

Jacquemoud, S., Ustin, S. L., Verdebout, J., Schmuck, G., Andreoli, G., and Hosgood, B.: Estimating leaf biochemistry using the PROSPECT leaf optical properties model, Remote Sensing of Environment, 56, 194-202, https://doi.org/10.1016/0034-4257(95)00238-3, 1996.

Jarolmasjed, S., Sankaran, S., Kalcsits, L., and Khot, L. R.: Proximal hyperspectral sensing of stomatal conductance to monitor the efficacy of exogenous abscisic acid applications in apple trees, Crop Protection, 109, 42-50,

https://doi.org/10.1016/j.cropro.2018.02.022, 2018.

Joiner, J., Guanter, L., Lindstrot, R., Voigt, M., Vasilkov, A. P., Middleton, E. M., Huemmrich, K. F., Yoshida, Y., and Frankenberg, C.: Global monitoring of terrestrial chlorophyll fluorescence from moderate-spectral-resolution near-infrared satellite measurements: methodology, simulations, and application to GOME-2, Atmos. Meas. Tech., 6, 2803-2823, 10.5194/amt-6-2803-2013, 2013.

Jung, M., Reichstein, M., Margolis, H. A., Cescatti, A., Richardson, A. D., Arain, M. A., Arneth, A., Bernhofer, C., Bonal, D., Chen, J., Gianelle, D., Gobron, N., Kiely, G., Kutsch, W., Lasslop, G., Law, B. E., Lindroth, A., Merbold, L., Montagnani, L., Moors, E. J., Papale, D., Sottocornola, M., Vaccari, F., and Williams, C.: Global patterns of land-atmosphere fluxes of carbon dioxide, latent heat, and sensible heat derived from eddy covariance, satellite, and meteorological observations, Journal of Geophysical Research: Biogeosciences, 116, n/a-n/a, 10.1029/2010jg001566, 2011.

Keenan, T. F., Carbone, M. S., Reichstein, M., and Richardson, A. D.: The model–data fusion pitfall: assuming certainty in an uncertain world, Oecologia, 167, 587, 10.1007/s00442-011-2106-x, 2011.

Kerr, G., Avbelj, J., Carmona, E., Eckardt, A., Gerasch, B., Graham, L., Günther, B., Heiden, U., Krutz, D., Krawczyk, H., Makarau, A., Miller, R., Müller, R., Perkins, R., and Walter, I.: The hyperspectral sensor DESIS on MUSES: Processing and applications, 2016 IEEE International Geoscience and Remote Sensing Symposium (IGARSS), 2016, 268-271,

Kidnie, S., Cruz, M. G., Gould, J., Nichols, D., Anderson, W., and Bessell, R.: Effects of curing on grassfires: I. Fuel dynamics in a senescing grassland, International Journal of Wildland Fire, 24, 828-837, 2015.

Kljun, N., Calanca, P., Rotach, M. W., and Schmid, H. P.: A simple two-dimensional parameterisation for Flux Footprint Prediction (FFP), Geosci. Model Dev., 8, 3695-3713, 10.5194/gmd-8-3695-2015, 2015.

Koetz, B., Baret, F., Poilvé, H., and Hill, J.: Use of coupled canopy structure dynamic and radiative transfer models to

estimate biophysical canopy characteristics, Remote Sensing of Environment, 95, 115-124, https://doi.org/10.1016/j.rse.2004.11.017, 2005.





Lee, C. M., Cable, M. L., Hook, S. J., Green, R. O., Ustin, S. L., Mandl, D. J., and Middleton, E. M.: An introduction to the NASA Hyperspectral InfraRed Imager (HyspIRI) mission and preparatory activities, Remote Sensing of Environment, 167, 6-19, https://doi.org/10.1016/j.rse.2015.06.012, 2015.

Leuning, R.: Modelling Stomatal Behaviour and and Photosynthesis of <I>Eucalyptus grandis</I>, Functional Plant Biology, 17, 159-175, 1990.

Leuning, R.: A critical appraisal of a combined stomatal-photosynthesis model for C3 plants, Plant, Cell & Environment, 18, 339-355, 10.1111/j.1365-3040.1995.tb00370.x, 1995.

Leuning, R., van Gorsel, E., Massman, W. J., and Isaac, P. R.: Reflections on the surface energy imbalance problem,
Agricultural and Forest Meteorology, 156, 65-74, https://doi.org/10.1016/j.agrformet.2011.12.002, 2012.

Limousin, J.-M., Misson, L., Lavoir, A.-V., Martin, N. K., and Rambal, S.: Do photosynthetic limitations of evergreen Quercus ilex leaves change with long-term increased drought severity?, Plant, Cell & Environment, 33, 863-875, 10.1111/j.1365-3040.2009.02112.x, 2010.

Lu, X., Liu, Z., An, S., Miralles, D. G., Maes, W., Liu, Y., and Tang, J.: Potential of solar-induced chlorophyll fluorescence
to estimate transpiration in a temperate forest, Agricultural and Forest Meteorology, 252, 75-87, https://doi.org/10.1016/j.agrformet.2018.01.017, 2018.

Luo, X., Croft, H., Chen, J. M., He, L., and Keenan, T. F.: Improved estimates of global terrestrial photosynthesis using information on leaf chlorophyll content, Global Change Biology, 25, 2499-2514, 10.1111/gcb.14624, 2019.

Luo, Y., El-Madany, T., Filippa, G., Ma, X., Ahrens, B., Carrara, A., Gonzalez-Cascon, R., Cremonese, E., Galvagno, M.,
Hammer, T., Pacheco-Labrador, J., Martín, M., Moreno, G., Perez-Priego, O., Reichstein, M., Richardson, A., Römermann, C., and Migliavacca, M.: Using Near-Infrared-Enabled Digital Repeat Photography to Track Structural and Physiological Phenology in Mediterranean Tree–Grass Ecosystems, Remote Sensing, 10, 1293, 2018.

Luo, Y. Q., Randerson, J. T., Abramowitz, G., Bacour, C., Blyth, E., Carvalhais, N., Ciais, P., Dalmonech, D., Fisher, J. B., Fisher, R., Friedlingstein, P., Hibbard, K., Hoffman, F., Huntzinger, D., Jones, C. D., Koven, C., Lawrence, D., Li, D. J.,
Mahecha, M., Niu, S. L., Norby, R., Piao, S. L., Qi, X., Peylin, P., Prentice, I. C., Riley, W., Reichstein, M., Schwalm, C., Wang, Y. P., Xia, J. Y., Zaehle, S., and Zhou, X. H.: A framework for benchmarking land models, Biogeosciences, 9, 3857-3874, 10.5194/bg-9-3857-2012, 2012.

Marcolla, B., and Cescatti, A.: Geometry of the hemispherical radiometric footprint over plant canopies, Theor. Appl. Climatol., 134, 981-990, 10.1007/s00704-017-2326-z, 2018.

Mauder, M., and Foken, T.: Processing and quality control of eddy covariance measurements, European Geosciences Union General Assembly, Vienna, 2005,

Melendo-Vega, J., Martín, M., Pacheco-Labrador, J., González-Cascón, R., Moreno, G., Pérez, F., Migliavacca, M., García, M., North, P., and Riaño, D.: Improving the Performance of 3-D Radiative Transfer Model FLIGHT to Simulate Optical Properties of a Tree-Grass Ecosystem, Remote Sensing, 10, 2061, 2018.



Mendiguren, G., Pilar Martín, M., Nieto, H., Pacheco-Labrador, J., and Jurdao, S.: Seasonal variation in grass water content estimated from proximal sensing and MODIS time series in a Mediterranean Fluxnet site, Biogeosciences, 12, 5523-5535, 10.5194/bg-12-5523-2015, 2015.

Migliavacca, M., Meroni, M., Busetto, L., Colombo, R., Zenone, T., Matteucci, G., Manca, G., and Seufert, G.: Modeling Gross Primary Production of Agro-Forestry Ecosystems by Assimilation of Satellite-Derived Information in a Process-Based
Model, Sensors, 9, 922, 2009.

Migliavacca, M., Dosio, A., Kloster, S., Ward, D. S., Camia, A., Houborg, R., Houston Durrant, T., Khabarov, N., Krasovskii, A. A., San Miguel-Ayanz, J., and Cescatti, A.: Modeling burned area in Europe with the Community Land Model, Journal of Geophysical Research: Biogeosciences, 118, 265-279, 10.1002/jgrg.20026, 2013.

Migliavacca, M., Perez-Priego, O., Rossini, M., El-Madany, T. S., Moreno, G., van der Tol, C., Rascher, U., Berninger, A.,
Bessenbacher, V., Burkart, A., Carrara, A., Fava, F., Guan, J.-H., Hammer, T. W., Henkel, K., Juarez-Alcalde, E., Julitta, T., Kolle, O., Martín, M. P., Musavi, T., Pacheco-Labrador, J., Pérez-Burgueño, A., Wutzler, T., Zaehle, S., and Reichstein, M.: Plant functional traits and canopy structure control the relationship between photosynthetic CO2 uptake and far-red sun-induced fluorescence in a Mediterranean grassland under different nutrient availability, New Phytologist, n/a-n/a, 10.1111/nph.14437, 2017.

Nair, R. K. F., Morris, K. A., Hertel, M., Luo, Y., Moreno, G., Reichstein, M., Schrumpf, M., and Migliavacca, M.: N : P stoichiometry and habitat effects on Mediterranean savanna seasonal root dynamics, Biogeosciences, 16, 1883-1901, 10.5194/bg-16-1883-2019, 2019.

North, P. R. J.: Three-dimensional forest light interaction model using a Monte Carlo method, IEEE Transactions on Geoscience and Remote Sensing, 34, 946-956, 10.1109/36.508411, 1996.

Omlin, M., and Reichert, P.: A comparison of techniques for the estimation of model prediction uncertainty, Ecological Modelling, 115, 45-59, https://doi.org/10.1016/S0304-3800(98)00174-4, 1999.

Pacheco-Labrador, J., González-Cascón, R., Martín, M. P., and Riaño, D.: Understanding the optical responses of leaf nitrogen in Mediterranean Holm oak (Quercus ilex) using field spectroscopy, International Journal of Applied Earth Observation and Geoinformation, 26, 105-118, http://dx.doi.org/10.1016/j.jag.2013.05.013, 2014.

Pacheco-Labrador, J., Martín, M. P., Riaño, D., Hilker, T., and Carrara, A.: New approaches in multi-angular proximal sensing of vegetation: Accounting for spatial heterogeneity and diffuse radiation in directional reflectance distribution models, Remote Sensing of Environment, 187, 447-457, http://dx.doi.org/10.1016/j.rse.2016.10.051, 2016.

Pacheco-Labrador, J., El-Madany, T., Martín, M., Migliavacca, M., Rossini, M., Carrara, A., and Zarco-Tejada, P. J.: Spatio-Temporal Relationships between Optical Information and Carbon Fluxes in a Mediterranean Tree-Grass Ecosystem, Remote
Sensing, 9, 608, 2017a.

Pacheco-Labrador, J., Perez-Priego, O., El-Madany, T. S., Julitta, T., Rossini, M., Guan, J., Moreno, G., Carvalhais, N., Martín, M. P., Gonzalez-Cascon, R., Kolle, O., Reischtein, M., van der Tol, C., Carrara, A., Martini, D., Hammer, T. W., Moossen, H., and Migliavacca, M.: Multiple-constraint inversion of SCOPE. Evaluating the potential of GPP and SIF for the



retrieval of plant functional traits, Remote Sensing of Environment, 234, 111362, https://doi.org/10.1016/j.rse.2019.111362,
865    2019.

Pacheco-Labrador, J., El-Madany, T. S., van der Tol, C., Martín, M. P., Gonzalez-Cascon, R., Perez-Priego, O., Guan, J.,
Moreno, G., Carrara, A., Reichstein, M., and Migliavacca, M.: senSCOPE: Modeling radiative transfer and biochemical
processes in mixed canopies combining green and senescent leaves with SCOPE, bioRxiv, 2020.2002.2005.935064,
10.1101/2020.02.05.935064, 2020.

Pallas, J. E., Michel, B. E., and Harris, D. G.: Photosynthesis, Transpiration, Leaf Temperature, and Stomatal Activity of
Cotton Plants under Varying Water Potentials, Plant Physiol, 42, 76-88, 10.1104/pp.42.1.76, 1967.

Papale, D., Reichstein, M., Aubinet, M., Canfora, E., Bernhofer, C., Kutsch, W., Longdoz, B., Rambal, S., Valentini, R.,
Vesala, T., and Yakir, D.: Towards a standardized processing of Net Ecosystem Exchange measured with eddy covariance
technique: algorithms and uncertainty estimation, Biogeosciences, 3, 571-583, 10.5194/bg-3-571-2006, 2006.

Perez-Priego, O., El-Madany, T. S., Migliavacca, M., Kowalski, A. S., Jung, M., Carrara, A., Kolle, O., Martín, M. P.,
Pacheco-Labrador, J., Moreno, G., and Reichstein, M.: Evaluation of eddy covariance latent heat fluxes with independent
lysimeter and sapflow estimates in a Mediterranean savannah ecosystem, Agricultural and Forest Meteorology, 236, 87-99,
http://dx.doi.org/10.1016/j.agrformet.2017.01.009, 2017.

Plate, E. J., and Quraishi, A. A.: Modeling of Velocity Distributions Inside and Above Tall Crops, Journal of Applied
Meteorology (1962-1982), 4, 400-408, 1965.

Prichard, S. J., Kennedy, M. C., Andreu, A. G., Eagle, P. C., French, N. H., and Billmire, M.: Next-Generation Biomass
Mapping for Regional Emissions and Carbon Inventories: Incorporating Uncertainty in Wildland Fuel Characterization,
Journal of Geophysical Research: Biogeosciences, n/a, 10.1029/2019JG005083, 2019.

Rast, M., and Painter, T. H.: Earth Observation Imaging Spectroscopy for Terrestrial Systems: An Overview of Its History,
Techniques, and Applications of Its Missions, Surveys in Geophysics, 40, 303-331, 10.1007/s10712-019-09517-z, 2019.

Reichstein, M., Tenhunen, J., Roupsard, O., Ourcival, J.-M., Rambal, S., Miglietta, F., Peressotti, A., Pecchiari, M., Tirone,
G., and Valentini, R.: Inverse modeling of seasonal drought effects on canopy CO2/H2O exchange in three Mediterranean
ecosystems, Journal of Geophysical Research: Atmospheres, 108, 10.1029/2003JD003430, 2003.

Reichstein, M., Falge, E., Baldocchi, D., Papale, D., Aubinet, M., Berbigier, P., Bernhofer, C., Buchmann, N., Gilmanov, T.,
Granier, A., Grünwald, T., Havránková, K., Ilvesniemi, H., Janous, D., Knohl, A., Laurila, T., Lohila, A., Loustau, D.,
Matteucci, G., Meyers, T., Miglietta, F., Ourcival, J.-M., Pumpanen, J., Rambal, S., Rotenberg, E., Sanz, M., Tenhunen, J.,
Seufert, G., Vaccari, F., Vesala, T., Yakir, D., and Valentini, R.: On the separation of net ecosystem exchange into
assimilation and ecosystem respiration: review and improved algorithm, Global Change Biology, 11, 1424-1439,
10.1111/j.1365-2486.2005.001002.x, 2005.

Reichstein, M., Mahecha, M. D., Ciais, P., Seneviratne, S. I., Blyth, E. M., Carvalhais, N., and Luo, Y.: Elk-testing climate-
carbon cycle models: a case for pattern-oriented system analysis, iLEAPS Newsletter, 11, 14–21, 2011.



Rogers, A.: The use and misuse of V c,max in Earth System Models, Photosynthesis Research, 119, 15-29, 10.1007/s11120-013-9818-1, 2014.

Rogers, A., Medlyn Belinda, E., Dukes Jeffrey, S., Bonan, G., Caemmerer, S., Dietze Michael, C., Kattge, J., Leakey Andrew, D. B., Mercado Lina, M., Niinemets, Ü., Prentice, I. C., Serbin Shawn, P., Sitch, S., Way Danielle, A., and Zaehle, S.: A roadmap for improving the representation of photosynthesis in Earth system models, New Phytologist, 213, 22-42, 10.1111/nph.14283, 2016.

Schaefer, K., Schwalm Christopher, R., Williams, C., Arain, M. A., Barr, A., Chen Jing, M., Davis Kenneth, J., Dimitrov, D., Hilton Timothy, W., Hollinger David, Y., Humphreys, E., Poulter, B., Raczka Brett, M., Richardson Andrew, D., Sahoo, A., Thornton, P., Vargas, R., Verbeeck, H., Anderson, R., Baker, I., Black, T. A., Bolstad, P., Chen, J., Curtis Peter, S., Desai Ankur, R., Dietze, M., Dragoni, D., Gough, C., Grant Robert, F., Gu, L., Jain, A., Kucharik, C., Law, B., Liu, S., Lokipitiya, E., Margolis Hank, A., Matamala, R., McCaughey, J. H., Monson, R., Munger, J. W., Oechel, W., Peng, C., Price David, T., Ricciuto, D., Riley William, J., Roulet, N., Tian, H., Tonitto, C., Torn, M., Weng, E., and Zhou, X.: A model-data comparison of gross primary productivity: Results from the North American Carbon Program site synthesis, Journal of Geophysical Research: Biogeosciences, 117, 10.1029/2012JG001960, 2012.

Schaepman, M. E., Ustin, S. L., Plaza, A. J., Painter, T. H., Verrelst, J., and Liang, S.: Earth system science related imaging spectroscopy—An assessment, Remote Sensing of Environment, 113, Supplement 1, S123-S137, http://dx.doi.org/10.1016/j.rse.2009.03.001, 2009.

Schimel, D., Schneider, F. D., Carbon, J., and Participants, E.: Flux towers in the sky: global ecology from space, New Phytologist, 224, 570-584, 10.1111/nph.15934, 2019.

Schwalm, C. R., Williams, C. A., Schaefer, K., Anderson, R., Arain, M. A., Baker, I., Barr, A., Black, T. A., Chen, G., Chen, J. M., Ciais, P., Davis, K. J., Desai, A., Dietze, M., Dragoni, D., Fischer, M. L., Flanagan, L. B., Grant, R., Gu, L., Hollinger, D., Izaurralde, R. C., Kucharik, C., Lafleur, P., Law, B. E., Li, L., Li, Z., Liu, S., Lokipitiya, E., Luo, Y., Ma, S., Margolis, H., Matamala, R., McCaughey, H., Monson, R. K., Oechel, W. C., Peng, C., Poulter, B., Price, D. T., Riciutto, D. M., Riley, W., Sahoo, A. K., Sprintsin, M., Sun, J., Tian, H., Tonitto, C., Verbeeck, H., and Verma, S. B.: A model-data intercomparison of CO2 exchange across North America: Results from the North American Carbon Program site synthesis, Journal of Geophysical Research: Biogeosciences, 115, 10.1029/2009JG001229, 2010.

Segl, K., Guanter, L., Rogass, C., Kuester, T., Roessner, S., Kaufmann, H., Sang, B., Mogulsky, V., and Hofer, S.: EeteS-The EnMAP End-to-End Simulation Tool, IEEE Journal of Selected Topics in Applied Earth Observations and Remote Sensing, 5, 522-530, 10.1109/JSTARS.2012.2188994, 2012.

Sepulcre-Cantó, G., Zarco-Tejada, P. J., Jiménez-Muñoz, J. C., Sobrino, J. A., Miguel, E. d., and Villalobos, F. J.: Detection of water stress in an olive orchard with thermal remote sensing imagery, Agricultural and Forest Meteorology, 136, 31-44, https://doi.org/10.1016/j.agrformet.2006.01.008, 2006.





Serbin, S. P., Singh, A., Desai, A. R., Dubois, S. G., Jablonski, A. D., Kingdon, C. C., Kruger, E. L., and Townsend, P. A.:
Remotely estimating photosynthetic capacity, and its response to temperature, in vegetation canopies using imaging
spectroscopy, Remote Sensing of Environment, 167, 78-87, https://doi.org/10.1016/j.rse.2015.05.024, 2015.

Shan, N., Ju, W., Migliavacca, M., Martini, D., Guanter, L., Chen, J., Goulas, Y., and Zhang, Y.: Modeling canopy
conductance and transpiration from solar-induced chlorophyll fluorescence, Agricultural and Forest Meteorology, 268, 189-
201, https://doi.org/10.1016/j.agrformet.2019.01.031, 2019.

Silva-Perez, V., Molero, G., Serbin, S. P., Condon, A. G., Reynolds, M. P., Furbank, R. T., and Evans, J. R.: Hyperspectral
reflectance as a tool to measure biochemical and physiological traits in wheat, Journal of Experimental Botany, 69, 483-496,
10.1093/jxb/erx421, 2018.

Sims, D. A., and Gamon, J. A.: Relationships between leaf pigment content and spectral reflectance across a wide range of
species, leaf structures and developmental stages, Remote Sensing of Environment, 81, 337-354,
http://doi.org/10.1016/S0034-4257(02)00010-X, 2002.

Smith, G. M., and Milton, E. J.: The use of the empirical line method to calibrate remotely sensed data to reflectance,
International Journal of Remote Sensing, 20, 2653-2662, 10.1080/014311699211994, 1999.

Sprintsin, M., Chen, J. M., Desai, A., and Gough, C. M.: Evaluation of leaf-to-canopy upscaling methodologies against
carbon flux data in North America, Journal of Geophysical Research: Biogeosciences, 117, 10.1029/2010JG001407, 2012.

Suárez, L., Zarco-Tejada, P. J., Berni, J. A. J., González-Dugo, V., and Fereres, E.: Modelling PRI for water stress detection
using radiative transfer models, Remote Sensing of Environment, 113, 730-744, https://doi.org/10.1016/j.rse.2008.12.001,
2009.

Sun, Y., Frankenberg, C., Wood, J. D., Schimel, D. S., Jung, M., Guanter, L., Drewry, D. T., Verma, M., Porcar-Castell, A.,
Griffis, T. J., Gu, L., Magney, T. S., Köhler, P., Evans, B., and Yuen, K.: OCO-2 advances photosynthesis observation from
space via solar-induced chlorophyll fluorescence, Science, 358, 10.1126/science.aam5747, 2017.

Thom, A. S.: Momentum, mass and heat exchange of vegetation, Quarterly Journal of the Royal Meteorological Society, 98,
124-134, 10.1002/qj.49709841510, 1972.

Thorp, K. R., Wang, G., West, A. L., Moran, M. S., Bronson, K. F., White, J. W., and Mon, J.: Estimating crop biophysical
properties from remote sensing data by inverting linked radiative transfer and ecophysiological models, Remote Sensing of
Environment, 124, 224-233, https://doi.org/10.1016/j.rse.2012.05.013, 2012.

Ungar, S. G., Pearlman, J. S., Mendenhall, J. A., and Reuter, D.: Overview of the Earth Observing One (EO-1) mission,
IEEE Transactions on Geoscience and Remote Sensing, 41, 1149-1159, 10.1109/TGRS.2003.815999, 2003.

Ustin, S. L., and Gamon, J. A.: Remote sensing of plant functional types, New Phytologist, 186, 795-816, 10.1111/j.1469-
8137.2010.03284.x, 2010.

van der Tol, C., Verhoef, W., Timmermans, J., Verhoef, A., and Su, Z.: An integrated model of soil-canopy spectral
radiances, photosynthesis, fluorescence, temperature and energy balance, Biogeosciences, 6, 3109-3129, 10.5194/bg-6-3109-
2009, 2009.





Vaz, M., Pereira, J. S., Gazarini, L. C., David, T. S., David, J. S., Rodrigues, A., Maroco, J., and Chaves, M. M.: Drought-induced photosynthetic inhibition and autumn recovery in two Mediterranean oak species (Quercus ilex and Quercus suber), Tree Physiology, 30, 946-956, 10.1093/treephys/tpq044, 2010.

Vaz, M., Maroco, J., Ribeiro, N., Gazarini, L. C., Pereira, J. S., and Chaves, M. M.: Leaf-level responses to light in two co-occurring Quercus (Quercus ilex and Quercus suber): leaf structure, chemical composition and photosynthesis, Agroforestry Systems, 82, 173-181, 10.1007/s10457-010-9343-6, 2011.

Verhoef, W.: Light scattering by leaf layers with application to canopy reflectance modeling: The SAIL model, Remote Sensing of Environment, 16, 125-141, http://dx.doi.org/10.1016/0034-4257(84)90057-9, 1984.

Vilfan, N., Van der Tol, C., Yang, P., Wyber, R., Malenovský, Z., Robinson, S. A., and Verhoef, W.: Extending Fluspect to simulate xanthophyll driven leaf reflectance dynamics, Remote Sensing of Environment, 211, 345-356, https://doi.org/10.1016/j.rse.2018.04.012, 2018.

Walker, A. P., Beckerman, A. P., Gu, L., Kattge, J., Cernusak Lucas, A., Domingues, T. F., Scales Joanna, C., Wohlfahrt, G., Wullschleger, S. D., and Woodward, F. I.: The relationship of leaf photosynthetic traits – Vcmax and Jmax – to leaf nitrogen, leaf phosphorus, and specific leaf area: a meta-analysis and modeling study, Ecology and Evolution, 4, 3218-3235, 10.1002/ece3.1173, 2014.

Walker, A. P., Quaife, T., Bodegom Peter, M., De Kauwe Martin, G., Keenan Trevor, F., Joiner, J., Lomas Mark, R., MacBean, N., Xu, C., Yang, X., and Woodward, F. I.: The impact of alternative trait-scaling hypotheses for the maximum photosynthetic carboxylation rate (Vcmax) on global gross primary production, New Phytologist, 215, 1370-1386, 10.1111/nph.14623, 2017.

Wang, R., Chen, J. M., Luo, X., Black, A., and Arain, A.: Seasonality of leaf area index and photosynthetic capacity for better estimation of carbon and water fluxes in evergreen conifer forests, Agricultural and Forest Meteorology, 279, 107708, https://doi.org/10.1016/j.agrformet.2019.107708, 2019.

Werner, R. A., Bruch, B. A., and Brand, W. A.: ConFlo III – an interface for high precision δ13C and δ15N analysis with an extended dynamic range, Rapid Communications in Mass Spectrometry, 13, 1237-1241, 10.1002/(SICI)1097-0231(19990715)13:13<1237::AID-RCM633>3.0.CO;2-C, 1999.

Werner, R. A., and Brand, W. A.: Referencing strategies and techniques in stable isotope ratio analysis, Rapid Communications in Mass Spectrometry, 15, 501-519, 10.1002/rcm.258, 2001.

Williams, M., Richardson, A. D., Reichstein, M., Stoy, P. C., Peylin, P., Verbeeck, H., Carvalhais, N., Jung, M., Hollinger, D. Y., Kattge, J., Leuning, R., Luo, Y., Tomelleri, E., Trudinger, C. M., and Wang, Y. P.: Improving land surface models with FLUXNET data, Biogeosciences, 6, 1341-1359, 10.5194/bg-6-1341-2009, 2009.

Wolf, A., Akshalov, K., Saliendra, N., Johnson, D. A., and Laca, E. A.: Inverse estimation of Vcmax, leaf area index, and the Ball-Berry parameter from carbon and energy fluxes, Journal of Geophysical Research: Atmospheres, 111, 10.1029/2005JD005927, 2006.





Wullschleger, S. D., Epstein, H. E., Box, E. O., Euskirchen, E. S., Goswami, S., Iversen, C. M., Kattge, J., Norby, R. J., van Bodegom, P. M., and Xu, X.: Plant functional types in Earth system models: past experiences and future directions for application of dynamic vegetation models in high-latitude ecosystems, Annals of Botany, 114, 1-16, 10.1093/aob/mcu077, 2014.

Wutzler, T., and Carvalhais, N.: Balancing multiple constraints in model-data integration: Weights and the parameter block approach, Journal of Geophysical Research: Biogeosciences, 119, 2112-2129, 10.1002/2014JG002650, 2014.

Wutzler, T., Lucas-Moffat, A., Migliavacca, M., Knauer, J., Sickel, K., Šigut, L., Menzer, O., and Reichstein, M.: Basic and extensible post-processing of eddy covariance flux data with REddyProc, Biogeosciences, 15, 5015-5030, 10.5194/bg-15-5015-2018, 2018.

Xie, X., Li, A., Jin, H., Yin, G., and Nan, X.: Derivation of temporally continuous leaf maximum carboxylation rate (Vcmax) from the sunlit leaf gross photosynthesis productivity through combining BEPS model with light response curve at tower flux sites, Agricultural and Forest Meteorology, 259, 82-94, https://doi.org/10.1016/j.agrformet.2018.04.017, 2018.

Xin, Q., Gong, P., and Li, W.: Modeling photosynthesis of discontinuous plant canopies by linking the Geometric Optical Radiative Transfer model with biochemical processes, Biogeosciences, 12, 3447-3467, 10.5194/bg-12-3447-2015, 2015.

Yang, P., Verhoef, W., and van der Tol, C.: The mSCOPE model: A simple adaptation to the SCOPE model to describe reflectance, fluorescence and photosynthesis of vertically heterogeneous canopies, Remote Sensing of Environment, 201, 1-11, https://doi.org/10.1016/j.rse.2017.08.029, 2017.

Zarco-Tejada, P. J., González-Dugo, V., and Berni, J. A. J.: Fluorescence, temperature and narrow-band indices acquired from a UAV platform for water stress detection using a micro-hyperspectral imager and a thermal camera, Remote Sensing

of Environment, 117, 322-337, https://doi.org/10.1016/j.rse.2011.10.007, 2012.

Zarco-Tejada, P. J., González-Dugo, V., Williams, L. E., Suárez, L., Berni, J. A. J., Goldhamer, D., and Fereres, E.: A PRI-based water stress index combining structural and chlorophyll effects: Assessment using diurnal narrow-band airborne imagery and the CWSI thermal index, Remote Sensing of Environment, 138, 38-50, https://doi.org/10.1016/j.rse.2013.07.024, 2013.

Zhang, Y., Guanter, L., Berry Joseph, A., Joiner, J., Tol, C., Huete, A., Gitelson, A., Voigt, M., and Köhler, P.: Estimation of vegetation photosynthetic capacity from space-based measurements of chlorophyll fluorescence for terrestrial biosphere models, Global Change Biology, 20, 3727-3742, 10.1111/gcb.12664, 2014.

Zhang, Y., Guanter, L., Joiner, J., Song, L., and Guan, K.: Spatially-explicit monitoring of crop photosynthetic capacity through the use of space-based chlorophyll fluorescence data, Remote Sensing of Environment, 210, 362-374,

https://doi.org/10.1016/j.rse.2018.03.031, 2018.

Zheng, T., Chen, J., He, L., Arain, M. A., Thomas, S. C., Murphy, J. G., Geddes, J. A., and Black, T. A.: Inverting the maximum carboxylation rate (Vcmax) from the sunlit leaf photosynthesis rate derived from measured light response curves at tower flux sites, Agricultural and Forest Meteorology, 236, 48-66, https://doi.org/10.1016/j.agrformet.2017.01.008, 2017.





Zhou, Y., Ju, W., Sun, X., Hu, Z., Han, S., Black, T. A., Jassal, R. S., and Wu, X.: Close relationship between spectral
vegetation indices and Vcmax in deciduous and mixed forests, Tellus B: Chemical and Physical Meteorology, 66, 23279,
10.3402/tellusb.v66.23279, 2014.