# Peer review of "Combining hyperspectral remote sensing and eddy covariance data streams for estimation of vegetation functional traits."

_Biogeosciences, 2019_

## Referee Comment (RC1) · Anonymous Referee #1 · 26 Mar 2020

Review for Pacheco-Labrador, biogeosciences

The paper by Pacheco-Laborador et al. jointly uses airborne hyperspectral reflectance data and eddy covariance data to retrieve ecosystem traits in a Mediterranean tree-grass ecosystem. They use 17 hyperspectral images over three different flux towers (control, N addition, N+P addition) in an inversion framework which couples radiative transfer and soil-vegetation atmosphere transfer using a modified version of the SCOPE model to incorporate leaf senescence. The results suggest that such a framework can estimate vegetation traits and energy fluxes in this ecosystem. The authors also 'scale' their results using synthetic emulated hyperspectral satellite imagery to

place in the context of future hyperspectral missions.

The work described here is a significant effort, integrating many different datasets collected across a range of temporal and spatial scales over the course of 6 years. While this effort is very much appreciated, the many different data sources and complexity of the approach make it challenging to review. It requires a significant amount of background knowledge on the many papers previously published by the authors to completely understand the approach. Despite my best attempt at this, I still found this manuscript very difficult to evaluate. There are too many assumptions made for a complete evaluation of the paper's rigor, leaving the reader to have to place a lot of trust in the authors. If the assumptions are indeed valid (but again, too many to look into to fully address each one) then the paper comes across as a sound methodological approach. Despite these limitations, I think there is value in this work, but I would recommend the authors consider re-evaluating how to best distill this complex story into something more tangible and coherent.

To me (and I could be missing the point), the paper reads very much like a methodological paper, perhaps better fit for a journal like EGU's Geoscientific Model Development. The paper does not go far enough into describing the "interactions between the biological, chemical, and physical processes in terrestrial or extraterrestrial life with the geosphere, hydrosphere, and atmosphere" – as stated as a goal of Biogeosciences. There is very little information regarding what the authors have learned about this ecosystem; the main result is that a seemingly complex approach can produce key functional parameters of vegetation that are robust to several sources of uncertainty. The discussion of sources of uncertainty, in particular, is extremely robust and very much appreciated.

Based on the strengths of this paper, I would suggest a path forward might be to remove the analysis in Figs. 5 and 7. To me, these figures raise more questions than answers. The attempt by the authors to say something more ecological about how vegetation traits co-vary takes away from the paper. Focusing on the key results, Fig. 3, 4, and 6 (Figs. 1 and 2 are also nice) seems like it would help to distill the information content. A

reduction in the amount of parameters the authors are trying to predict might also help (moving the rest to the supplementary material). Focusing on a few key vegetation parameters – as opposed to trying to model everything, all at once – followed by a concrete discussion on where and why model-data mismatch or over/underprediction happens might also be a path forward. Currently - while there is a lot of good content in the discussion - it should relate explicitly back to the key results and answer bigger questions about how such analytical techniques could be used to map vegetation traits going forward. I realize this a fairly vague suggestion, but a substantial reframing of the story will also help this paper reach a broader audience.

Minor comments are as follows:

Abstract (and elsewhere): The authors use the word "prove" to describe their findings. This language is too strong, consider "suggest."

The first sentence of the introduction, I would perhaps not mention just climate change as an application for this work, as climate change is never again discussed and by using it as the only potential application, it implies that this might feature into the work more prominently.

The introduction is well written, the authors touch on pretty much every aspect of the paper. If one had time to read all of these papers from a wide range of disciplines, it would certainly make the methods and results easier to interpret. In order to reach a broader audience, I'd suggest a little more 'hand-holding' though, particularly with regard to what exactly some of the plant functional traits are and why they are important.

One main point made clear in the introduction is that an attempt to jointly retrieve functional traits using hyperspectral imagery combined with EC data is lacking. But it's not clear why we need this? How are the other methods failing that require this new approach?

Line 111: The authors note that only one of the examples from the previous

paragraph validates retrievals against actual measurements from gas-exchange measurements. . .but this paper doesn't do that. They make assumptions about other traits or use data from existing literature in combinations that is difficult to follow.

Lines 124-130: The attempt to relate this work to future satellite missions is appreciated, but the amount of detail necessary to introduce readers to how the emulation works is lacking.

Line 143: Describe CT. . .I'm guessing Control Treatment

Line 157: 'mayor' to 'major'. . .there are quite a few grammatical mistakes throughout, I won't comment on them, but please address these. Given the large quantity of co-authors, one would think these could be addressed.

Table 1: This table is appreciated, but for the many other variables used during this entire study it would help to add them as additional columns.

Line 213: Why aren't data from these additional campaigns included?

Line 210-250: There are many assumptions made regarding the biophysical variables used. For example, deriving Vcmax from Nmass,green and a relationship from an existing paper. While there isn't much of an alternative, it should be noted that many of the biophysical parameters are very much inferred.

Line 269: To assume that carotenoid concentration will covary with Chl concentration (derived from a SPAD meter) is one example of gross oversimplification.

Line 299: 'close to solar noon'. Are the actual flight times used to compare to the EC data? Solar noon is much less relevant here as are the incident irradiance conditions. Diffuse/direct fraction, time of year, solar zenith angle. . .it's unclear how these are considered.

Figure 2: This is a useful figure, the axes need labels.

Figure 3: The x axes are displayed as a time series, at equally spaced intervals that

[Figure]

make this difficult to interpret. Consider removing the vertical lines and the x ticks, and simply note the date of acquisition horizontally in each shaded or non-shaded bin.

Figure 4: Many of these fits violate the assumptions of linear regression, in which case I don't think it's useful to include a line of best fit, or the statistics. Also the figure legend has subplots labeled wrong.

Figure 5 and Fig 7: I feel that these take away from the main message the authors are trying to communicate. How are these figures adding to the main results? Am I missing something?

Another confusing part of the analysis is that it appears as if the authors are predicting ecosystem traits, which are a combination of both grass and tree

Generally, there is a lot of good information in the discussion. However, much of it reads as 'intro' material and it does not relate directly back to the results. While a lot of the points regarding uncertainty are important, I do not feel as though the discussion drives home the main results, or how such an analysis could be used in the future. The authors have a deep understanding about many of the uncertainties associated with their approach, and that is much appreciated. It is of my (potentially naïve) opinion, that their discussion is not useful for informing future research that is conducted outside of their own particular research group. I would advise the authors to pay close attention to how this work is perceived by individuals outside of their niche team. After all, this will not only help the authors consider the broader importance of their work, but it will help the rest of the research community.

---

## Referee Comment (RC2) · Anonymous Referee #2 · 27 Mar 2020

The work by Pacheco-Labrador et al. attempt to combine measured and emulated hyperspectral images with Eddy covariance (EC) flux measurements, to retrieve the tree-grass ecosystem physiological traits. In the work, the authors use a running fertilization experiment to build a model to predicate the ecosystem physiological traits such as Vcmax and Ball-berry slope parameter (m). the authors do an inversion to the SCOPE model, and specifically the senSCOPE model that takes into consideration the senescence of leaves in the ecosystem. The measurements include three flux towers, one for each fertilization treatment. High spatial resolution airborne hyperspectral images have been taken during the experiment over the experiment. Also, isotopic samples were taken from the ecosystem as well. The emulated data used to

introduce the potential of future satellite missions for ecosystem physiological traits retrieval. It is highly noticeable that the works contain a large amount of data from many years of measurements. Moreover, the combination of the fertilization treatments, in the heterogeneous ecosystem, using a wide range of measurements should bring to a robust understanding of the ecosystem physiological behavior. The use in the SCOPE model also allows to combine spectral and physical parameters measurements to retrieve ecosystem physiological parameters. However, reading through the manuscript leads to the filling that the current work was mainly the building of the model and less to achieve an understanding of the ecosystem relation between the measured spectral data and the physiological traits. Moreover, it is a bit problematic to estimate the model performance in Infront of other estimated values (with their on uncertainties) and not with actual measured values. Estimation of Vcmax from leaves N content (which is also estimated part of the times, according to the authors) required a large number of assumptions and should be done carefully. To my opinion, this work has a high potential to bring to a better understanding of the ecosystem physiological response through hyperspectral and EC measurements, however, several changes are required:

- All the graphs (except to figures 2&3) do not mention the fertilization treatments, maybe this addition can explain part of the variance in the graphs.

- It looks like the summer measurements are not responding to the model, maybe the authors should consider excluding these results from the model, or at least to model them separately.

- In general, the discussion is mainly explaining the technical reasons for the model behavior against the other parameters. Maybe connecting the model to the actual physiology measured in the field will lead to a better understanding of the model strengths and weaknesses.

Short comments through the MS:

- Line 77: Reference is required. - Line 79-80: This is a very simplified assumption.

many works demonstrated that the atmospheric demand is highly relevant to the transpiration and stomatal response.

- Line 86-87: Reference is required.

- Line 112: Authors should consider referring to Fu et al. (2020), PCE "Estimating photosynthetic traits from reflectance spectra: A synthesis of spectral indices, numerical inversion, and partial least square regression".

- Line 215-216: WC is a tricky parameter; the leaf relative water content is a more reliable parameter in terms of plant water status.

- Line 226: it is not clear how N content was measured please explain or add a reference.

- Line 226: which model was used? Reference.

- Line 230: Please note, if it is possible, if the estimation of Cab was done from estimated Nmass or only from measured values.

- Fig. 4: Fig. 4 please fit the letters in the legend to the figure.

- Fig. 6: to see all the points on the graph and avoid overlap, the authors should make them a bit transparent.

- Fig. 7 please add parameters to the fitted curve and RMSE value.

- Line 565: replace "response".

---

## Author Comment (AC1) · 28 Apr 2020

Dear Referee #1, we would like to thank the valuable comments received. Different modifications have been planned accordingly in order to improve the readability and to better present the manuscript contents. We think part of the suggestions and criticisms received are motivated by an unclear description of the implications of the work for the community, as well as an unclear description of the aims and the methodology for the evaluation of the results. Therefore, we will improve the description of the aims, the motivations behind this analysis and improve the readability.

We try also to clarify the aim and contribution of this manuscript here: Literature shows

that understanding and modeling of carbon and water fluxes suffers from the static parameterization of plant functional traits (see Rogers et al., 2017 or Walker et al., 2017); in this context, the use of novel remote sensing data, such as hyperspectral imagery, can contribute to better monitor and characterize vegetation function (see Schimel et al., 2019). However, traits describing vegetation function are only weakly encoded in the optical reflectance vegetation; and previous works involving coauthors of this manuscript showed that to retrieve such traits, hyperspectral data must be combined with thermal radiation and fluxes (Pacheco-Labrador et al., 2019). This idea is also summarized by Schimel et al., (2019). Our manuscript hypothesize that it is possible characterizing the temporal variability of functional traits at ecosystem scale combining eddy covariance and remote sensing imagery (which in this case is emulated from airborne data since no time series of satellite hyperspectral imagery are available at the study site yet). We demonstrate that this is possible with certain limitations that we thoroughly discussed. This manuscript provides the Biogeosciences community with an innovative methodology for the estimation of key functional traits in eddy covariance stations. This work can be a first step towards the characterization of functional traits that could happen when we have hyperspectral data available at several ecosystem stations. Later on, once functional traits have been characterized in a sufficiently large number of sites and ecosystems it would be possible globally upscaling this information (see Moreno et al., 2018 or Walker et al., 2017); filling this way a knowledge gap that limits the understanding and modeling of carbon and water fluxes. Therefore, we consider that our manuscript makes a relevant contribution for the Biogeosciences community and is suitable for this journal.

We will modify the manuscript to better explain and justify this idea and the logic behind our analyses; and to better explain the potential of the method in a broader context. This is also discussed in the point-by-point reply to the Referee #1 comments and questions below.

Also, notice that in the new version we have introduced two changes:

1) A bug in the code that preserved carotenoids in the senescent leaves was been corrected. This has produced minimal differences in the results compared with the previous manuscript version.

2) A third step in the inversion has been implemented and tested to improve the characterization of the relationship between soil moisture and soil resistance to evaporation from the pore space. This had been only commented as a possibility in the discussion, but has been tested in the new version to confirm whether this could increase the certainty of this characterization without strongly modifying the estimates of other functional parameters.

References:

Rogers, A., Medlyn, B.E., Dukes, J.S., Bonan, G., von Caemmerer, S., Dietze, M.C., Kattge, J., Leakey, A.D.B., Mercado, L.M., Niinemets, Ü., Prentice, I.C., Serbin, S.P., Sitch, S., Way, D.A., & Zaehle, S. (2017). A roadmap for improving the representation of photosynthesis in Earth system models. New Phytologist, 213, 22-42Walker, A.P., Beckerman, A.P., Gu, L., Kattge, J., Cernusak Lucas, A., Domingues, T.F., Scales Joanna, C., Wohlfahrt, G., Wullschleger, S.D., & Woodward, F.I. (2014). The relationship of leaf photosynthetic traits – Vcmax and Jmax – to leaf nitrogen, leaf phosphorus, and specific leaf area: a meta‐analysis and modeling study. Ecology and Evolution, 4, 3218-3235 Schimel, D., Schneider, F.D., Carbon, J., & Participants, E. (2019). Flux towers in the sky: global ecology from space. New Phytologist, 224, 570-584 Pacheco-Labrador, J., Perez-Priego, O., El-Madany, T.S., Julitta, T., Rossini, M., Guan, J., Moreno, G., Carvalhais, N., Martín, M.P., Gonzalez-Cascon, R., Kolle, O., Reischtein, M., van der Tol, C., Carrara, A., Martini, D., Hammer, T.W., Moossen, H., & Migliavacca, M. (2019). Multiple-constraint inversion of SCOPE. Evaluating the potential of GPP and SIF for the retrieval of plant functional traits. Remote Sensing of Environment, 234, 111362 Moreno-Martínez, Á., Camps-Valls, G., Kattge, J., Robinson, N., Reichstein, M., van Bodegom, P., Kramer, K., Cornelissen, J.H.C., Reich, P., Bahn, M., Niinemets, Ü., Peñuelas, J., Craine, J.M., Cerabolini, B.E.L., Minden, V.,

Laughlin, D.C., Sack, L., Allred, B., Baraloto, C., Byun, C., Soudzilovskaia, N.A., & Running, S.W. (2018). A methodology to derive global maps of leaf traits using remote sensing and climate data. Remote Sensing of Environment, 218, 69-88

Referee #1 comment: The paper by Pacheco-Labrador et al. jointly uses airborne hyperspectral reflectance data and eddy covariance data to retrieve ecosystem traits in a Mediterranean tree-grass ecosystem. They use 17 hyperspectral images over three different flux towers (control, N addition, N+P addition) in an inversion framework which couples radiative transfer and soil-vegetation atmosphere transfer using a modified version of the SCOPE model to incorporate leaf senescence. The results suggest that such a framework can estimate vegetation traits and energy fluxes in this ecosystem. The authors also 'scale' their results using synthetic emulated hyperspectral satellite imagery to place in the context of future hyperspectral missions. The work described here is a significant effort, integrating many different datasets collected across a range of temporal and spatial scales over the course of 6 years. While this effort is very much appreciated, the many different data sources and complexity of the approach make it challenging to review. It requires a significant amount of background knowledge on the many papers previously published by the authors to completely understand the approach. Despite my best attempt at this, I still found this manuscript very difficult to evaluate. There are too many assumptions made for a complete evaluation of the paper's rigor, leaving the reader to have to place a lot of trust in the authors. If the assumptions are indeed valid (but again, too many to look into to fully address each one) then the paper comes across as a sound methodological approach. Despite these limitations, I think there is value in this work, but I would recommend the authors consider re-evaluating how to best distill this complex story into something more tangible and coherent.

Authors' response: We agree with the Referee #1 that this manuscript makes use of a wide range of measurements and observations, and there are some assumptions that are discussed in previous works of the co-authors. We agree that probably we did

not explain the details in an easy way to follow by the reader without knowing previous our work. In order to improve the readability of the manuscript we will modify the methodological section including a diagram showing how these datasets are generated and combined. Moreover, we will include, in the supplementary material, a more extended description of the generation and scaling of field observations and estimates of biophysical parameters in order to facilitate a better understanding of the different datasets, so there is no need to consult additional literature.

Referee #1 comment: To me (and I could be missing the point), the paper reads very much like a methodological paper, perhaps better fit for a journal like EGU's Geoscientific Model Development. The paper does not go far enough into describing the "interactions between the biological, chemical, and physical processes in terrestrial or extraterrestrial life with the geosphere, hydrosphere, and atmosphere" – as stated as a goal of Biogeosciences. There is very little information regarding what the authors have learned about this ecosystem; the main result is that a seemingly complex approach can produce key functional parameters of vegetation that are robust to several sources of uncertainty. The discussion of sources of uncertainty, in particular, is extremely robust and very much appreciated.

Authors' response: We agree with the Referee #1 that our manuscript presents a methodology; however we still think it fits Biogeosciences' goals as the method proposed allows estimating key biophysical parameters but also -and this is the most innovative part-, functional parameters of vegetation, and their relationship with other variables. This is done by combining hyperspectral remote sensing and eddy covariance datastreams. These estimates are relevant to improve our understanding of the vegetation-atmosphere interactions and to parameterize terrestrial biosphere models. For instance Rogers et al., (2017) suggests the use for novel remote sensing datastreams (e.g. hyperspectral satellite missions) as a way forward to characterize the evolution in time of functional traits useful for earth system modeling. Our work was inspired by Rogers et al (2017) and we think that the methodology developed can be of

interest for a community that is beyond the model developers. Also, our work is not a strict modeling work. Indeed we do not develop any new model (notice that senSCOPE is described and evaluated in a manuscript currently under review that has been openly archived in Pacheco-Labrador et al., (2020)); rather we develop a model data integration schemes combining a variety of measurements. Similar approaches are behind recent papers published in Biogeosciences. For example, Dutta et al. (2019) presented a different method to estimate Vcmax and the Ball-Berry slope (m) combining remote sensing and eddy covariance data. Also Biogeosciences publishes articles that, even if methodological can be of interest for the community, for example Papale et al., (2006), and more recently Wutzler et al., (2018), or Kang et al., (2018) presented packages or methods to gap-fill and partition water and/or carbon fluxes measured with eddy covariance.

We leave the final decision to the Editor but we think that, like in these and other similar articles, our manuscript focuses on understanding the advantages and caveats of the method and its potential to provide robust estimates of parameters that are meaningful for the understanding and modeling of interactions between vegetation and atmosphere.

References:

Rogers, A., Medlyn, B.E., Dukes, J.S., Bonan, G., von Caemmerer, S., Dietze, M.C., Kattge, J., Leakey, A.D.B., Mercado, L.M., Niinemets, Ü., Prentice, I.C., Serbin, S.P., Sitch, S., Way, D.A., & Zaehle, S. (2017). A roadmap for improving the representation of photosynthesis in Earth system models. New Phytologist, 213, 22-42 Pacheco-Labrador, J., El-Madany, T.S., van der Tol, C., Martín, M.P., Gonzalez-Cascon, R., Perez-Priego, O., Guan, J., Moreno, G., Carrara, A., Reichstein, M., & Migliavacca, M. (2020). senSCOPE: Modeling radiative transfer and biochemical processes in mixed canopies combining green and senescent leaves with SCOPE. bioRxiv, 2020.2002.2005.935064 Dutta, D., Schimel, D.S., Sun, Y., van der Tol, C., & Frankenberg, C. (2019). Optimal inverse estimation of ecosystem parameters from

observations of carbon and energy fluxes. Biogeosciences, 16, 77-103 Papale, D., Re-ichstein, M., Aubinet, M., Canfora, E., Bernhofer, C., Kutsch, W., Longdoz, B., Rambal, S., Valentini, R., Vesala, T., & Yakir, D. (2006). Towards a standardized processing of Net Ecosystem Exchange measured with eddy covariance technique: algorithms and uncertainty estimation. Biogeosciences, 3, 571-583 Wutzler, T., Lucas-Moffat, A., Migliavacca, M., Knauer, J., Sickel, K., Šigut, L., Menzer, O., & Reichstein, M. (2018). Basic and extensible post-processing of eddy covariance flux data with REddyProc. Biogeosciences, 15, 5015-5030 Kang, M., Kim, J., Malla Thakuri, B., Chun, J., & Cho, C. (2018). New gap-filling and partitioning technique for H2O eddy fluxes measured over forests. Biogeosciences, 15, 631-647

Referee #1 comment: Based on the strengths of this paper, I would suggest a path forward might be to remove the analysis in Figs. 5 and 7. To me, these figures raise more questions than answers. The attempt by the authors to say something more ecological about how vegetation traits co-vary takes away from the paper. Focusing on the key results, Fig. 3, 4, and 6 (Figs. 1 and 2 are also nice) seems like it would help to distill the information content. A reduction in the amount of parameters the authors are trying to predict might also help (moving the rest to the supplementary material). Focusing on a few key vegetation parameters – as opposed to trying to model everything, all at once – followed by a concrete discussion on where and why model-data mismatch or over/underprediction happens might also be a path forward.

Authors' response: We appreciate the thoughts of the Referee #1 how to better present and align the results. Still, we feel that these are key results to the manuscript and want explain in more details why they should stay within the main text. Figures 5 and 7 do not aim to attach any ecological meaning to the retrievals, but to indirectly evaluate their feasibility, since direct evaluation is not possible. Notice that overarching goal of this manuscript is allowing the remote estimation of functional parameters such as the evaluated in these figures; these parameters are not traditionally estimated from satellite imagery since they have little effect on vegetation reflectance. This will be

better explained in the manuscript.

Here we detail part of this rationale: Remote sensing has traditionally provided estimates of biophysical parameters such as LAI or pigment's content which have a relatively strong influence on the signals perceived from remote sensors (e.g., reflected radiances). The interest on the estimation of parameters describing plant functions which have little effect on remote sensing signals, such as Vcmax or the Ball-Berry slope m, is lately increasing. This is the main contribution and innovation of our manuscript; which proposes a method combining radiative transfer, energy balance and photosynthesis models with remote sensing data and eddy covariance fluxes to constrain these functional parameters describing vegetation function. Nonetheless, these biophysical parameters have to be also retrieved since they strongly control the absorption of light and therefore photosynthesis and energy balance.

Figures 4, 5 and 7 evaluate the estimates of biophysical and functional parameters using different approaches, which are selected according to the field data available. In order to understand the quality of the estimates of functional parameters, we need first to understand the capability of the approach to estimate the biophysical parameters, since these have a strong control on the absorption of radiation in the canopy. However, the direct evaluation of all these parameters is not always possible. One of the challenges that this and other research works face in Mediterranean tree-grass ecosystems (and others) is the evaluation of remote sensing based estimates. This ecosystem features high species richness and spatial variability in the grassland, and a structural heterogeneity imposed by the coexistence of scattered tress and the grassland itself. This spatial variability must be accounted for during the estimation of ecosystem-scale vegetation parameters; thus measurements must be taken at different locations and vegetation types, and then integrated according to the representativeness of the different samples in the ecosystem. Therefore, ecosystem-scaled parameters always carry uncertainties arising during their integration. Despite these uncertainties, sufficient sampling effort (e.g., number, distribution and size of samples) allows obtaining robust

scaled values, representative of the study area averages. Years of sampling experience have shown the researchers working in this site the needs, in terms of sampling strategies, to characterize some vegetation biophysical parameters such as leaf biochemical contents or leaf area index (e.g., see Mendiguren et al., 2015 or Melendo-Vega et al., 2017). Therefore, we have relied on these ground-based estimates, scaled at ecosystem level, to evaluate biophysical vegetation parameters estimated with the approach proposed in this manuscript.

Specifically, the assessment of the biophysical parameters analysis is presented in Figure 4 and Figure 5a-b. When available, ecosystem-scaled measurements of the biophysical parameters are directly compared with the estimates. However, in the case of chlorophyll no field measurements in the grassland are available for several of the field campaigns; for this reason, this parameter is also evaluated indirectly. We used the relationship between chlorophyll content (Cab) and nitrogen (N) of the data available in these and other unrelated campaigns (see Melendo-Vega et al., 2017) to estimate grass Cab when missing, and then we scaled using trees Cab estimated in the field with a SPAD meter. On the contrary, the measurement of Cab in trees leaves using a SPAD meter took place in all the campaigns. It relies on solid and extensive datasets as well as on laboratory analyses that coauthors of this manuscript specifically refined to improve the photometric determination of pigments in the Holm oak leaves (Gonzalez-Cascon et al., 2017). Since most of the field estimates rely on the grass Cab-N relationship, we do not compare estimated and field Cab directly, but rather look at their relationship with N at ecosystem scale. This will be also clarified in the text.

After assessing biophysical parameters, we assess the retrieval of functional parameters of vegetation in Figure 5c-h and of a functional ecosystem relationship in Figure 7. However, the aim of Figure 5 and 7 is evaluating the plausibility of the functional parameter estimates; not establishing ecological conclusions about how vegetation traits co-vary. Functional parameters such as Vcmax or the Ball-Berry model slope m cannot be determined from bulk samples of vegetation; but must be measured leaf by leaf

using gas exchange chambers for long periods of time (e.g. 40 min). Considering that the objective of such measurements would be providing values of the functional parameters representative of the eddy covariance footprint, the high spatial variability and species richness (quite evenly distributed in the grassland) would make necessary a huge number of measurements which could not be acquired due to time and technical limitations. Also, since these measurements would be species-based, the up-scaling process would be prone to high uncertainty. This problem is not only related to the study site, but to extensive areas comprehending numerous species or to diverse and rich ecosystems. For these reasons, we propose alternative methods (pattern-oriented evaluation approach) to assess estimates of functional parameters of vegetation. This approach relies on the capability of the model to reproduce expected patterns, either from the literature or from observations, with no prior knowledge about them. In order to evaluate Vcmax we rely on its relationship with nitrogen (N) assuming that the larger the presence of N in the leaf, the larger is the chance that this is placed in the Rubisco enzyme, therefore enhancing Vcmax. The specific relationship between both variables is species-dependent and changes according to different plant strategies. However, the existence of a positive relationship between N and Vcmax is known and has been shown for different vegetation types in the literature (e.g., Quebbeman and Ramirez, 2016; Walker et al., 2014; Feng and Dietze, 2013 or Kattge et al., 2011). Therefore, we exploit this knowledge to assess whether our estimates are plausible and reproduce expected relationships with other parameters or they are just loose equifinality or ill-posed solutions of the inversion. We are aware that there might be sources of uncertainty, but the fact that Vcmax scales with N according to what expect from a large body of literature shows that the retrieval of Vcmax is realistic. In the case of the Ball-Berry model slope m, we use the 13C discrimination as discussed and suggested by Seibt et al., 2008. Under certain conditions 13C discrimination and water use efficiency are inversely related. We are also aware of the limitations of this approach (e.g. Seibt et al., 2008; Medlyn et al., 2017); which we discussed in the manuscript. We took measures to minimize the effect of these additional factors, for example, we evaluated also underlying water use efficiency to minimize the effect of VPD, and considered only estimates close to the peak of the season, since 13C discrimination represents an integrative process whereas the m and uWUE vary in time or are rather instantaneous. Also in this case, the fact that the relationship found between the estimated m parameter and the independent measure of 13C is coherent with literature give us confidence on the robustness of the methodology.

In the case of Figure 7, we assess the retrievals of the soil resistance to evaporation from the pore space (rss); this resistance is known to increase as soil dries Mohamed et al., (1997), and is affected by other factors such as physical properties of soil, which make this relationship site-specific (e.g., Lawrence et al., 2011;Swenson and Lawrence 2014). Therefore, we use this knowledge to assess if the retrievals of rss are plausible. We acknowledge that this parameter is also potentially loose in the inversion, since its effect on the model outputs can saturate above some threshold (for Pacheco-Labrador et al., 2019); and in fact, the relationships between rss and soil moisture content presented in Figure 7 are poorly fit due to the presence of extreme values. Aware of this fact, we have implemented and tested a third step in the inversion where the relationship presented in Figure 7 is used as a prior to repeat the inversion carried out in Step #2. This leads to a much closer fit of the relationship between rss and soil moisture and more importantly, has little effect on the retrieval of Vcmax and m. This process was suggested in the discussion of the manuscript, but not carried out. We will include these results in the new version of the manuscript to show that a more robust relationship can be obtained.

The evaluation of our estimates is as thorough as possible given the constraints imposed by the ecosystem under study. We have carried out an evaluation effort not typically present in this sort of analyses, in order to assess the feasibility of our method to provide plausible estimates; however, this process requires relying on some assumptions; which we acknowledge in the manuscript. Notice that many of the works dealing with the inversion of SCOPE evaluate Vcmax against NDVI, or simply compare

predicted and observed fluxes. Our work proposes the use of new evaluation methods that could contribute to other studies in the future. This is also relevant since functional parameters such as Vcmax and m cannot be measured from destructive sampling of vegetation which can allow integrating the variability of the vegetation without specifically accounting for it; therefore technical and resource limitations to obtain validation data of these parameters is prone to appear in many other ecosystems featuring high species richness and variability; or when remote sensors feature low or mid spatial resolutions and therefore the estimates represent large areas.

We will more strongly justify this rationale in the manuscript. We will also better detail which are the assumptions behind the evaluations we carried out, especially in the case of the functional parameters, and clarify in the discussion what could be the consequences of their violation. We will strength the discussion of the relevance of indirect evaluations when direct one are not feasible; this is necessary since functional parameters are more and more often estimated from remote sensing, but not direct assessment is typically available at this scales.

References:

Pacheco-Labrador, J., Perez-Priego, O., El-Madany, T.S., Julitta, T., Rossini, M., Guan, J., Moreno, G., Carvalhais, N., Martín, M.P., Gonzalez-Cascon, R., Kolle, O., Reischtein, M., van der Tol, C., Carrara, A., Martini, D., Hammer, T.W., Moossen, H., & Migliavacca, M. (2019). Multiple-constraint inversion of SCOPE. Evaluating the potential of GPP and SIF for the retrieval of plant functional traits. Remote Sensing of Environment, 234, 111362 Gonzalez-Cascon, R., Jiménez-Fenoy, L., Verdú-Fillola, I., & Martín, M.P. (2017). Short communication: Aqueous-acetone extraction improves the drawbacks of using dimethylsulfoxide as solvent for photometric pigment quantification in Quercus ilex leaves. 2017, 26 Mendiguren, G., Pilar Martín, M., Nieto, H., Pacheco-Labrador, J., & Jurdao, S. (2015). Seasonal variation in grass water content estimated from proximal sensing and MODIS time series in a Mediterranean Fluxnet site. Biogeosciences, 12, 5523-5535 Melendo-Vega, J., Martín, M., Pacheco-Labrador,

J., González-Cascón, R., Moreno, G., Pérez, F., Migliavacca, M., García, M., North, P., & Riaño, D. (2018). Improving the Performance of 3-D Radiative Transfer Model FLIGHT to Simulate Optical Properties of a Tree-Grass Ecosystem. Remote Sensing, 10, 2061 Quebbeman, J.A., & Ramirez, J.A. (2016). Optimal allocation of leaf-level nitrogen: Implications for covariation of Vcmax and Jmax and photosynthetic downregulation. Journal of Geophysical Research: Biogeosciences, 121, 2464-2475 Walker, A.P., Beckerman, A.P., Gu, L., Kattge, J., Cernusak Lucas, A., Domingues, T.F., Scales Joanna, C., Wohlfahrt, G., Wullschleger, S.D., & Woodward, F.I. (2014). The relationship of leaf photosynthetic traits – Vcmax and Jmax – to leaf nitrogen, leaf phosphorus, and specific leaf area: a meta‐analysis and modeling study. Ecology and Evolution, 4, 3218-3235 Feng, X., & Dietze, M. (2013). Scale dependence in the effects of leaf ecophysiological traits on photosynthesis: Bayesian parameterization of photosynthesis models. New Phytologist, 200, 1132-1144 Kattge, J., Díaz, S., Lavorel, S., Prentice I, C., et al. (2011). TRY – a global database of plant traits. Global Change Biology, 17, 2905-2935 Seibt, U., Rajabi, A., Griffiths, H., & Berry, J.A. (2008). Carbon isotopes and water use efficiency: sense and sensitivity. Oecologia, 155, 441 Medlyn, B.E., De Kauw, e.M.G., Lin, Y.S., Knauer, J., Duursma, R.A., Williams, C.A., Arneth, A., Clement, R., Isaac, P., Limousin, J.M., Linderson, M.L., Meir, P., Martin‐StPaul, N., & Wingate, L. (2017). How do leaf and ecosystem measures of water‐use efficiency compare? New Phytologist, 216, 758-770 Mohamed, A.A.-R., Watanabe, K., & Kurokaw, U. (1997). Simple Method for Determining The Bare Soil Resistance to Evaporation. Journal of Groundwater Hydrology, 39, 97-113 Lawrence, D.M., Oleson, K.W., Flanner, M.G., Thornton, P.E., Swenson, S.C., Lawrence, P.J., Zeng, X., Yang, Z.-L., Levis, S., Sakaguchi, K., Bonan, G.B., & Slater, A.G. (2011). Parameterization improvements and functional and structural advances in Version 4 of the Community Land Model. Journal of Advances in Modeling Earth Systems, 3 Swenson, S.C., & Lawrence, D.M. (2014). Assessing a dry surface layer-based soil resistance parameterization for the Community Land Model using GRACE and FLUXNET-MTE data. Journal of Geophysical Research: Atmospheres, 119, 10,299-210,312
Referee #1 comment: Currently - while there is a lot of good content in the discussion - it should relate explicitly back to the key results and answer bigger questions about how such analytical techniques could be used to map vegetation traits going forward. I realize this a fairly vague suggestion, but a substantial reframing of the story will also help this paper reach a broader audience.

Authors' response: At the beginning of the discussion section we stated that this approach could be used in eddy covariance networks to characterize functional properties at large scale; which could lately contribute to improve our estimates and predictions of global carbon and water fluxes. We will strengthen this part of the discussion better showing the potential of the method at global scale over networks of eddy covariance stations, partly following the rationale shown in the first comment presented to Referee #1. We will discuss about the possibility of applying this method to a sufficiently large number of ecosystems and the later possibility of up-scaling these estimates globally sensu Moreno et al., (2018) or Walker et al., (2017).

References:

Moreno-Martínez, Á., Camps-Valls, G., Kattge, J., Robinson, N., Reichstein, M., van Bodegom, P., Kramer, K., Cornelissen, J.H.C., Reich, P., Bahn, M., Niinemets, Ü., Peñuelas, J., Craine, J.M., Cerabolini, B.E.L., Minden, V., Laughlin, D.C., Sack, L., Allred, B., Baraloto, C., Byun, C., Soudzilovskaia, N.A., & Running, S.W. (2018). A methodology to derive global maps of leaf traits using remote sensing and climate data. Remote Sensing of Environment, 218, 69-88 Walker, A.P., Beckerman, A.P., Gu, L., Kattge, J., Cernusak Lucas, A., Domingues, T.F., Scales Joanna, C., Wohlfahrt, G., Wullschleger, S.D., & Woodward, F.I. (2014). The relationship of leaf photosynthetic traits – Vcmax and Jmax – to leaf nitrogen, leaf phosphorus, and specific leaf area: a meta‐analysis and modeling study. Ecology and Evolution, 4, 3218-3235.

Minor comments are as follows:

Referee #1 comment: Abstract (and elsewhere): The authors use the word "prove"

to describe their findings. This language is too strong, consider "suggest." The first sentence of the introduction, I would perhaps not mention just climate change as an application for this work, as climate change is never again discussed and by using it as the only potential application, it implies that this might feature into the work more prominently.

Authors' response: We will replace prove by suggest. "Climate change" will be replaced by "environmental changes"

Referee #1 comment: The introduction is well written, the authors touch on pretty much every aspect of the paper. If one had time to read all of these papers from a wide range of disciplines, it would certainly make the methods and results easier to interpret. In order to reach a broader audience, I'd suggest a little more 'hand-holding' though, particularly with regard to what exactly some of the plant functional traits are and why they are important.

Authors' response: We will stress the need of obtaining estimates of plant functional traits from remote sensing in the introduction and why they are important for a broader audience. We will further develop the explanation of how the use of fixed values of these parameters in terrestrial biosphere models induce uncertainties in the prediction of global carbon and water fluxes; and we will detail what exactly these functional parameters represent in the photosynthetic process and why they are important.

Referee #1 comment: One main point made clear in the introduction is that an attempt to jointly retrieve functional traits using hyperspectral imagery combined with EC data is lacking. But it's not clear why we need this? How are the other methods failing that require this new approach?

Authors' response: We will extend this point in the introduction and in the discussion. Alternative methods exist, relying for example on the inversion of terrestrial biosphere models as presented in the introduction. These often make use of remote sensing products describing the spatial and temporal distributions of biophysical parameters

such as LAI. Other works have exploited optical signals such as sun induced fluo-
rescence or optical and thermal imagery. However, photosynthetic plant traits such
as Vcmax or Ball-Berry m have little influence on optical signals; and might be spuri-
ously related with these (e.g., for Vcmax, via chlorophyll). However, the combination
of remote sensing and eddy covariance data: 1) brings the best of both worlds: high
temporal frequency of fluxes and spatially resolved information of remote sensors and
2) multiple-constraint approaches combining remote sensing and eddy covariance in-
formation allowing for a simultaneous estimation of biophysical and functional traits,
regularizing the inverse problem.

Line 111: The authors note that only one of the examples from the previous paragraph
validates retrievals against actual measurements from gas-exchange measurements
but this paper doesn't do that. They make assumptions about other traits or use data
from existing literature in combinations that is difficult to follow. Authors' response: The
aim of this statement was showing that field data for the evaluation of these estimates
are not usually available, and that in some ecosystem or at certain scales their acquisi-
tion could just be not possible. Thus new methods to evaluate such retrievals, like the
ones used in this manuscript are needed. We will rephrase this statement to make this
idea neater.

Lines 124-130: The attempt to relate this work to future satellite missions is appre-
ciated, but the amount of detail necessary to introduce readers to how the emulation
works is lacking.

Authors' response: We think that a detailed description of the functioning of a remote
sensing mission emulator is out of the scope of this manuscript, and it is addressed to a
publication fully describing this tool. However, we will briefly describe what an emulator
is and what it does to generate synthetic imagery

Line 143: Describe CT. . .I'm guessing Control Treatment

Authors' response: Thanks for noticing this, the description of this acronym will be

added.

Line 157: 'mayor' to 'major'. . .there are quite a few grammatical mistakes throughout, I won't comment on them, but please address these. Given the large quantity of coauthors, one would think these could be addressed.

Authors' response: This correction will be applied. The updated version of the manuscript will be carefully reviewed by a native speaker.

Table 1: This table is appreciated, but for the many other variables used during this entire study it would help to add them as additional columns.

Authors' response: We will add an additional table with the description of all the variables.

Line 213: Why aren't data from these additional campaigns included?

Authors' response: Field campaigns including vegetation destructive sampling are carried out regularly in the study site. However, not all these campaigns are carried out simultaneously to the acquisition of hyperspectral airborne imagery. Due to logistic constrains some variables were not measured in all the airborne campaigns. We have gap-filled variables missing in some of the airborne campaigns exploiting annual time series in the case of the trees, which are much less dynamic than the grassland, or the relationships between variables measured at the site in some of the airborne campaigns used in the manuscript as well as others. We will improve the description of these processes in the methods section and in the supplementary material produced to better describe this gap filling and the scaling of field measurements.

Line 210-250: There are many assumptions made regarding the biophysical variables used. For example, deriving Vcmax from Nmass,green and a relationship from an existing paper. While there isn't much of an alternative, it should be noted that many of the biophysical parameters are very much inferred.

Authors' response: As previously discussed, the generation of values representative

of the ecosystem requires an integration process to scale spatial samples from trees and grasses. This exercise is compulsory in an ecosystem with the structure and species richness as the one under study; this limitation was already acknowledged in the discussion section. However, we will state more clearly the existence of this scaling process and better described it in the new version of the manuscript.

Concerning the connection Vcmax – N, as we will also clarify in connection with a previous comment, that we use this relationship as an indirect evaluation of our estimates and that the use of literature data is just a reference to compare patterns. We will better justify and describe the aim and limitations of the pattern-oriented evaluation of our estimates in the manuscript.

Line 269: To assume that carotenoid concentration will covary with Chl concentration (derived from a SPAD meter) is one example of gross oversimplification.

Authors' response: We are aware that the relationship between chlorophyll and carotenoids (Car) content is more complex. This choice is a compromise between equifinality of the inversion and accuracy of the prediction. We also included random noise in this relationship to increase variability of the Cab / Car ratio. We tried to use a "generalizable" relationship according to the ratio found by Sims and Gamon 2002 in several species where pigments were determined by leaf extractions and a spectrophotometer. The ratio reported by Sims and Gamon 2002 is similar to values determined from vegetation samples and laboratory analyses in our study site; however, these values were not used to prove that a more general relationship could be used instead of a local one; and that the method did not necessarily depend on this site-specific information. Nonetheless, specificities of different ecosystems can require adapting this assumption. Notice that Sims and Gamon 2002 determined pigments concentrations using a spectrophotometer and pigment e, not from a SPAD meter. The ratio reported by Sims and Gamon 2002 was only used to train the neural network predicting the green fraction of LAI from averaged leaf parameters, so that this variable was not totally unconstrained during inversion. Estimated Cab and Car do not stick to the

relationship during the training of the Neural Network, which proves that the relationship was not forced into the solution. This is described in the manuscript presenting senSCOPE model; however, in order to clarify this and support our choice, this fact and the comparison with the Cab / Car relationship found in our site will be included in the discussion.

References:

Sims, D.A., & Gamon, J.A. (2002). Relationships between leaf pigment content and spectral reflectance across a wide range of species, leaf structures and developmental stages. Remote Sensing of Environment, 81, 337-354 Pacheco-Labrador, J., El-Madany, T.S., van der Tol, C., Martín, M.P., Gonzalez-Cascon, R., Perez-Priego, O., Guan, J., Moreno, G., Carrara, A., Reichstein, M., & Migliavacca, M. (2020). senSCOPE: Modeling radiative transfer and biochemical processes in mixed canopies combining green and senescent leaves with SCOPE. bioRxiv, 2020.2002.2005.935064

Line 299: 'close to solar noon'. Are the actual flight times used to compare to the EC data? Solar noon is much less relevant here as are the incident irradiance conditions. Diffuse/direct fraction, time of year, solar zenith angle. It's unclear how these are considered.

Authors' response: For the Step#1 of the inversion, data are matched to the time of the overpass. Illumination conditions are considered for each individual overpass since solar angles are inputs of the radiative transfer model. Diffuse and direct irradiances are internally estimated by senSCOPE from standard atmospheric transfer functions and scaled according to observed long and short wave down-welling irradiances measured by the eddy covariance sensors. In order to clarify this, we will improve this description in the manuscript.

Figure 2: This is a useful figure, the axes need labels.

Authors' response: Axes labels will be added

Figure 3: The x axes are displayed as a time series, at equally spaced intervals that make this difficult to interpret. Consider removing the vertical lines and the x ticks, and simply note the date of acquisition horizontally in each shaded or non-shaded bin.

Authors' response: X-ticks will be removed and dates will be centered to the period of each campaign. However, shaded areas will be left to separate the different campaigns. Notice that not all the campaigns were carried out when three eddy covariance towers operated at the site. Before April 2014 only the control tower was present.

Figure 4: Many of these fits violate the assumptions of linear regression, in which case I don't think it's useful to include a line of best fit, or the statistics. Also the figure legend has subplots labeled wrong.

Authors' response: The references to the subplots in the caption will be corrected. Regarding the assumptions of the linear regression, we carried out Shapiro-Wilk and Levene's tests on the residuals of all the linear models adjusted in Figure 4 as well as in Figure 5e-f. We will only plot the regression lines when the hypothesis of normality and homoscedasticity of the residuals could not be rejected for a significance level of 0.05. This will be clarified in the manuscript as "Shapiro-Wilk (Shapiro and Wilk, 1965) and Levene's (Levene and Olkin, 1960) tests assessed the normality and homoscedasticity of the model residuals in all the cases with a 95 % of confidence, respectively. Linear regression models are shown only when these assumptions could not be rejected." Consequently the models and statistics will be preserved in the figures when meet statistical assumptions are met.

References:

Shapiro, S.S., & Wilk, M.B. (1965). An Analysis of Variance Test for Normality (Complete Samples). Biometrika, 52, 591-611 Levene, H., & Olkin, I. (1960). Robust tests for equality of variances.

Figure 5 and Fig 7: I feel that these take away from the main message the authors are

trying to communicate. How are these figures adding to the main results? Am I missing something? Another confusing part of the analysis is that it appears as if the authors are predicting ecosystem traits, which are a combination of both grass and tree

Authors' response: As we explained in a previous comment, these figures are relevant for the analysis of our results, and for the assessment of parameters with a more functional nature with little effect on spectroradiometric signals captured by remote sensors. We will better stress these ideas in the new version of the manuscript.

Referee #1 comment: Generally, there is a lot of good information in the discussion. However, much of it reads as 'intro' material and it does not relate directly back to the results. While a lot of the points regarding uncertainty are important, I do not feel as though the discussion drives home the main results, or how such an analysis could be used in the future. The authors have a deep understanding about many of the uncertainties associated with their approach, and that is much appreciated. It is of my (potentially naïve) opinion, that their discussion is not useful for informing future research that is conducted outside of their own particular research group. I would advise the authors to pay close attention to how this work is perceived by individuals outside of their niche team. After all, this will not only help the authors consider the broader importance of their work, but it will help the rest of the research community.

Authors' response: Thanks for the comments and suggestions. We discussed many of them in the responses above and we hope we have clarified the aim and objectives of the manuscript. We will extend the discussion, especially the first part to stress the potential of this method and the need to test it in more and different ecosystems. It should be noted that we demonstrated that the method is applicable in 3 different eddy covariance systems and with multiple imagery. We therefore think the method can be generally used, and the fact that it has been tested in a challenging ecosystem suggests that it could better perform in ecosystems where model assumptions are better met. The next steps will be an application on multiple sites with multiple hyperspectral imageries as soon as they will be available from recent or forthcoming space mission

such as EnMAP, PRISMA, SBG, and/or DESIS, among others.

Regarding the part of the discussion on the uncertainties, we will try to streamline this section to meet the comment of Referee #1, but we also think that an open discussion on the uncertainties is very useful for the community. Some of the uncertainties discussed are specific of the ecosystem under study but not exclusive and a can affect also grasslands, or other ecosystems structurally heterogeneous. For example, it is well known that unidimensional homogeneous radiative transfer models do not accurately represent canopies with strong geometrical scattering components due to the presence of occluding volumes. It is also known that the absorption coefficients and the refractive index used by leaf radiative transfer models are effective averages determined from different species. Thus, the properties of some types of vegetation might not be always accurately represented. Our manuscript does not cover a large and diverse range of ecosystems, but we deal with problems that, with some differences, can be found in other remote sensing studies and sites. We have tried to reinforce this idea in the discussion, and reinforce the need of thorough evaluation of these estimates. Notice, that the aim of this manuscript and the main value is not a set of estimated parameters, but the method its robustness, and the potential of the alternative methods for evaluation of estimates, which is what we aim to discuss. In the discussion will emphasize the general applicability of this method to other ecosystems, and that results for those are likely to be better as many of the complexities from the analyzed ecosystem might not occur.

---

## Author Comment (AC2) · 28 Apr 2020

Authors' response: We would like to thank Referee #2 for the valuable comments. We will modify the manuscript to better explain many of the aspects that might have not been sufficiently clear. Some of the comments of Referee #2 suggest a change of the aim of our manuscript in order to exploit the existence of a fertilization experiment in the study area. However, the evaluation of the effects of fertilization is not the main target of this manuscript. Our work rather presents and evaluates the performance of a method capable of providing estimates of biophysical and functional parameters of vegetation combining hyperspectral and eddy covariance data. We will better stress the aim and

scope and improve the discussion about the potential use of this approach at broader scales in the new version of the manuscript. We will also include in the discussion some considerations about the connection of our estimates with the fertilization, more specifically, by analyzing if the estimates are precise enough to discriminate these effects. More specific questions are addressed below.

Also, notice that in the new version we have introduced two changes:

1) A bug in the code that preserved carotenoids in the senescent leaves was been corrected. This has produced minimal differences in the results compared with the previous manuscript version.

2) A third step in the inversion has been implemented and tested to improve the characterization of the relationship between soil moisture and soil resistance to evaporation from the pore space. This had been only commented as a possibility in the discussion, but has been tested in the new version to confirm whether this could increase the certainty of this characterization without strongly modifying the estimates of other functional parameters.

Referee #2 comment: The work by Pacheco-Labrador et al. attempt to combine measured and emulated hyperspectral images with Eddy covariance (EC) flux measurements, to retrieve the tree-grass ecosystem physiological traits. In the work, the authors use a running fertilization experiment to build a model to predict the ecosystem physiological traits such as Vcmax and Ball-berry slope parameter (m). the authors do an inversion to the SCOPE model, and specifically the senSCOPE model that takes into consideration the senescence of leaves in the ecosystem. The measurements include three flux towers, one for each fertilization treatment. High spatial resolution airborne hyperspectral images have been taken during the experiment over the experiment. Also, isotopic samples were taken from the ecosystem as well. The emulated data used to introduce the potential of future satellite missions for ecosystem physiological traits retrieval. It is highly noticeable that the works contain a large amount of data

from many years of measurements. Moreover, the combination of the fertilization treatments, in the heterogeneous ecosystem, using a wide range of measurements should bring to a robust understanding of the ecosystem physiological behavior. The use in the SCOPE model also allows to combine spectral and physical parameters measurements to retrieve ecosystem physiological parameters. However, reading through the manuscript leads to the filling that the current work was mainly the building of the model and less to achieve an understanding of the ecosystem relation between the measured spectral data and the physiological traits.

Authors' response: We agree with Referee #2 that the aim of our manuscript is not achieving an understanding of the ecosystem relation between the measured spectral data and the physiological traits. We aim to provide a method capable of filling a knowledge gap on the spatio-temporal distributions of key functional traits controlling carbon and water exchange, -such as Vcmax or the Ball-Berry slope m- by combining novel spaceborne hyperspectral imagery and eddy covariance fluxes. This gap implies for example the use of tabulated values of these parameters in terrestrial biosphere models which inflate uncertainties of predicted fluxes (see Rogers et al., 2017 or Walker et al., 2017); in this context, the use of novel remote sensing data, such as hyperspectral imagery, can contribute to better monitor and characterize vegetation function (Schimel et al., 2019)). This manuscript is a first step in a hypothesis that if successful, would allow estimating the temporal variability of these parameters in numerous ecosystem stations covering different biomes; and later on use this information to globally upscale this information (see Moreno et al., 2018 or Walker et al., 2017); which would allow filling a knowledge gap that limits the understanding and modeling of carbon and water fluxes.

Therefore, our manuscript focuses on presenting and testing the robustness of a method that could potentially be the basis of a global spatiotemporal characterization of key functional traits of vegetation. In this context, we do not aim to assess the effects of fertilization but rather the capability of capturing temporal dynamics. We use a study

site undergoing manipulation since it offers some advantages to test this methodology such as repeated eddy covariance fluxes in each campaign (three towers operate) and spatial variability in the biochemical, structure and function of vegetation that allow us to understand the robustness of the retrievals. The evaluation of vegetation responses to fertilization is a very interesting research question and several coauthors of this manuscript lead research in that direction, existing currently several works in preparation for submission, under review, or even accepted for publication in different journals. In general, these works require time series of different observations which are denser than the yearly airborne campaigns that we use in this manuscript.

We would also like to stress that in this manuscript we do not develop senSCOPE; this model is presented in another manuscript that for practical purposes has been openly archived (Pacheco-Labrador et al., 2020) and that is currently under review in another journal. The current manuscript presents an inversion approach that can be applied both to senSCOPE and SCOPE with the main aim of simultaneously estimating biophysical and functional traits. We test this approach in a complex ecosystem combining two vegetation layers with very different properties and phenology. Results suggest that the method is robust to several sources of uncertainty and that it would likely perform even better in other sites where models assumptions are better met.

We will clarify the aim and overarching goal of our manuscript in the text. Also, we will discuss which parameters might have been estimated with precision enough to reproduce responses expected from fertilization, in particular of Nitrogen.

References:

Rogers, A., Medlyn Belinda, E., Dukes Jeffrey, S., Bonan, G., Caemmerer, S., Dietze Michael, C., Kattge, J., Leakey Andrew, D.B., Mercado Lina, M., Niinemets, Ü., Prentice, I.C., Serbin Shawn, P., Sitch, S., Way Danielle, A., & Zaehle, S. (2017). A roadmap for improving the representation of photosynthesis in Earth system models. New Phytologist, 213, 22-42 Walker, A.P., Beckerman, A.P., Gu, L., Kattge, J., Cernusak Lucas,

A., Domingues, T.F., Scales Joanna, C., Wohlfahrt, G., Wullschleger, S.D., & Woodward, F.I. (2014). The relationship of leaf photosynthetic traits – Vcmax and Jmax – to leaf nitrogen, leaf phosphorus, and specific leaf area: a meta‐analysis and modeling study. Ecology and Evolution, 4, 3218-3235 Schimel, D., Schneider, F.D., Carbon, J., & Participants, E. (2019). Flux towers in the sky: global ecology from space. New Phytologist, 224, 570-584 Moreno-Martínez, Á., Camps-Valls, G., Kattge, J., Robinson, N., Reichstein, M., van Bodegom, P., Kramer, K., Cornelissen, J.H.C., Reich, P., Bahn, M., Niinemets, Ü., Peñuelas, J., Craine, J.M., Cerabolini, B.E.L., Minden, V., Laughlin, D.C., Sack, L., Allred, B., Baraloto, C., Byun, C., Soudzilovskaia, N.A., & Running, S.W. (2018). A methodology to derive global maps of leaf traits using remote sensing and climate data. Remote Sensing of Environment, 218, 69-88 Pacheco-Labrador, J., El-Madany, T.S., van der Tol, C., Martín, M.P., Gonzalez-Cascon, R., Perez-Priego, O., Guan, J., Moreno, G., Carrara, A., Reichstein, M., & Migliavacca, M. (2020). senSCOPE: Modeling radiative transfer and biochemical processes in mixed canopies combining green and senescent leaves with SCOPE. bioRxiv, 2020.2002.2005.935064

Referee #2 comment: Moreover, it is a bit problematic to estimate the model performance in Infront of other estimated values (with their on uncertainties) and not with actual measured values. Estimation of Vcmax from leaves N content (which is also estimated part of the times, according to the authors) required a large number of assumptions and should be done carefully.

Authors' response: This comment is in part related with the fact that the assessments of the estimates and the databases involved have not been adequately described in the manuscript. In connection with comments made by Referee #1, we will improve the description of the different datasets generated and used in the manuscript as well as better justify how the evaluation of the different parameters is carried out. Part of these details will be presented in an additional figure as well as in a supplementary material that will improve the understanding of how and why this evaluation is done.

We acknowledge that the evaluation Cab and Vcmax is indirect, and mainly relies on

nitrogen (N). We must stress that N was measured in the grassland and the trees in all the field campaigns where it is reported and was neither gap-filled nor estimated; this might not have been clear from the manuscript. This evaluation is not a direct comparison with observations of the assessed parameters as we acknowledge in the manuscript, but it is rather an alternative approach to overcome the lack of field observations. Since direct observations are not available for all the campaigns, we used additional estimates of these parameters as a reference to compare our estimates. However, we are aware and acknowledge that this is not a direct comparison. The lack of field observations representative of eddy covariance footprint-scale areas is usual problem in remote sensing, especially for functional traits controlling gas exchange rates such as Vcmax and m. Measurements of these parameters are strongly resource limited and strongly subject of scaling uncertainties when the study features numerous species. In this context, one of the contributions of the manuscript is proposing the use pattern-oriented evaluation approach to assess the estimates of functional traits that might not be frequently available the field.

One of the challenges that this and many other research works face is the evaluation of remote sensing based estimates. Mediterranean tree-grass (and other) ecosystems feature high species richness and spatial variability in the grassland, and a heterogeneity imposed by the coexistence of scattered tress and the grassland itself. This spatial variability must be accounted for during the estimation of ecosystem-scale vegetation parameters; which means that measurements must be taken at different locations and vegetation types, and then integrated according to the representativeness of the different samples in the ecosystem. This requires a sampling large enough (e.g., number, distribution and size of samples) to provide robust values, representative of the ecosystem. This is possible for biophysical parameters estimated via destructive sampling of vegetation material or canopy-scale technics: leaf biochemical contents or leaf area index. These parameters can be determined from samples of vegetation where the representativeness and individual values of the parameters of each species do not need to be individually measured; which reduces uncertainties propagated in the upscaling (e.g., sampling all the vegetation material inside quadrants of known area). Years of sampling experience have shown the researchers working in our study site, the sampling strategy required to properly characterize the different vegetation types, according to their spatial and temporal variability (e.g., see Mendiguren et al., 2015 or Melendo-Vega et al., 2017).

However, functional parameters such as Vcmax or the Ball-Berry model slope m cannot be determined from bulk samples of vegetation; but must be measured leaf by leaf using gas exchange chambers for long periods of time (e.g. 40 min). Considering that the objective of such measurements would be providing values of the functional parameters representative of the eddy covariance footprint, the high spatial variability and species richness (quite evenly distributed in the grassland) would make necessary a huge number of measurements which could not be acquired due to time and technical limitations. Also, since these measurements would be species-based, the up-scaling process would be prone to high uncertainty. This problem is not only related to the study site, but to extensive areas comprehending numerous species or to diverse and rich ecosystems. For these reasons, we propose alternative methods (pattern-oriented evaluation approach) to assess estimates of functional parameters of vegetation. This approach relies on the capability of the model to reproduce expected patterns, either from the literature or from observations, with no prior knowledge about them. In order to evaluate Vcmax we rely on its relationship with nitrogen (N) assuming that the larger the presence of N in the leaf, the larger is the chance that this is placed in the Rubisco enzyme, therefore enhancing Vcmax. The specific relationship between both variables is species-dependent and changes according to different plant strategies. However, the existence of a positive relationship between N and Vcmax is known and has been shown for different vegetation types in the literature (e.g., Quebbeman and Ramirez, 2016; Walker et al., 2014; Feng and Dietze, 2013 or Kattge et al., 2011). Therefore, we exploit this knowledge to assess whether our estimates are plausible and reproduce expected relationships with other parameters or they are just loose equifinality or ill-posed solutions of the inversion. We are aware that there might sources of uncertainty

but the fact that Vcmax scales with N according to what expect from a large body of literature shows that the retrieval of Vcmax is realistic. In the case of the Ball-Berry model slope m, we use the 13C discrimination as discussed and suggested by Seibt et al., 2008. Under certain conditions 13C discrimination and water use efficiency are inversely related. We are also aware of the limitations of this approach (e.g. Seibt et al., 2008; Medlyn et al., 2017); which we discussed in the manuscript. We took measures to minimize the effect of these additional factors, for example, we evaluated also underlying water use efficiency to minimize the effect of VPD, and considered only estimates close to the peak of the season, since 13C discrimination represents an integrative process whereas the m and uWUE vary in time or are rather instantaneous.

Another vegetation parameter that had to be evaluated indirectly was chlorophyll content since no field measurements in the grassland were available for most of the field campaigns; for this reason, this parameter was also evaluated indirectly. We used the relationship between chlorophyll content (Cab) and nitrogen (N) of the data available in these and other unrelated campaigns (see Melendo-Vega et al., 2017) to estimate grass Cab when missing, and then we scaled using trees Cab estimated in the field with a SPAD meter. On the contrary, the measurement of Cab in trees leaves using a SPAD meter took place in all the campaigns. It relies on solid and extensive datasets as well as on laboratory analyses that coauthors of this manuscript specifically refined to improve the photometric determination of pigments in the Holm oak leaves (Gonzalez-Cascon et al., 2017). Since most of the field estimates rely on the grass Cab-N relationship, we did not compare estimated and field Cab directly, but we rather looked at their relationship with N at ecosystem scale. This will be also clarified in the text.

The evaluation of our estimates is as thorough as possible given the constraints imposed by the ecosystem under study and the availability of data. We have carried out an evaluation effort not typically addressed in this sort of analysis, in order to provide plausible estimates; however, this process requires relying on some assumptions;

which we acknowledge in the manuscript. Many of the works dealing with the inversion of SCOPE evaluate Vcmax against NDVI, or simply comparing predicted and observed fluxes. Our work proposes the use of new evaluation methods that could contribute to other studies in the future.

In order to make this rationale more clear, we have stressed the assumptions behind the evaluations we carried out, especially in the case of the functional parameters, and increased the discussion of the consequences of their violation. We have also stressed the relevance of indirect evaluations when direct one are not feasible; this is necessary since functional parameters are more and more often evaluated from remote sensing, but not direct assessment is always available at this scales.

References:

Mendiguren, G., Pilar Martín, M., Nieto, H., Pacheco-Labrador, J., & Jurdao, S. (2015). Seasonal variation in grass water content estimated from proximal sensing and MODIS time series in a Mediterranean Fluxnet site. Biogeosciences, 12, 5523-5535 Melendo-Vega, J., Martín, M., Pacheco-Labrador, J., González-Cascón, R., Moreno, G., Pérez, F., Migliavacca, M., García, M., North, P., & Riaño, D. (2018). Improving the Performance of 3-D Radiative Transfer Model FLIGHT to Simulate Optical Properties of a Tree-Grass Ecosystem. Remote Sensing, 10, 2061 Quebbeman, J.A., & Ramirez, J.A. (2016). Optimal allocation of leaf-level nitrogen: Implications for covariation of Vcmax and Jmax and photosynthetic downregulation. Journal of Geophysical Research: Biogeosciences, 121, 2464-2475 Walker, A.P., Beckerman, A.P., Gu, L., Kattge, J., Cernusak Lucas, A., Domingues, T.F., Scales Joanna, C., Wohlfahrt, G., Wullschleger, S.D., & Woodward, F.I. (2014). The relationship of leaf photosynthetic traits – Vcmax and Jmax – to leaf nitrogen, leaf phosphorus, and specific leaf area: a meta‐analysis and modeling study. Ecology and Evolution, 4, 3218-3235 Feng, X., & Dietze, M. (2013). Scale dependence in the effects of leaf ecophysiological traits on photosynthesis: Bayesian parameterization of photosynthesis models. New Phytologist, 200, 1132-1144 Kattge, J., Díaz, S., Lavorel, S., Prentice I, C., et al. (2011). TRY – a

global database of plant traits. Global Change Biology, 17, 2905-2935 Seibt, U., Rajabi, A., Griffiths, H., & Berry, J.A. (2008). Carbon isotopes and water use efficiency: sense and sensitivity. Oecologia, 155, 441 Medlyn, B.E., De Kauw, e.M.G., Lin, Y.S., Knauer, J., Duursma, R.A., Williams, C.A., Arneth, A., Clement, R., Isaac, P., Limousin, J.M., Linderson, M.L., Meir, P., Martin‐StPaul, N., & Wingate, L. (2017). How do leaf and ecosystem measures of water‐use efficiency compare? New Phytologist, 216, 758-770 Gonzalez-Cascon, R., Jiménez-Fenoy, L., Verdú-Fillola, I., & Martín, M.P. (2017). Short communication: Aqueous-acetone extraction improves the drawbacks of using dimethylsulfoxide as solvent for photometric pigment quantification in Quercus ilex leaves. 2017, 206

Referee #2 comment: To my opinion, this work has a high potential to bring to a better understanding of the ecosystem physiological response through hyperspectral and EC measurements, however, several changes are required: - All the graphs (except to figures 2&3) do not mention the fertilization treatments, maybe this addition can explain part of the variance in the graphs.

Authors' response: We think as the Referee #2 that the analysis of the response to the fertilization of the ecosystem is indeed an interesting point, which is the objective of the experimental effort. However, this is not the aim of this manuscript. We'd rather exploit the variability induced by fertilization to test the robustness of the retrieval method to different conditions. We have stressed this idea now in the methods' section; however, we have also extended the discussion to explain which parameters were estimated precisely enough to discriminate the effects induced by fertilization.

Referee #2 comment: - It looks like the summer measurements are not responding to the model, maybe the authors should consider excluding these results from the model, or at least to model them separately.

Authors' response: We agree with the Referee #2 in the fact that larger uncertainties occur during the dry period. The reasons for this are analyzed in the discussion section. We think that, rather than excluding this period it is important to include it in order to understand the potential risks and problems related with the study of this and similar ecosystems during the dry season. Notice that the inversion of the model is independently carried out for each date and tower; thus the presence of these data does not compromise the retrievals in other campaigns/periods. During evaluation, these data will be kept together to represent the total performance of the method. But we will stress this aspect as suggested by Referee #2.

Referee #2 comment: - In general, the discussion is mainly explaining the technical reasons for the model behavior against the other parameters. Maybe connecting the model to the actual physiology measured in the field will lead to a better understanding of the model strengths and weaknesses.

Authors' response: In response to a previous Referee #2 comment, we explained that the overarching goal of this manuscript, which is contributing to fill the knowledge gap about spatiotemporal distributions of key functional vegetation traits controlling carbon and water exchange in terrestrial biosphere modeling. We also explained that the scope of this work limits to the proposition and evaluation of a methodology that, when applied in numerous eddy covariance stations might eventually allow global up-scaling of these traits' distributions. In order to improve the understanding of the relevance and the aim of this manuscript, we will extend the discussion section, especially the first part, to stress the potential of this method and the need to test it in more and different ecosystems. We will stress that we demonstrated that the method is applicable in 3 different eddy covariance systems and with multiple imagery. We therefore think the method can be generally used, and the fact that it has been tested in a challenging ecosystem suggests that it could better perform in other sites where model assumptions are better met. The next steps will be an application on multiple sites with multiple hyperspectral imageries as soon as they will be available from recent or forthcoming space mission such as EnMAP, PRISMA, SBG, and/or DESIS, among others.

Regarding the part of the discussion on the uncertainties, we will try to streamline this

section to meet the comment of Referee #2, but we also think that an open discussion on the uncertainties is very useful for the community. Some of the uncertainties discussed are specific of the ecosystem under study but not exclusive and a can affect also grasslands, or other ecosystems structurally heterogeneous. For example, it is well known that unidimensional radiative transfer models do not accurately represent canopies with strong geometrical scattering components due to the presence of occluding volumes. It is also known that the absorption coefficients and the refractive index used by leaf radiative transfer models are effective averages determined from different species. Thus, the properties of some types of vegetation might not be not always accurately represented. Our manuscript does not cover a large and diverse range of ecosystems, but we deal with problems that, with some differences, can be found in other remote sensing studies and sites. We will reinforce this idea in the discussion, and we will reinforce the need of thorough evaluation of these estimates. In this context, we will also strength the connection of our results with the phenology or the physiology of the site in order to discuss the reliability of the estimates and limitations of the method. However, notice that this is not the aim of this manuscript, and more complete and dedicated research has been carried out in parallel works using additional datasets such as denser time series of other measurements. Moreover, a deeper analysis of the model senSCOPE and its connection with physiology both measured and also simulated by the original model SCOPE is presented in the senSCOPE manuscript (Pacheco-Labrador et al., 2020).

References:

Pacheco-Labrador, J., El-Madany, T.S., van der Tol, C., Martín, M.P., Gonzalez-Cascon, R., Perez-Priego, O., Guan, J., Moreno, G., Carrara, A., Reichstein, M., & Migliavacca, M. (2020). senSCOPE: Modeling radiative transfer and biochemical processes in mixed canopies combining green and senescent leaves with SCOPE. bioRxiv, 2020.2002.2005.935064

Short comments through the MS:

- Line 77: Reference is required.

Authors' response: A reference will be added

- Line 79-80: This is a very simplified assumption. many works demonstrated that the atmospheric demand is highly relevant to the transpiration and stomatal response.

Authors' response: The statement will be rephrased to acknowledge also this fact.

- Line 86-87: Reference is required.

Authors' response: A reference will be added

- Line 112: Authors should consider referring to Fu et al. (2020), PCE "Estimating photosynthetic traits from reflectance spectra: A synthesis of spectral indices, numerical inversion, and partial least square regression". Authors' response: Thanks for the suggestion, we have considered the contents of this manuscript and we will include a comment to it in this part of the introduction.

- Line 215-216: WC is a tricky parameter; the leaf relative water content is a more reliable parameter in terms of plant water status.

Authors' response: This parameter is measured since it is one of the input parameters of the leaf radiative transfer model of senSCOPE. Its comparison with estimates is not presented in the manuscript but we confirmed that due to the lack of spectral reflectance data in the SWIR bands this parameter could not be constrained. This was commented in an early version of the manuscript but this comment was removed at some point during the preparation of the submitted document. We will bring it back to the results section as "The lack of information in the short wave infrared prevented adequate constrain of Cw (not shown)."

- Line 226: it is not clear how N content was measured please explain or add a reference.

Authors' response: We will improve the description of the field data used in this

manuscript and on how these are integrated to ecosystem level. N determination is now explained in a dedicated supplementary material

- Line 226: which model was used? Reference.

Authors' response: This model was produced for time series of field data available in the study site. We will describe this analysis in a new supplementary material section.

- Line 230: Please note, if it is possible, if the estimation of Cab was done from estimated Nmass or only from measured values.

Authors' response: N was measured in all the campaigns where Cab of the grassland was not; therefore we used the observed relationship in the study site during additional campaigns (not concurrent to airborne overpasses) where grass N and Cab were simultaneously measured to gap-fill Cab in the campaigns where it was not available. This is now more extensively described in the dedicated supplementary material.

- Fig. 4: Fig. 4 please fit the letters in the legend to the figure.

Authors' response: The references to the subplots in the caption will be corrected.

- Fig. 6: to see all the points on the graph and avoid overlap, the authors should make them a bit transparent.

Authors' response: We will add transparency to the points in the plots.

- Fig. 7 please add parameters to the fitted curve and RMSE value.

Authors' response: Curve and parameters will be added. Figure 7 assess the retrievals of the soil resistance to evaporation from the pore space (rss) against soil moisture content. We acknowledge that this parameter is also potentially loose in the inversion, since its effect on the model outputs can saturate above some threshold (for Pacheco-Labrador et al., 2019); and in fact, the relationships between rss and soil moisture content presented in Figure 7 are poorly fit due to the presence of extreme values. Aware of this fact, we have implemented and tested a third step in the inversion where

the relationship presented in Figure 7 is used as a prior to repeat the inversion carried out in Step #2. This leads to a much closer fit of the relationship between rss and soil moisture and more importantly, has little effect on the retrieval of Vcmax and m. This process was suggested in the discussion of the manuscript, but not carried out. We will include these results in the new version of the manuscript to show that a more robust relationship can be obtained.

- Line 565: replace "response".

Authors' response: Thanks, "response" was duplicated and will be removed.